# Transversal functional connectivity and scene-specific processing in the human entorhinal-hippocampal circuitry

Xenia Grande[1,2]*, Magdalena M Sauvage[3,4,5], Andreas Becke[1,2], Emrah Düzel[1,2,6], David Berron[2,5,7]*

[1]Institute of Cognitive Neurology and Dementia Research, Otto von Guericke University Magdeburg, Magdeburg, Germany; [2]German Center for Neurodegenerative Diseases, Magdeburg, Germany; [3]Functional Architecture of Memory Department, Leibniz-Institute for Neurobiology, Magdeburg, Germany; [4]Functional Neuroplasticity Department, Otto-von-Guericke University Magdeburg, Magdeburg, Germany; [5]Center for Behavioral Brain Sciences, Otto-von-Guericke University Magdeburg, Magdeburg, Germany; [6]Institute of Cognitive Neuroscience, University College London, London, United Kingdom; [7]Clinical Memory Research Unit, Department of Clinical Sciences Malmö, Lund University, Lund, Sweden

*For correspondence:
xenia.grande@dzne.de (XG);
david.berron@dzne.de (DB)

**Competing interest:** The authors declare that no competing interests exist.

**Abstract** Scene and object information reach the entorhinal-hippocampal circuitry in partly segregated cortical processing streams. Converging evidence suggests that such information-specific streams organize the cortical – entorhinal interaction and the circuitry's inner communication along the transversal axis of hippocampal subiculum and CA1. Here, we leveraged ultra-high field functional imaging and advance Maass et al., 2015 who report two functional routes segregating the entorhinal cortex (EC) and the subiculum. We identify entorhinal subregions based on preferential functional connectivity with perirhinal Area 35 and 36, parahippocampal and retrosplenial cortical sources (referred to as $EC_{Area35-based}$, $EC_{Area36-based}$, $EC_{PHC-based}$, $EC_{RSC-based}$, respectively). Our data show specific scene processing in the functionally connected $EC_{PHC-based}$ and distal subiculum. Another route, that functionally connects the $EC_{Area35-based}$ and a newly identified $EC_{RSC-based}$ with the subiculum/CA1 border, however, shows no selectivity between object and scene conditions. Our results are consistent with transversal information-specific pathways in the human entorhinal-hippocampal circuitry, with anatomically organized convergence of cortical processing streams and a unique route for scene information. Our study thus further characterizes the functional organization of this circuitry and its information-specific role in memory function.

## Editor's evaluation

Grande and colleagues provide important new insights into how different regions of the entorhinal cortex functionally interact with specific cortical brain areas and how, in turn, subregions of the entorhinal cortex interact with the hippocampus during 'scene' and 'object' processing. The study is well-motivated, well-designed, and provides convincing evidence using appropriate methodology. This paper is relevant to cognitive neuroscientists with an interest in the entorhinal cortex – hippocampal pathways and 'scene' and 'object' representation in the medial temporal lobe.

# Introduction

Entorhinal and hippocampal subregions form a critical functional circuitry that binds cortical information into cohesive representations (*Eichenbaum et al., 2007*; *Ritchey et al., 2015*). The interaction of the entorhinal-hippocampal circuitry with large-scale cortical information streams and the circuitry's inner communication are key to the formation of these cohesive representations. Here, we advance insight into how the human entorhinal cortex (EC) receives information from cortical streams and how information proceeds between the EC and the transversal axis of hippocampal subiculum and CA1 (here referred to as transversal sub/CA1 axis). These insights are relevant to our understanding of the circuitry's fundamental role in cognitive functions such as episodic memory.

Large-scale cortical information streams, that originate in the visual 'Where' and 'What' pathways and process scene and object information (*Berron et al., 2018*; *Haxby et al., 1991*; *Ranganath and Ritchey, 2012*; *Ritchey et al., 2015*; *Ungerleider and Haxby, 1994*), map onto the EC in a complex manner and define functional EC subregions. [Note, in light of confusing nomenclature, here we adhere to scene and object information - elsewhere referred to as contextual, spatial or "Where" and content, non-spatial, item or "What" information, respectively.] Recent rodent research updates the former conception of a parallel mapping of scene and object information via parahippocampal and perirhinal cortices onto medial versus lateral EC subregions (*cf.* posterior-medial versus anterior-lateral EC subregions as the human homologues; *Maass et al., 2015*; *Navarro Schröder et al., 2015*). Instead of a strict parallel mapping, profound cross-projections exist from the parahippocampal cortex towards the perirhinal cortex and the lateral EC (*Nilssen et al., 2019*). In accordance, information seems to converge in the rodent lateral EC (*Doan et al., 2019*). The update, thus, implies a more complex functional organization than parallel scene and object information mapping. Moreover, this advance highlights the retrosplenial cortex as an additional source to convey information directly from the cortical scene processing stream onto the EC. The retrosplenial cortex projects to the medial EC and, like the parahippocampal cortex, is part of the scene processing stream (e.g. involved in scene translation; *Vann et al., 2009*; *Nilssen et al., 2019*; *Witter et al., 2017*). The update, furthermore, evokes the question how cortical sources of information uniquely map onto the EC and which kind of information is processed in the resulting functional EC subregions.

Within the entorhinal-hippocampal circuitry, an important direct way of communication exists between the EC and hippocampal subiculum and CA1. How functional EC subregions communicate towards the transversal sub/CA1 axis in humans is, however, unclear. Similarly, the extent to which specific scene and object information processing routes might emerge, despite information convergence in the EC, is unknown. On one hand, rodent research indicates a transversal organization where scene and object information is processed along two anatomically wired routes, the medial EC – distal subiculum – proximal CA1 route and the lateral EC – proximal subiculum – distal CA1 route, respectively (*Witter et al., 2017*; note sparse functional evidence in the subiculum: *Ku et al., 2017*; *Cembrowski et al., 2018*; but frequent reports in the rodent CA1 region: *Henriksen et al., 2010*; *Nakamura et al., 2013*; *Igarashi et al., 2014*; *Nakazawa et al., 2016*; *Beer et al., 2018*). Initial functional and structural connectivity data also indicate such a transversal connectivity profile in humans (*Maass et al., 2015*; *Syversen et al., 2021*). In accordance, scene information seems to be preferentially processed in the distal subiculum (*Dalton et al., 2018*; *Dalton and Maguire, 2017*; *Zeidman et al., 2015*) and hints exist for preferential object processing at the subiculum/CA1 border (*Dalton et al., 2018*). On the other hand, anatomical projections in the monkey show a longitudinal profile on top of the transversal profile with mainly the anterior-lateral and posterior-lateral entorhinal portions projecting to the distal subiculum – proximal CA1 and proximal subiculum – distal CA1, respectively (*Witter and Amaral, 2020*). According to information convergence in the EC, a recent report finds convergence along the rodent transversal CA1 axis (*Vandrey et al., 2021*). In humans, visual stream projections towards the entorhinal-hippocampal circuitry similarly suggest convergence of scene and object information in the subiculum/CA1 border region but preserved scene processing in the distal subiculum (*Dalton and Maguire, 2017*). A detailed examination of the latter hypothesis is, however, lacking. The diversity of findings emerging from the literature calls for a thorough investigation to elucidate whether multiple transversal processing routes exist within the human entorhinal-hippocampal circuitry.

To summarize, our conception of how information travels towards the entorhinal-hippocampal circuitry underwent key changes which warrant an extensive exploration of the circuitry's functional organization. First, rodent research shows that there is no strict parallel mapping of cortical information

from the perirhinal and parahippocampal cortex towards the EC. Second, information seems to converge already before the hippocampus. These changes add to several knowledge gaps. First, it is unclear in which subregions of the entorhinal-hippocampal circuitry scene and object information are processed. The general connectivity patterns in the human entorhinal-hippocampal circuitry have not yet been directly related to information processing. Moreover, it is unclear how scene information from the retrosplenial cortex maps onto the human EC as a critical source of the cortical scene processing stream. Hence, it is also unclear how retrosplenial information is communicated between the EC and the hippocampus. Finally, it remains elusive whether a transversal functional segregation can be extended towards the human CA1 region in analogy to the rodent literature.

Here, we leverage ultra-high field 7 Tesla functional imaging (fMRI) data and advance the earlier findings on human entorhinal subregions and a transversal intrinsic functional connectivity pattern in the subiculum (*Maass et al., 2015*). With a combination of functional connectivity and information processing analyses, we seek to answer two sets of questions. Regarding functional connectivity, we ask where the parahippocampal, perirhinal, and retrosplenial cortical sources uniquely map onto the human EC and how these functionally connected routes continue between EC subregions and the transversal sub/CA1 axis. Regarding information processing, we ask whether and where scene and object information are specifically processed in the EC and along the transversal sub/CA1 axis. We test the hypotheses of (1) a transversal functional connectivity pattern and (2) multiple information processing routes within the entorhinal-hippocampal circuitry. Thus, following the updated conception of a non-parallel cortical scene and object information mapping onto the EC in rodents, we will show how cortical information streams map onto the EC in humans. This mapping will then be our detailed starting point to investigate the functional connectivity and information processing within the entorhinal-hippocampal circuitry.

## Results

We seek to comprehensively investigate functional connectivity within the entorhinal-hippocampal circuitry and the contribution of cortical scene and object information processing streams. In an initial step, we identified where cortical sources map uniquely onto the entorhinal cortex (building upon *Maass et al., 2015*). The identified entorhinal subregions are based on their voxel's preferred intrinsic functional connectivity with the retrosplenial cortex, parahippocampal cortex, perirhinal Area 35 or Area 36 regions ('sources'). Note, that we evaluate both perirhinal subregions, Area 35 and Area 36 as separate sources as accumulating evidence suggests their structural and functional distinction (e.g. *Berron et al., 2021*; *Burwell, 2000*; *Suzuki and Naya, 2014*; *van Strien et al., 2009*). Next, we evaluated the continuation of the functional connectivity streams within the entorhinal-hippocampal circuitry and examined the intrinsic functional connectivity pattern between the identified entorhinal subregions ('seeds') and hippocampal subiculum and CA1 in the hippocampal body. Therefore, temporal fluctuations of BOLD signal were correlated in a seed-to-voxel manner within each participant. The resulting statistical correlational maps were aligned between participants. Repeated measures ANOVAs were calculated on connectivity preferences with seeds and transversal segments as factors to determine statistical differences in connectivity topography. All functional connectivity analyses were performed on the dataset after task-related effects have been regressed out, creating a dataset that resembles resting-state data (*Gavrilescu et al., 2008*; *Maass et al., 2015*).

Finally, we identified the corresponding bias in scene (here operationalized with room stimuli) and object information processing within entorhinal subregions and along the transversal sub/CA1 axis using the same dataset. Therefore, we extracted parameter estimates from a mnemonic discrimination task with scene and object conditions from aligned statistical maps across participants. Repeated measures ANOVAs were calculated on parameter estimates in entorhinal subregions and transversal sub/CA1 segments to determine biases in information processing within the entorhinal-hippocampal circuitry.

In the following, we first describe the four obtained entorhinal seeds and display the intrinsic functional connectivity pattern with the entorhinal seed regions along the transversal sub/CA1 axis. Thereafter, we report the information processing characteristics of the entorhinal and hippocampal subregions. Note that all results have been obtained with independent analyses in the left and right hemispheres. The largely similar left hemisphere results can be found in appendix 1. Source data and statistical maps are provided under Grande, X., Berron, D. (2022). Open Science Framework.

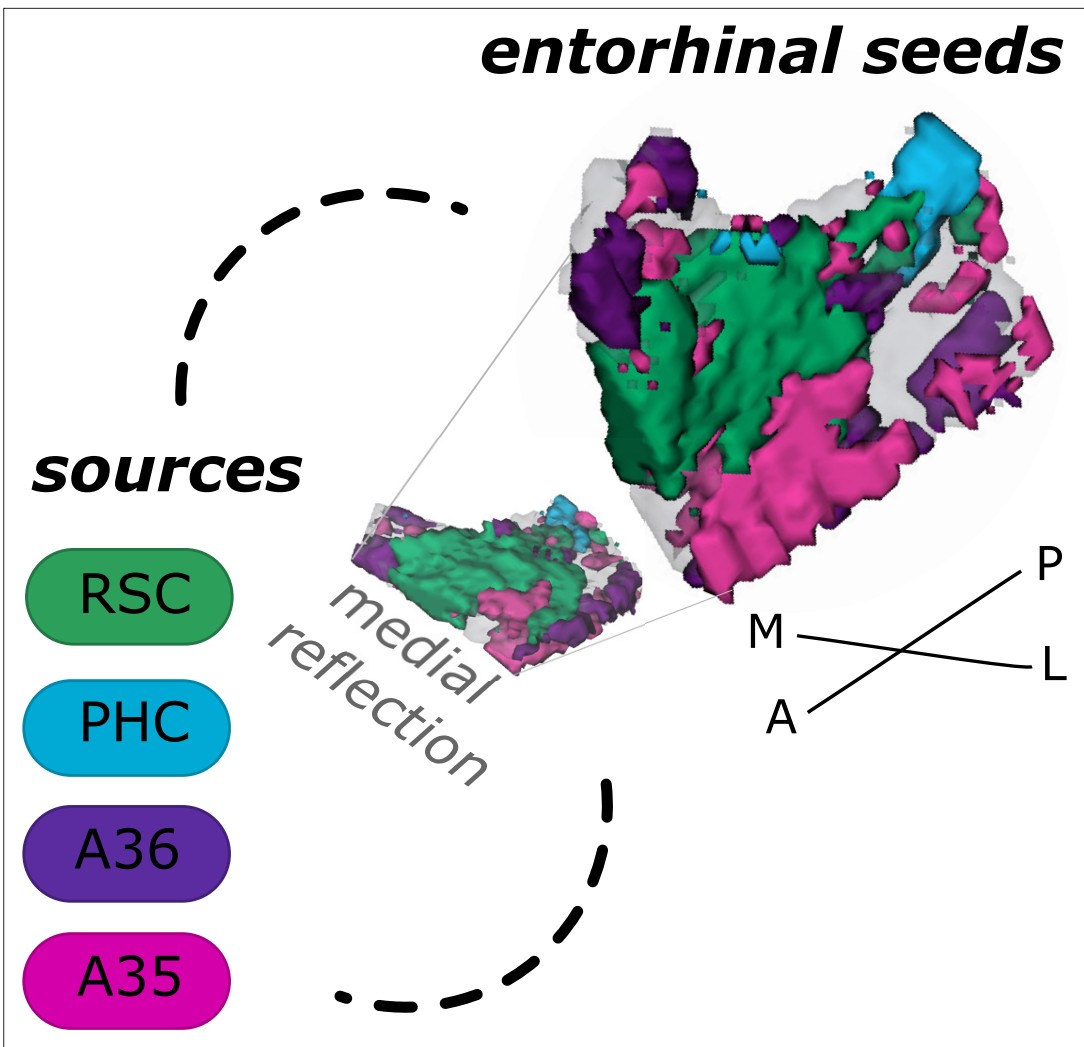

**Figure 1.** Entorhinal seed regions based on connectivity preferences to cortical regions. Displayed is the right EC as a 3D image with colored seed regions. The seed regions have been identified based on a source-to-voxel functional connectivity analysis and resulting connectivity preference to either the right retrosplenial cortex (RSC, green), parahippocampal cortex (PHC, blue), Area 36 (A36, purple), or Area 35 (A35, pink) sources. Note that preferences to Area 36 are best visible from a medial perspective on the EC as depicted in the medial reflection. Seed regions have been determined based on the thresholded (T>3.1) maximum voxels across four one-sample T-tests at group level, one per source, sample size n = 32. M – medial; L – lateral; A – anterior; P – posterior.

ID 9v3qp. Source Data from Functional Connectivity and Information Processing in the Entorhinal-Hippocampal Circuitry. https://osf.io/9v3qp.

## Four cortical sources divide the EC in retrosplenial-, parahippocampal, Area 35- and Area 36-based seeds

The four entorhinal subregions that we later used as seeds to determine the topography of entorhinal-hippocampal connectivity are based on intrinsic functional connectivity preferences with either the parahippocampal cortex, the retrosplenial cortex, perirhinal Area 36 or Area 35. These cortical regions are in general concordance with *Maass et al., 2015* but consider recent advances that put forward the retrosplenial cortex as a critical source from the cortical scene processing stream (*Nilssen et al., 2019*) and evaluate perirhinal Area 35 and 36 separately.

Based on functional connectivity preferences with the four sources - parahippocampal cortex (*Source code 8*), retrosplenial cortex (*Source code 7*, Area 36 (*Source code 6*), and Area 35 (*Source code 5*) - we obtained four entorhinal seeds. The seeds refer to different parts of the EC whose voxels

expressed preferential functional connectivity to either cortical source. For the $EC_{PHC-based}$ seed, the majority of voxels can roughly be described as clustering in the posterior-medial entorhinal portion, for the $EC_{RSC-based}$ seed in the anterior-medial portion, for the $EC_{Area35-based}$ seed in the anterior-lateral portion and for the $EC_{Area36-based}$ seed in the posterior-lateral entorhinal portion (see appendix 2 for exact voxel counts). Note that both perirhinal-based entorhinal seeds extended along the anterior to posterior axis such that the $EC_{Area35-based}$ progressed more along the outer EC (i.e. laterally, with a main focus anteriorly) and the $EC_{Area36-based}$ along the inner EC (i.e. medially, with a main focus posteriorly, see *Figure 1* and the medial reflection of the EC seeds). It is important to note that these are rough qualitative descriptions of the main clusters, without quantification or an established relationship to coherent cytoarchitectonic regions. We will therefore continue to refer to them as $EC_{RSC-based}$, $EC_{PHC-based}$, $EC_{Area35-based}$ and $EC_{Area36-based}$ seeds.

## Distal subiculum is functionally connected with the $EC_{PHC-based}$ seed while the subiculum/CA1 border is connected with $EC_{RSC-based}$ and $EC_{Area35-based}$ seeds

Following the characterization of entorhinal seeds, we focused on the functional connectivity between these entorhinal subregions and hippocampal subiculum and CA1 to test the hypothesis of a transversal functional connectivity pattern. We predicted that while some EC subregions have a preference to functionally connect with the subiculum/CA1 border, others preferentially connect with the distal subiculum and proximal CA1. In the previous step, we identified EC subregions based on unique cortical source contributions. Therefore, our predictions remained in accordance with *Maass et al., 2015*: We expected that the EC subregion preferentially connected with the parahippocampal cortex ($EC_{PHC-based}$ seed) maps towards the distal subiculum and EC subregions connected with the perirhinal cortex ($EC_{Area35-based}$ seed, $EC_{Area36-based}$ seed) map towards the proximal subiculum, a mapping that we predicted to be extended towards the distal CA1.

When extracting estimates of connectivity preferences across individuals from proximal and distal hippocampal subfield segments for either entorhinal seed, repeated measures ANOVAs revealed significant seed X segments interaction effects along the transversal sub/CA1 axis (see *Figure 2*; subiculum: $F(12,372) = 19.561$; $p<0.001$; CA1: $F(6,186) = 3.212$; $p=0.024$).

In the subiculum, additional repeated measures ANOVAs showed that the $EC_{Area35-based}$ ($F(4,124) = 8.913$; $p_{FDR} <0.001$), $EC_{RSC-based}$ ($F(4,124) = 10.538$; $p_{FDR} <0.001$) and $EC_{PHC-based}$ ($F(4,124) = 42.201$; $p_{FDR} <0.001$) seeds displayed a significant main effect across the transversal subiculum segments. These differential functional connectivity preferences across the transversal axis of the subiculum interacted significantly in a subsequent repeated measures ANOVA ($EC_{PHC-based}$ versus $EC_{RSC-based}$ seed preference interaction: $F(4,124) = 46.452$; $p_{FDR} <0.001$; $EC_{PHC-based}$ versus $EC_{Area35-based}$ seed preference interaction: $F(4,124) = 35.208$; $p_{FDR} <0.001$). This pattern provides statistical evidence for an increase in preferential functional connectivity with the $EC_{PHC-based}$ seed towards the distal portion of the subiculum while the preferential functional connectivity with the $EC_{Area35-based}$ as well as the $EC_{RSC-based}$ seeds rather increased towards the proximal portion of the subiculum.

In hippocampal CA1, additional repeated measures ANOVAs showed that the connectivity preference towards the $EC_{RSC-based}$ seed displays a significant main effect across the transversal axis of CA1 ($F(2,62) = 10.489$; $p_{FDR} <0.001$). In distal CA1, the preferential functional connectivity with the $EC_{RSC-based}$ seed was higher than in the proximal portion of CA1. In right CA1, a similar but weaker transversal pattern was observed for connectivity preferences with the $EC_{Area35-based}$ ($F(2,62) = 4.146$; $p_{FDR} = 0.041$; note in the left hemisphere a comparable transversal pattern was observed for the $EC_{PHC-based}$ and $EC_{RSC-based}$ portions, see appendix 1).

Thus, in the entorhinal-hippocampal circuitry, voxels in the distal subiculum were preferentially functionally connected with the $EC_{PHC-based}$ portion whereas voxels in the subiculum/CA1 border were preferentially connected with more anterior EC portions ($EC_{RSC-based}$ and $EC_{Area35-based}$).

## Distal subiculum and $EC_{PHC-based}$ exhibit higher functional activity in the scene condition while other subregions show no significant difference between conditions

Besides the intrinsic functional connectivity patterns within the entorhinal-hippocampal circuitry, we also examined the characteristics of scene and object information processing to test the hypothesis

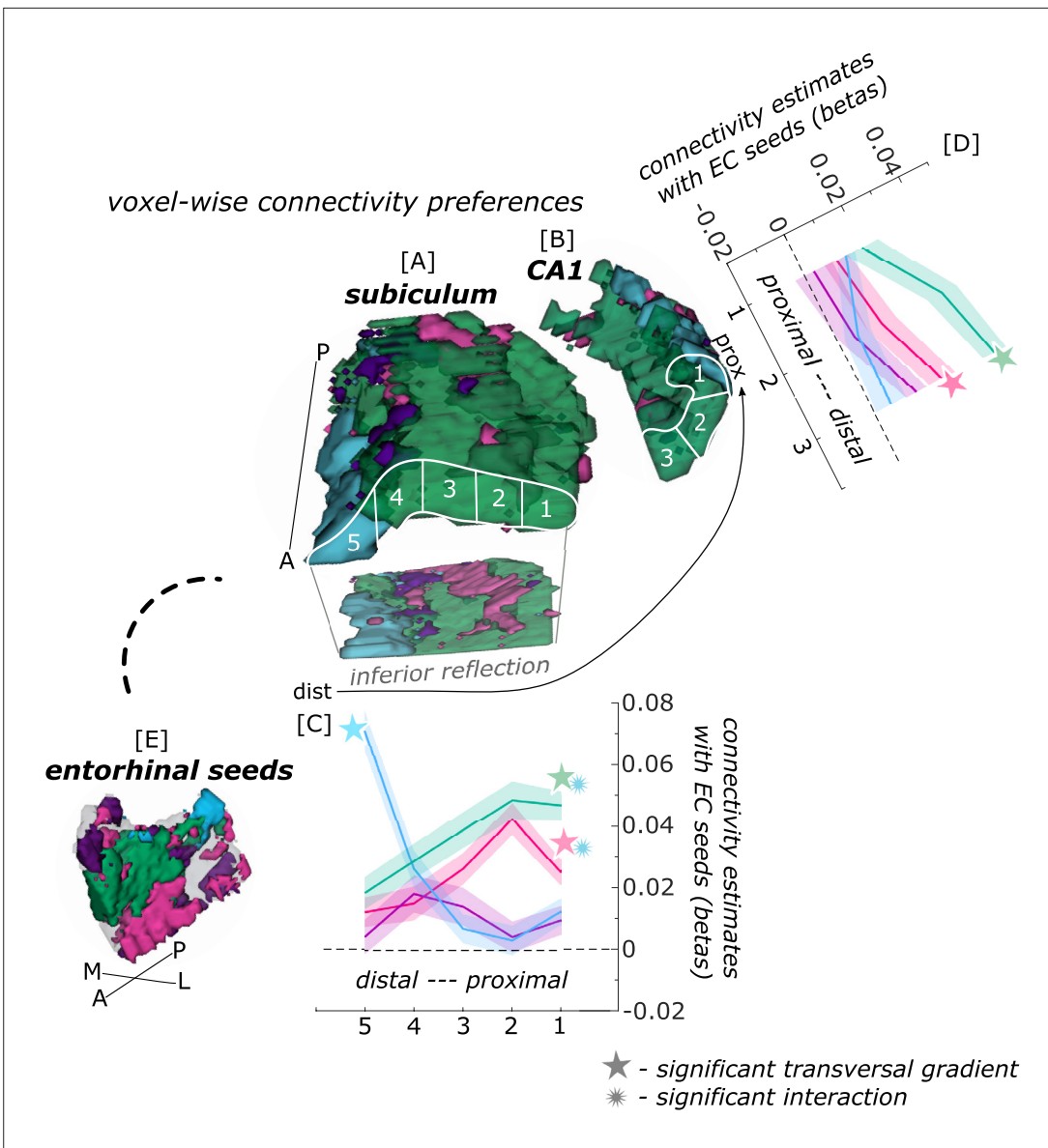

**Figure 2.** Functional connectivity preferences to entorhinal seeds along the transversal axis of subiculum and CA1. Displayed are the results of a seed-to-voxel functional connectivity analysis between the displayed right entorhinal seeds and the right subiculum and CA1 subregion. The 3D figure displays voxel-wise connectivity preferences to the entorhinal seeds (color coded to refer to the respective entorhinal seed [**E**]) on group level ([**A**] - subiculum; [**B**] - CA1; maps for connectivity preferences: **Source code 13** - $EC_{Area35-based}$, pink; **Source code 14** - $EC_{Area36-based}$, purple; **Source code 16** - $EC_{PHC-based}$, blue; **Source code 15** - $EC_{RSC-based}$ seed, green). Note that preferences to the $EC_{Area35-based}$ seed (pink) are located mainly in the inferior subiculum and CA1 and are therefore best visible in the inferior reflection. To display mean connectivity preferences across participants along the transversal sub/CA1 axis, beta estimates were extracted and averaged from equally sized segments from proximal to distal ends (five segments in subiculum [**A**], three segments in CA1 [**B**]; schematized in white on the 3D figures) on each coronal slice and averaged along the longitudinal axis. Repeated measures ANOVAs revealed significant differences in connectivity estimates along the transversal axis of CA1 [**D**] and subiculum [**C**] with interaction effects in the subiculum. Displayed significances were obtained by FDR-corrected post-hoc tests and refer to $p<0.05$. Shaded areas in the graphs refer to standard errors of the mean, sample size n = 32. EC – entorhinal; M – medial; L – lateral; A – anterior; P – posterior; prox – proximal; dist – distal. **Figure 2—source data 1** contains individual connectivity estimates per subregion (Sub – subiculum and CA1, respectively) and seed ($EC_{RSC-based}$ – RSCECseed, $EC_{Area35-based}$ – A35ECseed, $EC_{PHC-based}$ – PHCECseed, $EC_{Area36-based}$ – A36ECseed) for each transversal segment (1–5 or 1–3, respectively from proximal to distal).

*Figure 2 continued on next page*

*Figure 2 continued*

The online version of this article includes the following source data for figure 2:

**Source data 1.** Individual functional connectivity estimates to right entorhinal seeds, extracted from right subiculum and CA1 transversal segments.

of multiple information processing routes within the entorhinal-hippocampal circuitry. We predicted a route of specific scene processing and another route of convergent information processing. Following the proposal by *Dalton and Maguire, 2017* and the updated cross-projections from the scene to the object information processing stream (*Nilssen et al., 2019*), we expected scene processing in the distal subiculum. The updated parahippocampal cross-projections imply convergence wherever specific object processing had been expected previously. Thus, we explored whether any entorhinal-hippocampal subregions still process object information specifically. However, we largely expected to find evidence consistent with convergent processing of scene and object information within the entorhinal-hippocampal circuitry.

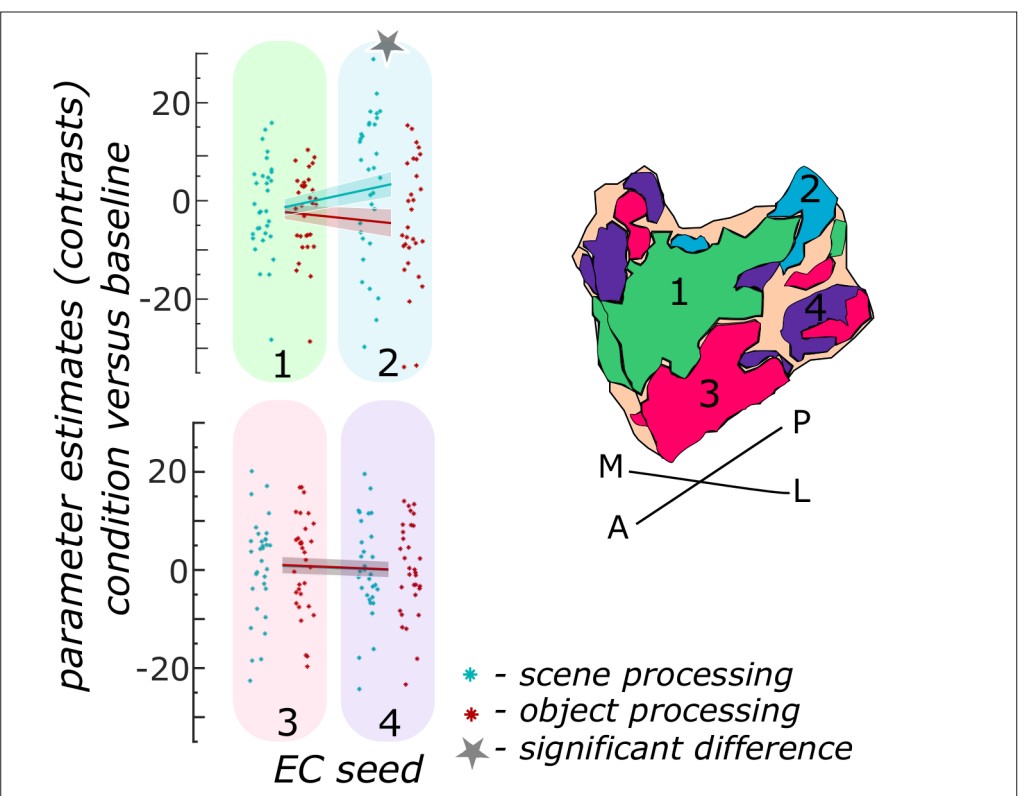

**Figure 3.** Functional activity during scene and object conditions in entorhinal seed regions. Displayed are the extracted parameter estimates for the object condition versus baseline contrast ('object information processing', red) and the scene condition versus baseline contrast ('scene information processing', cyan) from each entorhinal seed region per individual (dots) and summarized across individuals (lines). A schematic depiction of the respective entorhinal seed regions is displayed by a 3D drawing of the right EC. A repeated measures ANOVA revealed a significant interaction between condition and seed region. The displayed significant difference is obtained with FDR-corrected post-hoc tests and refers to $p<0.05$. During the object condition, participants were presented with 3D rendered objects on screen, during the scene condition with 3D rendered indoor rooms and during the baseline condition they saw scrambled pictures. The shaded area around the lines refers to standard errors of the mean, sample size n = 32. EC – entorhinal cortex; M – medial; L – lateral; A – anterior; P – posterior. *Figure 3— source data 1* contains extracted parameter values per individual and EC seed (isthmuscingulate – ECRSC-based, Area 35 – ECArea35-based, Area 36 – ECArea36-based, PHC – ECPHC-based seed) for the object versus baseline and scene versus baseline contrasts.

The online version of this article includes the following source data for figure 3:

**Source data 1.** Individual paramenter estimates for scene and object processing in right entorhinal seed regions.

We first focused on the entorhinal seed regions. When extracting task-related parameter estimates from object and scene conditions, a repeated measures ANOVA showed a significant interaction between region and information type (object versus scene, $F(3,93) = 20.9267$; $p<0.001$). Post-hoc t-tests revealed that only in the $EC_{PHC-based}$ seed region functional activity in the scene condition was significantly higher than in the object condition ($p_{FDR}<0.001$), while in the remaining three entorhinal seed regions no significant difference between scene and object conditions existed ($EC_{Area35-based}$: $p_{FDR}$ = 0.9129; $EC_{Area36-based}$: $p_{FDR}$ = 0.9129; $EC_{RSC-based}$: $p_{FDR}$ = 0.5646; see *Figure 3*).

When extracting task-related parameter estimates for scene and object conditions from proximal and distal segments of hippocampal subregions within each participant, we found a significant interaction between transversal segments and information type only in the subiculum ($F(4,124) = 15.994$; $p<0.001$) and not in CA1 ($F(2,62) = 2.553$; $p = 0.105$) as revealed by repeated measures ANOVAs. Post-hoc T-tests showed significantly higher functional activity in the scene than object condition (both $p_{FDR} <0.001$) only in the distal subiculum segments. In all other segments along the subiculum transversal axis, there was no significant difference in functional activity related to scene and object conditions (all $p_{FDR} = 0.1222$; see *Figure 4*).

## Discussion

This study aims to advance insight into the organizational principles of information processing within the entorhinal-hippocampal circuitry and the circuitry's embedding in designated cortical processing streams. Leveraging ultra-high field 7 Tesla fMRI, we find a resemblance between the intrinsic functional connectivity pattern and subregional biases in scene information processing in the entorhinal-hippocampal circuitry. In the EC, we observe a topographical mapping of regions from the cortical scene and object information processing streams, including the retrosplenial, parahippocampal and perirhinal Area 35 and Area 36 cortices. This mapping continues to determine a transversal organization of information processing routes between the EC and the human hippocampal circuitry. Our results unify previous evidence and uncover novel features in the human brain that can be a window into the circuitry's critical role in memory function and decline.

### Scene information is processed within an $EC_{PHC-based}$ – distal subiculum route

We identified regions in the entorhinal-hippocampal circuitry that are dedicated to process scene information. These regions consisted of two functionally connected portions: the $EC_{PHC-based}$ and the distal subiculum. The subiculum showed a transversal difference in intrinsic functional connectivity with a preference to the $EC_{PHC-based}$ in its distal portions (of note: the $EC_{PHC-based}$ was defined by entorhinal voxels with preferential functional connectivity to the parahippocampal cortex). Importantly, the distal subiculum and the $EC_{PHC-based}$ were the only studied entorhinal-hippocampal subregions that exhibited functional activity specifically in the scene condition (see appendix 5 for information processing in cortical source regions).

These findings provide clear evidence for a hypothesized transversal difference in scene information processing within the human subiculum. Our data also replicate the earlier functional and structural connectivity reports in humans as well as anatomical findings of a route between posterior-medial EC (based on parahippocampal connectivity) and distal subiculum (*Maass et al., 2015*; *Syversen et al., 2021*; *Witter et al., 2000*). The scene information processing bias has mainly been previously reported for the EC (in rodents, operationalized by spatial processing conditions: *Neunuebel et al., 2013*; *Keene et al., 2016*; in humans, operationalized by scene stimulus conditions: *Berron et al., 2018*; *Navarro Schröder et al., 2015*; *Reagh and Yassa, 2014*; *Schultz et al., 2015*). In animal studies, the importance of the subiculum as a translator of hippocampal information towards the entorhinal and other cortical structures is increasingly acknowledged (*O'Mara, 2006*; *Roy et al., 2017*). We here contribute to the sparse investigations regarding the nature of information processed along the transversal axis of the subiculum (see *Cembrowski et al., 2018*; *Ku et al., 2017*). Our observation is in line with the hypothesis that the distal subiculum is more involved in processing scenes than objects based on previous findings in the human brain. While the subiculum in general was associated with scene discrimination (*Hodgetts et al., 2017*), a growing body of evidence relates particularly the medial hippocampus to scene processing. This entails two medial areas, the pre- and parasubiculum, that we

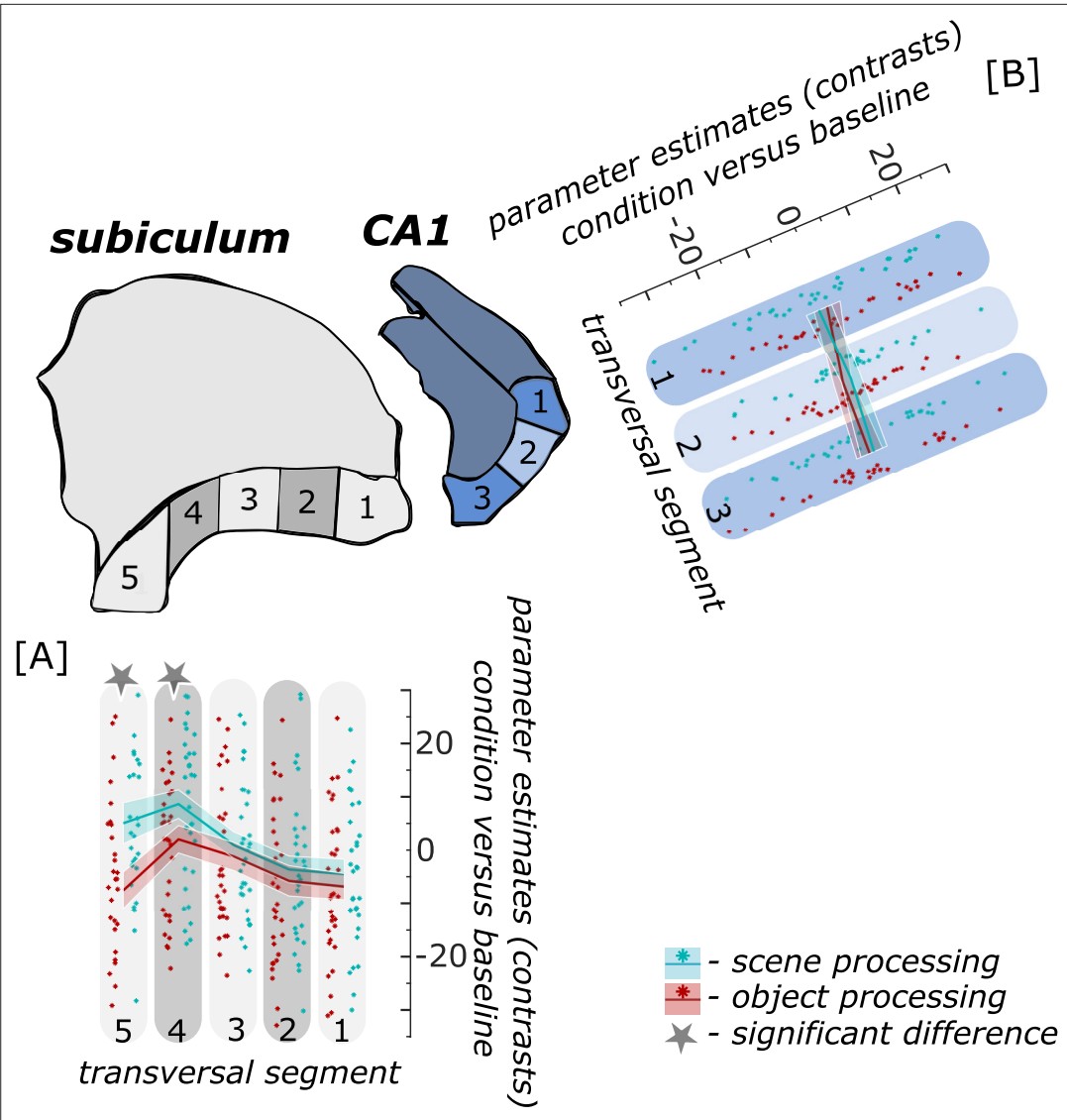

**Figure 4.** Functional activity during scene and object conditions along the transversal axis of subiculum and CA1. Displayed are the extracted parameter estimates for the object condition versus baseline contrast ('object information processing', red) and the scene condition versus baseline contrast ('scene information processing', cyan) from the respective transversal segments in the subiculum ([A] grey) and CA1 ([B] blue) per individual (dots) and summarized across individuals (lines). A schematic depiction of the respective transversal segment is displayed by a 3D drawing of the right subiculum and CA1 subregion. Repeated measures ANOVAs revealed a significant interaction between condition and seed region in the subiculum only. The displayed significant difference is obtained with FDR-corrected post-hoc tests and refers to $p < 0.05$. During the object condition, participants were presented with 3D rendered objects on screen, during the scene condition with 3D rendered indoor rooms and during the baseline condition they saw scrambled pictures. The shaded area around the lines refers to standard errors of the mean, sample size n = 32. **Figure 4—source data 1** contains extracted parameter values for each subregion (Sub – subiculum and CA1, respectively) per individual and transversal segment (1–5 and 1–3, respectively from proximal to distal) for the object versus baseline and scene versus baseline contrasts.

The online version of this article includes the following source data for figure 4:

**Source data 1.** Individual parameter estimates for scene and object processing in right transversal subiculum and CA1 segments.

attribute to the distal subiculum in our current segmentation. Especially the area that resembles the pre- (or here: distal) subiculum has been shown to be involved in scene construction (*Dalton et al., 2018*; *Zeidman et al., 2015*). Recently, a gradient with coarser voxel-wise autocorrelation signals in the medial hippocampus has been reported, a finding that implies larger representations in the distal subiculum (*Bouffard et al., 2022*). In the latter two studies, however, the authors did not specifically extract data from the transversal axis of hippocampal subfields. Our joint investigation of functional entorhinal-subiculum connectivity and type of information processing along the full transversal axis of the subiculum, is the first to show a clear preference of scene information towards the distal portion, in comparison to more proximal portions.

## Information processing is consistent with convergence within the anterior entorhinal portions – subiculum/CA1 border route

Our data revealed a route that did not show differences in scene and object information processing. Both, the $EC_{Area35-based}$ and the $EC_{RSC-based}$ portion exhibited preferential functional connectivity with the subiculum/CA1 border. Comparable levels of functional activity in scene and object conditions along these entorhinal-hippocampal routes are consistent with information convergence.

While we again confirm earlier findings and previously stated hypotheses, several features in our data are fundamentally novel. First, we provide initial human evidence for a functional connection between the $EC_{Area35-based}$ and the subiculum/CA1 border. Non-primate and primate anatomical data as well as ex vivo and in vivo structural connectivity data in humans show the possibility of information flow along that route (*Syversen et al., 2021*; *Witter et al., 2017*; *Witter and Amaral, 1991*; *Witter and Amaral, 2020*). Our results now underpin a functional relevance of that connection beyond the subiculum (for the subiculum see *Maass et al., 2015*). Our findings are derived based on a voxel-wise analysis, unconstrained by a priori selection of regions-of-interest. We thereby confirm the long-held proposal of a transversal functional organization in human subiculum and CA1.

Convergence of scene and object information is compatible with recent rodent work that shows joint coding of scene and object information (notably operationalized as spatial and non-spatial information, respectively) along CA1 and within the lateral EC (*Deshmukh, 2014*; *Doan et al., 2019*; *Vandrey et al., 2021*; *Wilson et al., 2013*; *Yeung et al., 2019*). Note that a supplemental analysis of information processing in the cortical source regions showed indeed, specific object processing in perirhinal source regions (see appendix 5). The lack of increased object processing in the anterior EC subregions and subiculum/CA1 border is thus likely not a result of increased noise in the object condition. Instead, increased object processing in perirhinal cortical source regions indicates subsequent convergence in entorhinal-hippocampal subregions, as hypothesized based on the updated cortical mapping scheme onto the EC.

Our results cannot confirm previous reports about higher functional activity for object than scene processing within these areas in the human brain (*Reagh and Yassa, 2014*; *Navarro Schröder et al., 2015*; *Berron et al., 2018*; also indicated in *Dalton et al., 2018* and *Schultz et al., 2015*). Neither did we observe proximodistal differences in CA1 for object versus scene information processing as suggested by several rodent studies (*Beer et al., 2018*; *Henriksen et al., 2010*; *Nakamura et al., 2013*; *Nakazawa et al., 2016*). Differences in experimental design and contrasts could have contributed to these discrepancies (i.e. specific object information processing versus convergence). Previous studies used a variety of different conditions to tackle scene and object information processing (e.g. objects in time versus objects in space in *Beer et al., 2018* or imagined objects on a 2D versus 3D grid in *Dalton et al., 2018*). In contrast to the current data, previous human studies did not derive functional data from specific, functionally defined entorhinal portions in the same dataset. As most previous studies were conducted in the light of the 'parallel mapping hypothesis', the related assumptions influenced the examined subregions, which may have altered the extracted measures.

Regarding the human proximal CA1, a firm conclusion is limited with our data. First, the functional connectivity results varied between hemispheres. In both hemispheres, proximal CA1 showed a different connectivity profile compared to distal CA1. However, even though statistically not significant, the preferences at the group level indicated increased functional connectivity with the $EC_{PHC-based}$ portion in the right proximal CA1 but with the $EC_{Area35-based}$ portion in the left hemisphere. Second, we do not prove similar information processing along the transversal CA1 axis. Instead, we find no significant difference in information processing along the transversal CA1 axis. As indicated in the previous

paragraph, we cannot rule out that our object versus scene processing conditions may not have been sensitive enough to tackle functional differences in CA1. Thus, future research will have to identify defining characteristics of information processing along the transversal CA1 axis in a less constraint manner to allow conclusions on distinct information processing in proximal CA1.

In addition, we observed an unreported resemblance in functional connectivity profiles of $EC_{RSC-based}$ and $EC_{Area35-based}$ portions in the anterior EC. The sources of these entorhinal portions are part of cortical scene and object processing streams, respectively (see also appendix 5 that shows increased scene processing in the retrosplenial and increased object processing in perirhinal cortical source regions). To our knowledge, the $EC_{RSC-based}$ portion has not yet been identified in earlier investigations. While anatomical projections from the retrosplenial to deep medial EC layers have been confirmed in rodents, they appear in the posterior EC (*Czajkowski et al., 2013*; *Sugar et al., 2011*). Very recently, *Syversen et al., 2021* found structural connectivity between the human retrosplenial cortex and the medial EC, but again not in the anterior part of the EC. The EC segmentation of Syversen and colleagues, however, followed different rules which may have contributed to differences in the topographical evaluation of the region. Also, structural and functional connectivity methods may yield different results, in particular as we identified EC subregions with a different set of cortical source regions. Under the assumption that retrosplenial connectivity defines the medial EC (*Witter et al., 2017*), the mapping of the $EC_{RSC-based}$ to the subiculum/CA1 border opposes conventional views that the medial EC communicates with the distal subiculum and proximal CA1 (based on rodent anatomy – see e.g. *Nilssen et al., 2019*). Whether species differences exist in the retrosplenial cortex – EC – hippocampus connectivity pattern or whether functional and structural connectivity diverge needs further investigation in the future.

## Relevance of the current findings for the functional organization of the entorhinal-hippocampal circuitry

The current findings (summarized in *Figure 5*) advance our insight into the organization of the entorhinal-hippocampal circuitry on multiple levels and contribute to cognitive and clinical research. Recent efforts to understand how the human entorhinal-hippocampal circuitry accomplishes conjunction and segregation of information largely focused on the longitudinal hippocampal axis (e.g. *Brunec et al., 2018*; *Brunec et al., 2020*; *Robin and Moscovitch, 2017*). The transversal axis of the hippocampus has been approached by studies in humans that did not directly relate connectivity findings to information processing and did not assess subfield-specific organization (*Vos de Wael et al., 2018*; *Plachti et al., 2019*; *Kharabian Masouleh et al., 2020*; *Paquola et al., 2020*; *Bouffard et al., 2022*; for an overview see *Genon et al., 2021*). *Dalton and Maguire, 2017*, however, made a relevant proposal based on visual processing pathways and information processing. In correspondence to our results, they proposed the subiculum/CA1 border as a point of convergence between scene and object information processing streams. While their conclusion was based on direct parahippocampal, retrosplenial and perirhinal connections to the hippocampus, we found that both, the $EC_{Area35-based}$ (that is connected with the cortical object processing stream) and the $EC_{RSC-based}$ (that is connected with the cortical scene processing stream) show connectivity with the subiculum/CA1 border (see also appendix V for information processing in cortical source regions). Convergence is potentially also achieved via recurrency within the entorhinal-hippocampal system and cortical regions (cf. *Koster et al., 2018* for evidence on recurrency). These considerations are an exciting future research avenue and remain speculative based on the current data due to insufficient temporal resolution. We nevertheless hypothesize the existence of two processing routes: one that processes converged object and scene information and one that processes scene information specifically. Thus, scene and object information processing might converge before the hippocampus. This presumably occurs within the anterior EC, given object-specific and scene-specific processing take place in the cortical source regions of the $EC_{Area35-based}$ and $EC_{RSC-based}$ portions, respectively (see appendix 5). Here, objects may be bound together with their defining scene-like or contextual features (akin to the 'object-in-location' idea in *Connor and Knierim, 2017*; *Knierim et al., 2014*). In addition, the dedicated scene processing that we observe along the $EC_{PHC-based}$ – distal subiculum route, may functionally underpin ideas about an anatomically graded contextual scaffold that the hippocampus utilizes to incorporate detailed information from the object-in-scene route into meaningful chunks of cohesive memory representations ('events'; *Behrens et al., 2018*; *Clewett et al., 2019*; *Robin, 2018*; *Robin and Olsen, 2019*).

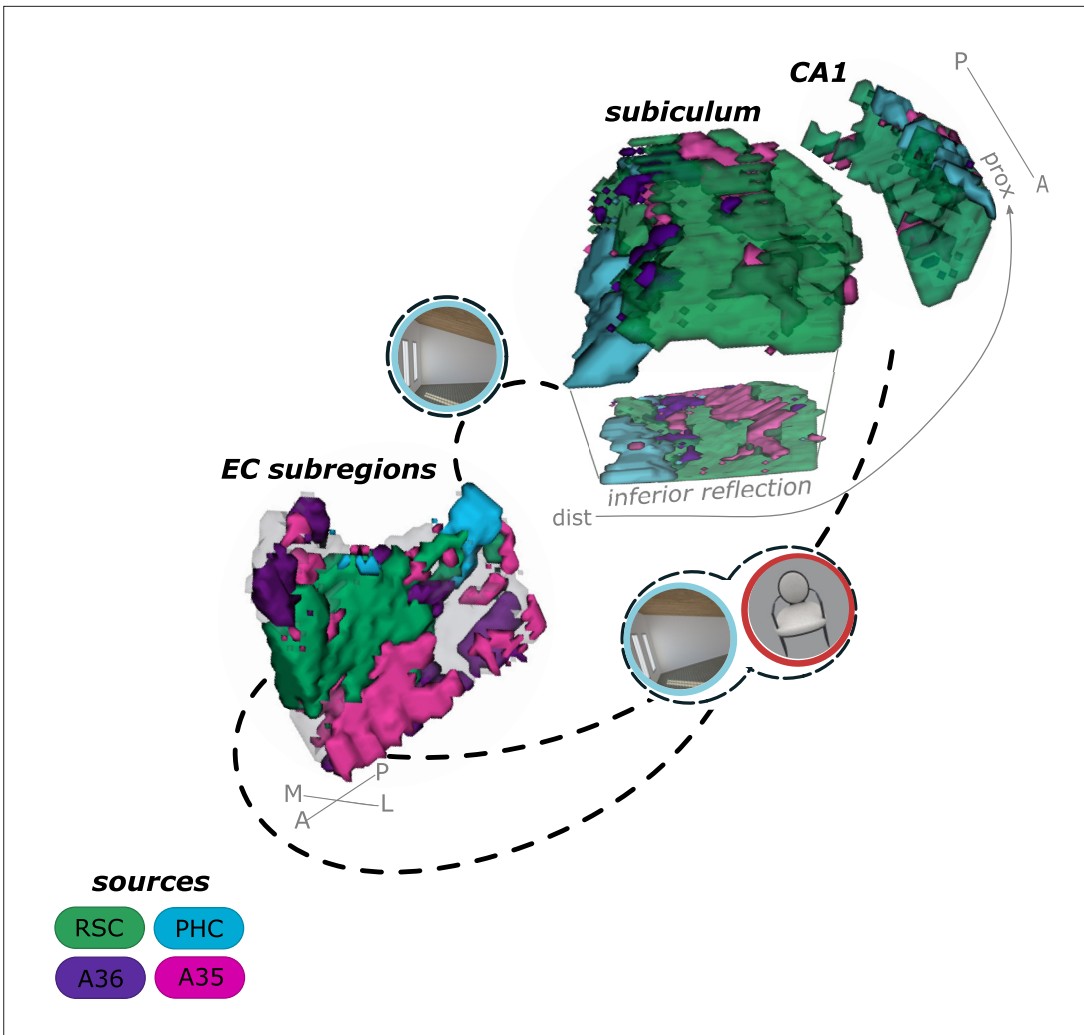

**Figure 5.** Summary of current results on the functional connectivity and information processing within the entorhinal-hippocampal circuitry. Displayed is a schematic overview of our results on the functional connectivity and information processing within the entorhinal-hippocampal circuitry with four entorhinal seed regions and a focus on the transversal axis of hippocampal subiculum and CA1. The four entorhinal seed regions are derived from preferential functional connectivity to retrosplenial (RSC, green), parahippocampal (PHC, blue) and perirhinal Area 36 (A36, purple) and Area 35 (A35, pink) sources. Routes of preferred functional connectivity are depicted with dashed lines and the preferred information processed in the connected areas is depicted with symbolizing icons (scene – blue and object – red; stimuli from the task performed by the participants). M – medial; L – lateral; A – anterior; P – posterior; prox – proximal; dist – distal.

For completeness, we noted differences in functional connectivity along the longitudinal axis of the subiculum. We observed, for instance, more widespread functional connectivity of the $EC_{Area35-based}$ in the posterior subiculum whereas functional connectivity with the $EC_{PHC-based}$ portion seems more prominent in the anterior subiculum. The latter is consistent with previous reports (*Dalton et al., 2019*). The former, however, needs to be explored further by taking different segmentation protocols and seed regions into account. Note, that *Maass et al., 2015* did not report longitudinal differences in connectivity strength between the EC and the subiculum. Future work needs to investigate how these observations relate to the reported gradient in functional connectivity and information resolution along the hippocampal longitudinal axis (e.g. *Brunec et al., 2018* but many more). Altogether, the functional organization indicates that when a memory is to be formed, some degree of information convergence happens already before the hippocampus, nevertheless keeping specific aspects of scene information separated. This conclusion is in accordance with the updated cortical mapping scheme onto the EC (*Nilssen et al., 2019*). The topographical specificity of our results supports the

necessity of functionally assessing the entorhinal-hippocampal circuitry with high spatial resolution and investigate memory function at a subregional level (*Lee et al., 2020*). The features we identified can inform future hypotheses on how the hippocampus achieves the formation of cohesive representations that serve memory function.

From a clinical research perspective, it is remarkable that the current functional connectivity pattern resembles the topology of early cortical tau pathology in Alzheimer's disease (*Lace et al., 2009*). An influential hypothesis suggests tau progression in Alzheimer's disease along functionally connected pathways in the human brain (*Franzmeier et al., 2020*; *Vogel et al., 2020*). Earliest cortical tau pathology in Alzheimer's disease accumulates in perirhinal Area 35 (also referred to as transentorhinal region) and the anterior-lateral EC before it can be found along the subiculum/CA1 border (*Braak and Braak, 1995*; *Berron et al., 2021*; *Kaufman et al., 2018*; *Lace et al., 2009*). The topology of early tau pathology in Alzheimer's disease thus mirrors the regions that we find biased towards $EC_{Area35-based}$ connectivity (*Braak and Braak, 1991*; *Lace et al., 2009*; *Roussarie et al., 2020*). Tau pathology in Alzheimer's disease is associated with memory impairment (*Bejanin et al., 2017*; *Berron et al., 2021*; *Nelson et al., 2012*) and information processing might be affected accordingly as reports have shown an association between Alzheimer's related tau pathology and object memory in early disease stages (*Berron et al., 2019*; *Maass et al., 2019*). However, given our finding of activity patterns consistent with object – scene convergence in those subregions of the hippocampal-entorhinal circuitry that are affected by early tau pathology, object-in-scene memory tasks might have increased sensitivity to memory impairment. Moreover, both, the entorhinal portion based on retrosplenial connectivity ($EC_{RSC-based}$) and the entorhinal portion based on Area 35 connectivity ($EC_{Area35-based}$), are functionally connected to the subiculum/CA1 border. This overlapping functional connectivity pattern in the hippocampus might be a way along which tau and amyloid pathologies in Alzheimer's disease could interact. This is consistent with early hypometabolism and cortical tau progression in the retrosplenial cortex and early amyloid in posterior parietal regions (*Grothe et al., 2017*; *Palmqvist et al., 2017*; *Ziontz et al., 2021*). The revealed functional connectivity and information processing profile may guide future hypotheses on the propagation of Alzheimer's pathology and related functional and cognitive impairment.

## Limitations

This study has a number of limitations. First, the biases in seed connectivity in the left hemisphere were generally weaker and proximal CA1 results were less consistent across hemispheres. We conducted all analyses independently for both hemispheres to allow internal replication of our findings, however, whether partially different effects indeed signal a lateralization of the entorhinal-hippocampal organization in humans or whether the task or another parameter influenced these observations, will require further research.

Second, while it is unlikely that our functional connectivity pattern is the result of spatial proximity, increased correlation between spatially adjacent regions is an inherent problem of functional connectivity analyses. Distances between seed and target regions differ and may determine patterns in the functional connectivity data. To diminish the influence of proximity, our smoothing kernel was smaller than two times the voxel size. It is important to stress moreover, that the pattern of our results is not easily explainable by spatial distance between seed and target regions. The $EC_{Area35-based}$ or $EC_{RSC-based}$, for instance, are not adjacent to the subiculum/CA1 border. Furthermore, we observed roughly comparable results for neighboring seeds and targets (e.g. $EC_{PHC-based}$ and distal subiculum) when we performed the functional connectivity analyses with seed and source regions in the contralateral hemisphere.

Third, our perspective was entirely functional and we cannot conclude on the directionality of our results. Also, it was beyond the scope of this study to examine direct connectivity between the cortical sources and hippocampal subregions. To what extent there is correspondence to structural connectivity (*Syversen et al., 2021*) remains to be determined, considering differences in the experimental task constraints and contrasts. Note that as a first step towards an understanding of the system's functional organization and to increase comparability with earlier studies, we assessed functional connectivity and information processing within the entorhinal-hippocampal circuitry with univariate methods. These allow relative comparisons between functional activity levels in different conditions. Consequently, we are neither able to assess what the EC is processing during

the baseline condition, meaning the absolute level of functional activity, nor are we able to verify that information processing is similar across conditions in for example the EC$_{Area35-based}$ seed. Univariate methods, moreover, average the signal over regions of interest. To capture hidden voxel-wise patterns of activity that scale with the processing of certain representations, future studies could examine information pathways with multivariate methods that evaluate informational content in the activity pattern of voxels instead of in an averaged manner (*Kragel et al., 2018*; *Kriegeskorte et al., 2008*). Moreover, recent methodological advances can be employed in the future that study functional connectivity based on the underlying content representations between regions (*Basti et al., 2020*).

Fourth, our study was originally conducted within the assumption that (functional) connectivity profiles reveal functional subregions. Based on that approach, the medial EC is identified based on i.a. retrosplenial connectivity. We, therefore conclude a surprisingly anterior yet medial EC mapping of the retrosplenial cortex. This approach has been followed by *Maass et al., 2015* and also in numerous anatomical connectivity studies in animals (see *Witter et al., 2017*). It is possible that species differences lead our EC$_{RSC-based}$ to be more anterior than one would expect based on animal studies. However, given that the medial subregion in the primate EC remains posterior (*cf.* posterior-medial EC homologue in *Maass et al., 2015*), another possibility is that our retrosplenial functional connectivity cluster maps onto the human anterior-lateral EC. Our data does not allow us to verify this latter option. It is unclear, however, why functional subregions in line with predictions from animal research can be identified for some cortical source-to-EC mappings (like the parahippocampal cortex) but not for others. In combination with closely matched histological or structural magnetic resonance imaging data, future work can further reveal the nature of retrosplenial mapping on the human EC.

In general, the quantification of the transversal connectivity pattern should be considered with some caution from the anatomist's perspective. The segmentation of subregions on functional data is an approximation because the anatomical ground truth cannot be captured by any segmentation protocol (even histological data can lead to divergent opinions). This shortcoming is amplified by group comparisons that do not account for participant-specific anatomy. Future research is needed to evaluate how the functionally derived entorhinal seeds in this study relate to histologically derived entorhinal subregions (*Oltmer et al., 2022*) or entorhinal subregions based on structural connectivity (*Syversen et al., 2021*). For a dedicated comparison of subregions, it is essential to pay close attention to the segmentation of the EC itself. Note moreover, that we excluded the head and the tail of the hippocampus from our investigation. The head is highly complex in its subfield topography (*Ding and Van Hoesen, 2015*; *Berron et al., 2017*) and prevents clear hypotheses regarding a transversal pattern. For the tail we lack an established segmentation protocol (*de Flores et al., 2020*; *DeKraker et al., 2018*). In the future, advanced segmentation methods and evaluations in the participant-space will improve this issue and reveal the organization in more detail.

## Conclusion

In sum, leveraging ultra-high field functional imaging, we provide a comprehensive in vivo exploration of the functional organization within the human entorhinal and hippocampal subregions and the circuitry`s embedding within cortical information processing streams. Within the entorhinal and hippocampal subiculum, our data partially support a continuation of cortical object and scene information processing with convergence in anterior and lateral entorhinal portions (EC$_{Area35-based}$, EC$_{RSC-based}$, EC$_{Area36-based}$), proximal subiculum and CA1, while the posterior-medial entorhinal portion (EC$_{PHC-based}$) and distal subiculum process scene information specifically. Topographically, this organization of information processing overlaps with our identified pattern of functional connectivity. The data yield spatially organized information processing along functionally connected subregions in the human EC and transversal sub/CA1 axis. Our high-resolution approach revealed unknown characteristics of functional connectivity and scene processing within the human entorhinal-hippocampal circuitry. These aid our understanding of how cortical information comes together and is further communicated within the entorhinal-hippocampal circuitry, underpinning the formation of cohesive memory representations. We provide essential insights for basic and clinical research that we believe to be crucial for the development of future hypotheses on memory function and decline.

## Methods

The current data is part of a larger study that examines exercise effects on cognition. The data that is subject to the current study have been acquired during the baseline measurement before any intervention took place. In the following, we focus on the study setup and methodological aspects of direct relevance for the current questions and data analyses.

## Participants

In total, 32 healthy participants (15 female) with a mean age of 25.5 years (range 19–35 years, standard deviation 4.3 years) were included in the current data analyses. All participants were right-handed, finished education on A-level (German Abitur or comparable) and reported absence of any neurological or psychiatric diseases. General exclusion criteria determined by the 7 Tesla MR scanning procedure were applied (e.g. metallic implants, tinnitus, known metabolic disorders). All participants gave informed consent prior to participation and received a monetary compensation. The study received approval by the ethics committee of Otto-von-Guericke University, Magdeburg (Germany) under reference number 128/14.

## Task

While functional images were acquired, participants engaged in a mnemonic discrimination task (see *Berron et al., 2018*). The object-scene task consisted of 64 objects and 64 rooms. In two runs, participants encoded always two stimuli, two 3D rendered objects in the object condition or two 3D rendered rooms in the scene condition and subsequently identified the following two same or similar stimuli as novel or old. Ten scrambled images were presented in blocks at the beginning and end of each run and served as baseline condition. All stimuli were presented for three seconds. In the recognition phase, participants had to respond during that time. Each stimulus was followed by a noise stimulus to prevent after-image and pop-out effects. The short alternating encoding/recognition sequences were embedded in an event-related design.

## Data acquisition

All MRI data was acquired with a 7 Tesla Siemens MR machine (Erlangen, Germany) using a 32-channel head coil. First structural images were obtained. A whole-brain MPRAGE volume was acquired with isotropic voxel size of 0.6 mm, TR 2500ms; TE 2.8ms, 288 slices in an interleaved manner (FOV 384x384x288). Thereafter, a partial structural T2*- weighted volume (TR 8000ms; TE 76ms, interleaved, 55 slices, FOV 512x512x55), orientated orthogonal to the main longitudinal hippocampal axis was obtained with a resolution of 0.4x0.4 mm in-plane and a slice thickness of 1 mm.

The subsequent acquisition of functional data took place in two runs à 14 min (332 volumes each) employing echo-planar imaging (EPI). The volumes were partial (40 slices, TR 2400ms, TE 22ms, FOV 216x216x40, interleaved slice acquisition), oriented along the longitudinal axis of the hippocampus and acquired with an isotropic voxel size of 1 mm.

All EPIs were distortion corrected with a point-spread function method and motion corrected during online reconstruction (*Zaitsev et al., 2004*).

## Data analyses

### Preprocessing

Preprocessing and statistical modeling of fMRI data was performed with SPM12 (Wellcome Department of Cognitive Neuroscience, University College, London UK; *Penny et al., 2011*). The individual functional images were slice time corrected and smoothed with a full-width half-maximum Gaussian kernel of 1.5 mm. To preserve a high level of anatomical specificity, smoothing was performed with a kernel smaller than two times the voxel size. The artifact detection toolbox ARTrepair (Mozes & Whitfield-Gabrieli, 2011) was subsequently used to identify outliers regarding mean image intensity and motion between scans (threshold in global intensity: 1.3%; movement threshold: 0.3 mm). Identified outliers are included as spike regressors in subsequent statistical modeling.

Task effects in the functional data were removed by fitting general linear models (with regressors for all task conditions, outliers and movement parameters) to the data. The obtained residual images were saved for the intrinsic functional connectivity analyses. Note that task-related parameter estimates were extracted for the final information processing analysis, as described later.

## Structural data processing and segmentation

### Structural template calculation (T1-weighted) and segmentation

To examine and illustrate group-level results later on, a group specific T1-weighted template was calculated using ANTS buildtemplateparallel.sh (*Avants et al., 2010*). For illustration purposes and to aid group analyses, in addition, the T1 template was manually segmented into subregions subiculum and CA1 within the hippocampal body with ITK-SNAP (*Yushkevich et al., 2006*) based on the segmentation rules described in *Berron et al., 2017*. The first slice in each hemisphere that did not contain the uncus anymore, served as start of the hippocampal body in all hippocampal subregions. The last segmented slice was the one at which both, the inferior and superior colliculi had completely disappeared, applied for each hemisphere separately. Moreover, to evaluate results across the transversal sub/CA1 axis, the subiculum masks in each hemisphere were cut in five equally wide segments from medial to lateral within each coronal image. As the CA1 region gets more and more tilted towards the hippocampal tail, the three transversal CA1 segments were determined based on manual segmentation following a geometrical rule. Therefore, the two outer borders along the transversal axis of CA1 were connected with a line. From the middle point of that line, two straight lines were drawn in a 60° angle to determine roughly equally sized transversal CA1 segments within each coronal slice and hemisphere (a figure displaying the cuts, the procedure for the CA1 segments and the numbers of voxels within each segment can be found in appendix 9). Related to the overall size of the subregions, we opted to build five subiculum and three CA1 segments along the transversal axis from proximal to distal ends.

### Segmentation of individual regions of interest

We manually segmented regions of interest (ROI) in the medial temporal lobe according to the segmentation protocol by *Berron et al., 2017*. Based on individual T2-weighted images, the parahippocampal cortex, Area 35, Area 36 and the EC are delineated (see appendix 8 for quality assurance measures). Moreover, we ran a Freesurfer 6.0 segmentation on the group T1 template to segment the isthmus cingulate cortex as retrosplenial mask (*Desikan et al., 2006*; *Fischl, 2012*). Note here that *Syversen et al., 2021* used a similar region, however excluded the most superior part. For individual retrosplenial masks, the obtained mask was co-registered from the group T1 template space to the individual T1 space by making use of the alignment matrices obtained during above described T1 group template calculation (see appendix 7 for co-registration procedure and alignment assessment). For this alignment process we used ANTS WarpImageMultiTransform.sh (*Avants et al., 2011*). The retrosplenial, parahippocampal and perirhinal Area 36 and Area 35 regions served as cortical source regions for an initial functional connectivity analysis that we conducted to obtain functional subregions within the entorhinal cortex (see upcoming paragraphs and appendix 2).

### Co-registration of individual structural data to functional data space

For later functional data extraction, the individual T1-weighted and T2-weighted structural images were co-registered and resliced to the echo-planar images. Therefore, ANTS was used to transfer the T2-weighted structural image to the participant's T1 space (*Avants et al., 2011*). For the co-registration between individual T1-weighted and EPIs, FSL epi_reg was applied (*Jenkinson and Smith, 2001*). All subsequently segmented individual masks were co-registered to the participant's functional (echo-planar) images using the obtained warping matrices (see appendix 7 for co-registration procedure and alignment assessment). ANTS WarpImageMultiTransform.sh was applied for T2 to T1 co-registration and FSL flirt was used for T1 to echo-planar image co-registration (*Avants et al., 2011*; *Jenkinson and Smith, 2001*).

### ROI preparation for seed regions in functional connectivity analyses

All masks that served as source and seed regions throughout the functional connectivity analyses (retrosplenial, parahippocampal, perirhinal Area 36 and Area 35 and the later defined entorhinal subregions) were thresholded according to mean intensity to prevent signal dropout and thus a distortion of the average functional signal extracted from seed regions for the connectivity analysis. Therefore, we followed *Libby et al., 2012* and *Maass et al., 2015*, to remove all voxels from each ROI that showed a mean intensity over time of less than two standard deviations from the mean intensity across all voxels. The thresholding was performed before each seed-to-voxel functional connectivity analysis.

## Functional connectivity analyses at the participant level

Two different functional connectivity analyses were performed that build upon the approach by *Maass et al., 2015*. The first analysis served to identify functional subregions ('seeds') within the entorhinal cortex that uniquely connect with functionally and clinically relevant cortical sources. The second, core analysis, then evaluated the intrinsic functional connectivity pattern between these entorhinal seeds and hippocampal subiculum and CA1. Both functional connectivity analyses were performed on residuals of task-related functional data, creating a dataset that resembles resting-state data (*Gavrilescu et al., 2008*; *Maass et al., 2015*). In the following we describe the analysis procedure in detail. Note, that all analyses were conducted independently in both hemispheres.

To determine functional entorhinal seed regions we first performed a seed-to-voxel semipartial correlation analysis (*Whitfield-Gabrieli and Nieto-Castanon, 2012*) between the individually extracted residuals from retrosplenial, parahippocampal and perirhinal Area 36 and Area 35 sources as well as entorhinal voxels. The regions we call cortical sources served as seeds in that analysis. Note that the semipartial correlations calculate the variance in a voxel that is uniquely explained by the source, excluding contributions from other sources. Please refer to appendix 6 for more details on this functional semipartial correlation analysis. To obtain entorhinal seeds for the core functional connectivity analysis, first, we calculated one-sample T-tests across participants on the individually obtained and aligned, standardized beta maps for each source, respectively. Second, the four resulting statistical maps (one for each source) have been thresholded at T>3.1. Each entorhinal voxel now was attributed to be preferentially connected with one of the four source regions, based on the voxel's maximum T value across the thresholded one-sample T-test maps. Those voxels that did not reach the threshold of T>3.1 in any of the four statistical maps have not been attributed to be preferentially connected with any of the four cortical sources. Finally, across hemispheres we selected for each source preference an equal number of these highest preference voxels across all T-tests (the number is determined by the hemisphere with the lowest relevant number of voxels). This procedure yielded four entorhinal subregions, one containing the entorhinal voxels that preferentially functionally connect with the retrosplenial (1530 voxels), one containing the entorhinal voxels that preferentially functionally connect with the parahippocampal cortex (145 voxels) and one each that contained the preferentially functionally connected voxels with perirhinal Area 35 (298 voxels) and Area 35 (751 voxels), respectively. All four entorhinal seed masks were determined on group level and co-registered to each participant. They then served as seed regions for the core functional connectivity analysis between entorhinal cortex seeds and hippocampal subregions.

For the core functional connectivity analysis (entorhinal seeds-to-hippocampal subregion voxels), an analogous seed-to-voxel semipartial correlation analysis was performed on the individual residual functional imaging data using the CONN toolbox (*Whitfield-Gabrieli and Nieto-Castanon, 2012*). Note again that the semipartial correlations calculate the variance in a voxel that is uniquely explained by the seed, excluding contributions from other seeds. Now the four entorhinal subregions served as seeds and functional connectivity was examined with the whole brain (later masked by the hippocampal subregion masks). For each functional connectivity analysis, mean time series were extracted from the respective seed region and entered as regressor of interest. White matter and CSF time series, realignment parameters and outliers served as regressor of no interest. The functional data from the residuals was band-pass filtered (0.01–0.1 Hz) and semipartial correlations were obtained between the seed timeseries and all other brain voxel's timeseries. The obtained beta maps contained Fisher-transformed correlation coefficients and were used for subsequent group analyses.

## Alignment between participants

To be able to perform group statistics on the resulting topography in the beta maps, the individual data was aligned to group space. Here, the T1 template image served as reference space. Using the inverse of the previously obtained individual warping matrices from individual T1 to EPI, first the standardized beta maps were co-registered from EPI to individual T1 space. In a further step, the statistical maps were then aligned between the individual T1 space and the group T1 template space, by making use of the alignment matrices obtained during above described T1 group template calculation. For this alignment process we used ANTS WarpImageMultiTransform.sh (*Avants et al., 2011*).

## Functional connectivity analysis at group level

To investigate the functional connectivity profile between the four entorhinal seeds and the subiculum and CA1 subregion across individuals, we evaluated connectivity preferences to either seed within all transversal segments of the subiculum and CA1 target regions. Therefore, mean values for connectivity estimates to either entorhinal cortex seed were extracted from the group aligned but participant-specific beta maps out of each transversal segment, averaged along all coronal slices. Note, that segment-based extraction is necessary due to the varying number of sagittal slices that cover the respective regions along the longitudinal axis of the hippocampal body. Based on these participant-level connectivity results, connectivity preference plots for all four entorhinal seeds have been created to depict tendencies along the transversal sub/CA1 axis.

A hierarchical repeated-measures ANOVA testing procedure was employed to reveal significant differences in the transversal hippocampal connectivity patterns between entorhinal seed regions. Therefore, in a first step, an overall repeated measures ANOVA (4 seed X transversal segments) was performed per target region (subiculum and CA1 in both hemispheres) to reveal whether significant differences in seed connectivity estimates exist across the transversal axis of the respective target region (subiculum or CA1). If the overall seed X transversal segment interaction effect was significant (false-discovery-rate corrected according to *Benjamini and Hochberg, 1995*), in a second step, one-way repeated measures ANOVAs have been performed for each seed to identify those entorhinal seeds that indeed show a differential connectivity pattern across the transversal axis of the target region (all false-discovery-rate corrected according to *Benjamini and Hochberg, 1995*). If more than one seed main effect was significant, finally we determined whether these seeds exhibit an opposing connectivity pattern across the transversal axis of subiculum or CA1, respectively, by evaluating the pair-wise seed X transversal segment interaction effects on the extracted connectivity estimates.

For a more detailed topographical display of the entorhinal-hippocampal connectivity results, we calculated one-sample t-tests on the aligned, standardized beta maps that we obtained in the first-level analyses for each seed respectively. Crucially, the resulting group-level one-sample T-test statistical maps were only used to display results but not for any further statistical inference. To depict the topography of the respective voxel-wise seed preferences, the resulting group-level T-maps were thresholded with T > |0.001| and masked with the respective subregion of interest. To depict general tendencies in the connectivity profile, for each voxel in the region of interest the preferred seed connectivity was determined by attributing it to the seed with the highest T value across the one-sample T-test maps. The resulting maps were depicted in 3D plots, generated with ITK-SNAP (*Yushkevich et al., 2006*) that provide an overview of each voxel's preference for the respective seed functional connectivity at a glance.

## Functional analysis of content-related activity at participant level

To investigate whether scene and object information is differentially processed within entorhinal seed regions and along the transversal sub/CA1 axis, the results from the initially fitted general linear models (used to remove task effects) were examined. Contrast estimates were calculated between the beta estimates obtained from task conditions in which individuals saw indoor rooms (scene) versus objects on the screen and conditions in which individuals saw the scrambled stimuli (baseline). The resulting contrast value maps for object > baseline and scene > baseline were then co-registered to the T1 group template space. Subsequently, individual mean contrast estimates have been extracted from the four entorhinal seed regions and from those transversal segments that had previously been used for the evaluation of the intrinsic functional connectivity results (three or five segments in CA1 and subiculum, respectively).

With repeated measures ANOVAs (content condition X entorhinal region or content condition X transversal hippocampal segment), we investigated whether contrast estimates differed under scene and object conditions in the respective regions. Effect of interest thus, was the interaction between the content condition and the subregion or segment, respectively. Post-hoc paired-samples T-test were performed if the respective interaction effect was significant, to reveal in which subregion or

segment functional activity between scene and object processing conditions differed significantly from each other (all false-discovery-rate corrected according to *Benjamini and Hochberg, 1995*).

## Acknowledgements

We thank the Leibniz Institute for Neurobiology in Magdeburg for providing access to the 7 Tesla MRI Scanner. We are grateful for the support of Anne Hochkeppler and Regina Schwarzer with manual segmentations of the medial temporal lobe subregions and for insightful discussions regarding functional connectivity with Yi Chen. This work was supported by the European Union's Horizon 2020 Research and Innovation Programme under grant agreements 785907 (HBP SGA2) and 945539 (HBP SGA3) as well as the Deutsche Forschungsgemeinschaft (DFG, German Research Foundation) – Project-ID 42589994. DB has received funding from the European Union's Horizon 2020 research and innovation programme under the Marie Skłodowska-Curie grant agreement No. 843074.

## Additional information

### Funding

| Funder | Grant reference number | Author |
|---|---|---|
| Deutsche Forschungsgemeinschaft | Project-ID 42589994 | Magdalena M Sauvage Emrah Düzel |
| HORIZON EUROPE Marie Sklodowska-Curie Actions | 843074 | David Berron |
| European Union's Horizon 2020 Framework Programme for Research and Innovation | 785907 (HBP SGA2) and 945539 (HBP SGA3) | Emrah Düzel |

The funders had no role in study design, data collection and interpretation, or the decision to submit the work for publication.

### Author contributions

Xenia Grande, Conceptualization, Formal analysis, Visualization, Writing - original draft, Writing – review and editing, Data Interpretation; Magdalena M Sauvage, Funding acquisition, Writing – review and editing, Data interpretation; Andreas Becke, Investigation, Project administration; Emrah Düzel, Funding acquisition, Writing – review and editing; David Berron, Conceptualization, Supervision, Funding acquisition, Investigation, Project administration, Writing – review and editing, Data Interpretation

### Author ORCIDs

Xenia Grande ![ORCID] http://orcid.org/0000-0002-2486-3201
Magdalena M Sauvage ![ORCID] http://orcid.org/0000-0002-7586-6410
David Berron ![ORCID] http://orcid.org/0000-0003-1558-1883

### Ethics

Human subjects: Informed consent and consent to publish was obtained from human participants. The study received approval by the ethics committee of Otto-von-Guericke University, Magdeburg (Germany) under reference number 128/14.

### Decision letter and Author response

Decision letter https://doi.org/10.7554/eLife.76479.sa1
Author response https://doi.org/10.7554/eLife.76479.sa2

## Additional files

### Supplementary files

- Transparent reporting form

• Source code 1. Group-level statistical map (T-statistics, one-sample T-test) for left Area 35 to left entorhinal voxels functional connectivity.

• Source code 2. Group-level statistical map (T-statistics, one-sample T-test) for left Area 36 to left entorhinal voxels functional connectivity.

• Source code 3. Group-level statistical map (T-statistics, one-sample T-test) for left retrosplenial to left entorhinal voxels functional connectivity.

• Source code 4. Group-level statistical map (T-statistics, one-sample T-test) for left parahippocampal to entorhinal voxels functional connectivity.

• Source code 5. Group-level statistical map (T-statistics, one-sample T-test) for right Area 35 to entorhinal voxels functional connectivity.

• Source code 6. Group-level statistical map (T-statistics, one-sample T-test) for right Area 36 to entorhinal voxels functional connectivity.

• Source code 7. Group-level statistical map (T-statistics, one-sample T-test) for right retrosplenial to entorhinal voxels functional connectivity.

• Source code 8. Group-level statistical map (T-statistics, one-sample T-test) for right parahippocampal to entorhinal voxels functional connectivity.

• Source code 9. Group-level statistical map (T-statistics, one-sample T-test) for left Area35-based entorhinal seed to hippocampal Subiculum/CA1 voxels functional connectivity.

• Source code 10. Group-level statistical map (T-statistics, one-sample T-test) for left Area36-based entorhinal seed to hippocampal Subiculum/CA1 voxels functional connectivity.

• Source code 11. Group-level statistical map (T-statistics, one-sample T-test) for left RSC-based entorhinal seed to hippocampal Subiculum/CA1 voxels functional connectivity.

• Source code 12. Group-level statistical map (T-statistics, one-sample T-test) for left PHC-based entorhinal seed to hippocampal Subiculum/CA1 voxels functional connectivity.

• Source code 13. Group-level statistical map (T-statistics, one-sample T-test) for right Area35-based entorhinal seed to hippocampal Subiculum/CA1 voxels functional connectivity.

• Source code 14. Group-level statistical map (T-statistics, one-sample T-test) for right Area36-based entorhinal seed to hippocampal Subiculum/CA1 voxels functional connectivity.

• Source code 15. Group-level statistical map (T-statistics, one-sample T-test) for right RSC-based entorhinal seed to hippocampal Subiculum/CA1 voxels functional connectivity.

• Source code 16. Group-level statistical map (T-statistics, one-sample T-test) for right PHC-based entorhinal seed to hippocampal Subiculum/CA1 voxels functional connectivity.

### Data availability

Source data that contain numerical data used to generate Figure 2, Figure 3, Figure 4, Appendix 1 Figure 2, Appendix 1 Figure 3, Appendix 1 Figure 4, Appendix 5 Figure 1 as well as group-level statistical maps (referred to as Source Code 1-16) that underlie Figure 1, Figure 2, Appendix 1 Figure 1, Appendix 1 Figure 2, Appendix 3 Figure 1, Appendix 3 Figure 2 and Appendix 4 Figure 1 have been provided under: Open Science Framework. ID 9v3qp. https://osf.io/9v3qp.

The following dataset was generated:

| Author(s) | Year | Dataset title | Dataset URL | Database and Identifier |
|---|---|---|---|---|
| Grande X, Berron D | 2022 | Source Data from Functional Connectivity and Information Processing in the Entorhinal-Hippocampal Circuitry | https://osf.io/9v3qp | Open Science Framework, 9v3qp |

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

## Appendix 1

### Left hemisphere results

### Four cortical sources divide the left entorhinal cortex (EC) in retrosplenial-, parahippocampal-, Area 35- and Area 36-based seeds

Based on functional connectivity preferences to the sources parahippocampal cortex (*Source code 4*), retrosplenial cortex (*Source code 3*), Area 36 (*Source code 2*) and Area 35 (*Source code 1*), we obtained four entorhinal seeds. The majority of voxels can roughly be described as clustering in the posterior-medial entorhinal portion for the $EC_{PHC-based}$, the anterior-medial (and posterior-medial) portion for the $EC_{RSC-based}$ seed, the anterior-lateral portion for the $EC_{Area35-based}$ and the posterior-lateral portion for the $EC_{Area36-based}$ seed (see appendix 2 for exact voxel counts). Note that both perirhinal-based entorhinal seeds extend along the anterior to posterior axis such that the $EC_{Area35-based}$ progresses more along deep entorhinal portions (with a main focus anteriorly) and the $EC_{Area36-based}$ along superficial entorhinal portions (with a main focus posteriorly, see *Appendix 1—figure 1* and the medial reflection of the EC seeds).

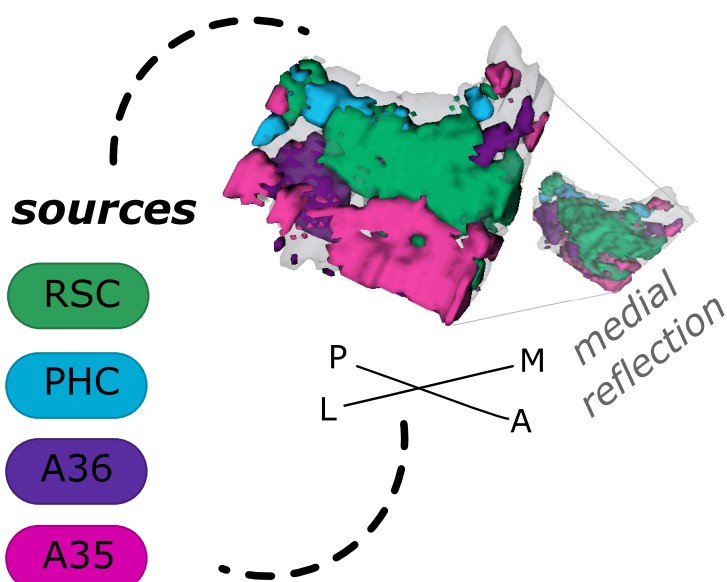

**Appendix 1—figure 1.** Left entorhinal seed regions based on connectivity preferences to cortical regions. Displayed is the left entorhinal cortex as a 3D image with colored seed regions. The seed regions have been identified based on a source-to-voxel functional connectivity analysis and resulting connectivity preference to either the left retrosplenial cortex (RSC, green), parahippocampal cortex (PHC, blue), Area 36 (A36, purple) or Area 35 (A35, pink) sources. Note that preferences to Area 36 are best visible from a medial perspective on the entorhinal cortex as depicted in the medial reflection. Seed regions have been determined based on the maximum voxels across four one-sample T-tests at group level, one per source, sample size n = 32. M – medial; L – lateral; A – anterior; P – posterior.

### Left distal subiculum is functionally connected with the $EC_{PHC-based}$ seed while the subiculum/CA1 border is connected with $EC_{RSC-based}$ and $EC_{Area35-based}$ seeds

When extracting estimates of connectivity preferences across individuals from proximal and distal hippocampal subfield segments for either entorhinal seed, repeated measures ANOVAs revealed significant seed X segments interaction effects along the transversal axis of the left subiculum and CA1 (subiculum: $F(12,372) = 4.609$; $p<0.001$; CA1: $F(6,186) = 2.458$; $p=0.047$; see *Appendix 1—figure 2*).

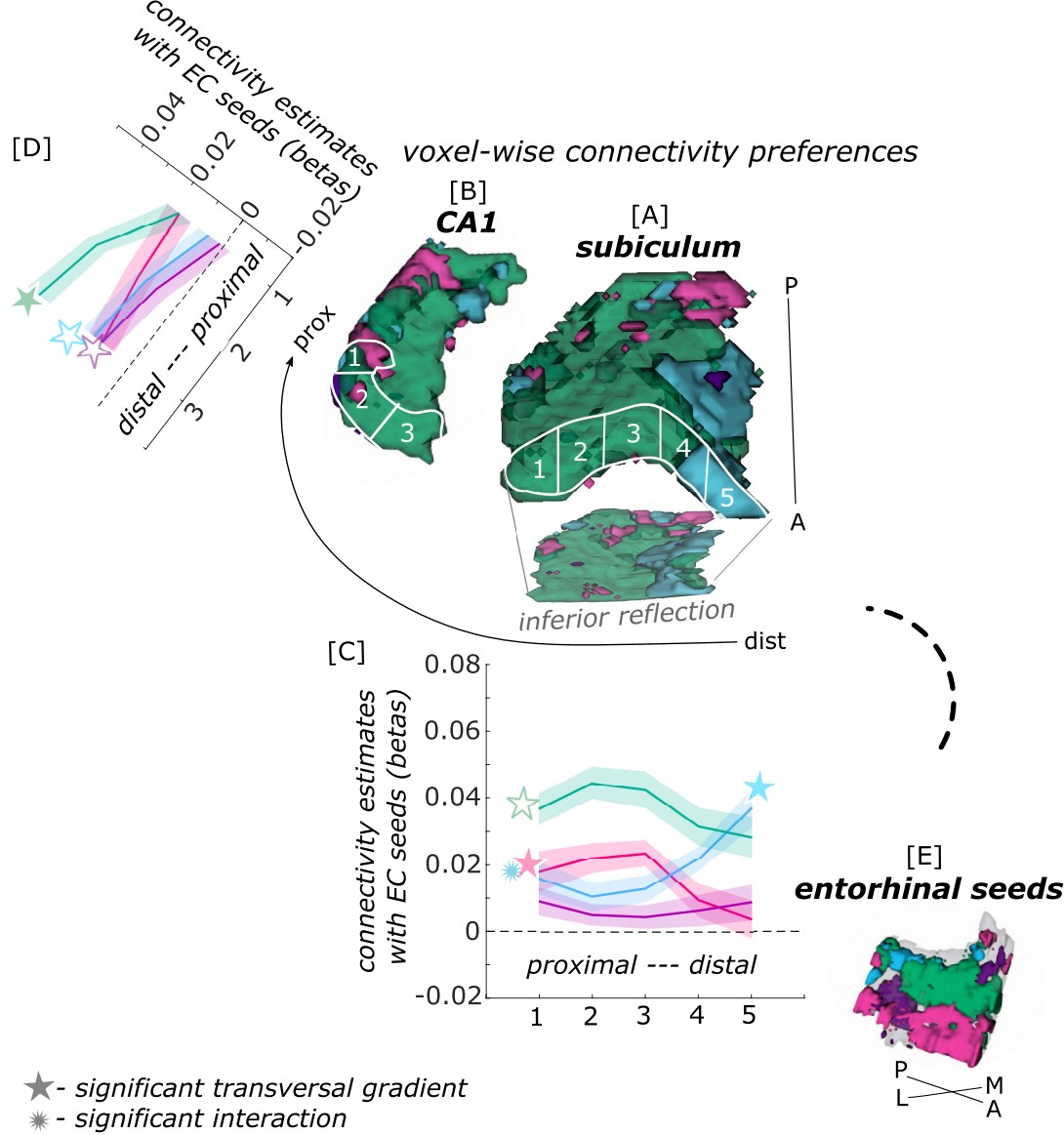

★ - *significant transversal gradient*
✳ - *significant interaction*

**Appendix 1—figure 2.** Functional connectivity preferences to entorhinal seeds along the subiculum and CA1 transversal axis, left hemisphere. Displayed are the results of a seed-to-voxel functional connectivity analysis between the displayed left entorhinal seeds and the left subiculum and CA1 subregion. The 3D figure shows voxel-wise connectivity preferences to the entorhinal seeds (color coded to refer to the respective entorhinal seed [**E**]) on group level ([**A**] - subiculum; [**B**] - CA1; maps for connectivity preferences: *Source code 9* - EC$_{\text{Area35-based}}$, pink; *Source code 10* - EC$_{\text{Area36-based}}$, purple; *Source code 12* - EC$_{\text{PHC-based}}$, blue; *Source code 11* - EC$_{\text{RSC-based}}$ seed, green). Note that preferences to the EC$_{\text{Area35-based}}$ seed (pink) are located mainly in the inferior subiculum and CA1 and are therefore visible in the inferior reflection. To display mean connectivity preferences across participants along the transversal axis, beta estimates were extracted and averaged from equally sized segments from proximal to distal ends (five segments in subiculum [**A**], three segments in CA1 [**B**]; schematized in white on the 3D figures) on each coronal slice and averaged along the longitudinal axis. Repeated measures ANOVAs revealed significant differences in connectivity estimates along the transversal axis in CA1 [**D**] and subiculum [**C**] with interaction effects in the subiculum. Displayed significances obtained by FDR-corrected post-hoc tests and refer to $p<0.05$. Empty asterisks refer to effects that did not reach significance under FDR-correction. Shaded areas in the graphs refer to standard errors of the mean, sample size n = 32. EC – entorhinal; M – medial; L – lateral; A – anterior; P – posterior; prox – proximal; dist – distal. *Appendix 1—figure 2—source data 1* contains individual connectivity estimates per subregion (Sub – subiculum and CA1, respectively) and seed (EC$_{\text{RSC-based}}$ – RSCECseed, EC$_{\text{Area35-based}}$ – A35ECseed, EC$_{\text{PHC-based}}$ – PHCECseed, EC$_{\text{Area36-based}}$ – A36ECseed) for each transversal segment (1–5 or 1–3, respectively from proximal to distal).

The online version of this article includes the following source data for appendix 1—figure 2:

**Appendix 1—figure 2—source data 1.** Individual functional connectivity estimates to left entorhinal seeds, extracted from left subiculum and CA1 transversal segments.

In the left subiculum, additional repeated measures ANOVAs showed that the $EC_{Area35-based}$ (F(4,124) = 4.489; $p_{FDR}$ = 0.025), and $EC_{PHC-based}$ (F(4,124) = 8.701; $p_{FDR}$ <0.001) seeds displayed a significant main effect across the transversal subiculum axis. Here, the transversal preference to the $EC_{RSC-based}$ entorhinal seed does not survive FDR correction (F(4,124) = 4.489; Huynh-Field uncorrected p=.05), shows however the same tendency as in the right hemisphere. The differential functional connectivity preferences for the $EC_{Area35-based}$ and $EC_{PHC-based}$ seed interacted significantly across the transversal axis, as shown in a subsequent repeated measures ANOVA (F(4,124) = 10.795; $p_{FDR}$ <0.001).

In the left CA1, additional repeated measures ANOVAs showed that the connectivity preference towards the $EC_{RSC-based}$ seed displayed a significant main effect across the transversal CA1 axis (F(2,62) = 6.753; p=0.024). In the distal CA1, the preferential functional connectivity with the $EC_{PHC-based}$ seed was higher than in the proximal portion of CA1. In the left CA1 a similar but weaker transversal pattern was observed for connectivity preferences with the $EC_{Area36-based}$ (F(2,62) = 3.841; $p_{FDR}$ = 0.051) and $EC_{PHC-based}$ seed regions (F(2,62) = 3.468; $p_{FDR}$ = 0.051).

## Left distal subiculum and $EC_{PHC-based}$ exhibit higher functional activity in the scene condition while other subregions show no significant difference between conditions

For the characteristics of information processing, we first focus on the left entorhinal seed regions. When extracting task-related parameter estimates for object and scene information conditions, a repeated measures ANOVA showed a significant interaction between region and information type (object versus scene; F(3,93) = 9.772; p<0.001). Post-hoc t-tests revealed that only in the $EC_{PHC-based}$ seed region functional activity for scene information was significantly higher than for object information ($p_{FDR}$ = 0.003), while in the remaining three left entorhinal seed regions no significant difference between object and scene conditions existed (all $p_{FDR}$ = 0.5776; see *Appendix 1—figure 3*).

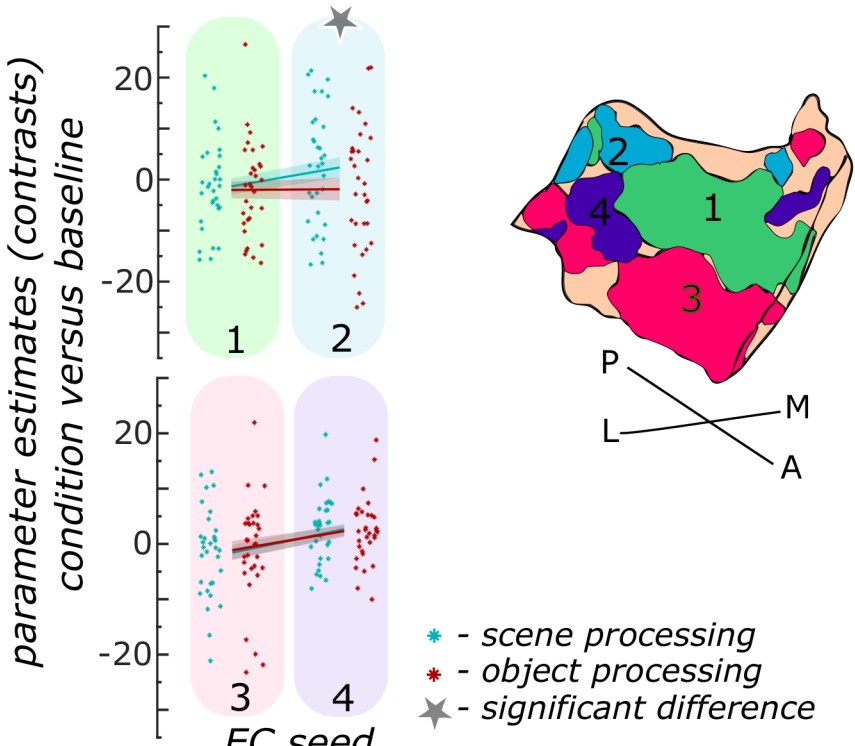

**Appendix 1—figure 3.** Functional activity during scene and object conditions in entorhinal seed regions, left hemisphere. Displayed are the extracted parameter estimates for the object versus baseline contrast ('object

*Appendix 1—figure 3 continued*

information processing', red) and the scene versus baseline contrast ('scene information processing', cyan) from each left entorhinal seed region per individual (dots) and summarized across individuals (lines). A schematic depiction of the respective entorhinal seed regions is displayed by a 3D drawing of the left EC. A repeated measures ANOVA revealed a significant interaction between condition and seed region. The displayed significant difference is obtained with FDR-corrected post-hoc tests and refers to p<0.05. During the object condition, participants were presented with 3D rendered objects on screen, during the scene condition with 3D rendered rooms and during the baseline condition they saw scrambled pictures. The shaded area around the lines refers to standard errors of the mean, sample size n = 32. EC – entorhinal; M – medial; L – lateral; A – anterior; P – posterior. *Appendix 1—figure 3—source data 1* contains extracted parameter values per individual and EC seed (isthmuscingulate – ECRSC-based, Area 35 – ECArea35-based, Area 36 – ECArea36-based, PHC – ECPHC-based seed) for the object versus baseline and scene versus baseline contrasts.

The online version of this article includes the following source data for appendix 1—figure 3:

**Appendix 1—figure 3—source data 1.** Individual parameter estimates for scene and object processing in left entorhinal seed regions.

In the left hippocampal subregions, extracting the task-related parameter estimates for object and scene conditions from proximal and distal segments within each participant showed a significant interaction between transversal segments and information type in the subiculum ($F(4,124) = 7.697$; p<0.001), not however in CA1 ($F(2,62) = 1.1925$; p = 0.3042) as revealed by repeated measures ANOVAs. Post-hoc T-tests showed that only in the distal subiculum segments and in the middle segment significantly more scene than object information was processed (from most distal to middle segment $p_{FDR}$ <0.001; $p_{FDR}$ = 0.0015; $p_{FDR}$ = 0.0274). In all other segments along the transversal axis, no significant difference in functional activity related to object and scene conditions existed (from medial to proximal: $p_{FDR}$ = 0.1009; $p_{FDR}$ = 0.2435; see *Appendix 1—figure 4*).

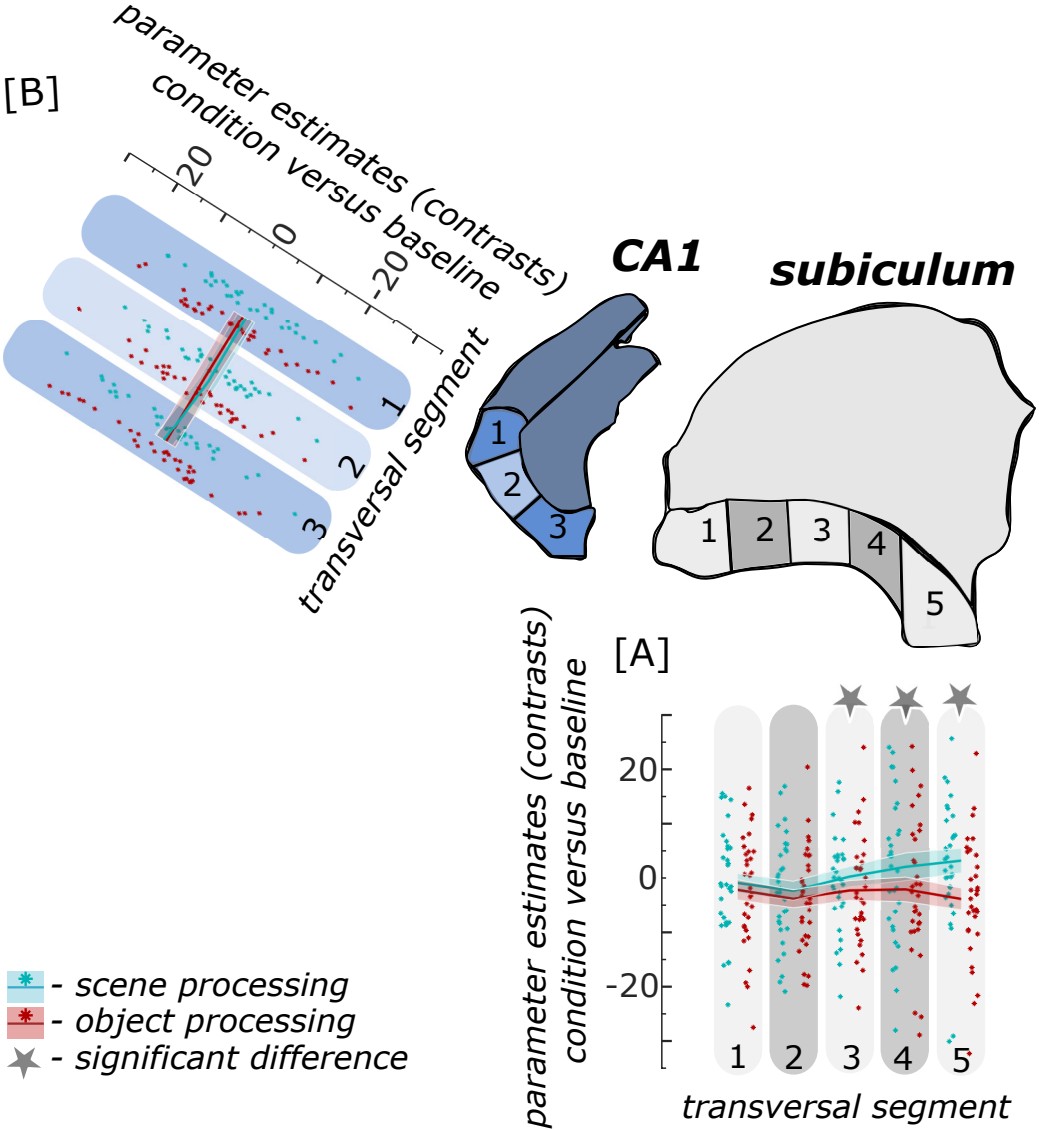

**Appendix 1—figure 4.** Functional activity during scene and object conditions along the transversal axis of subiculum and CA1, left hemisphere. Displayed are the extracted parameter estimates for the object versus baseline contrast (red) and the scene versus baseline contrast (cyan) from the respective transversal segments in the subiculum ([A] grey) and CA1 ([B] blue) per individual (dots) and summarized across individuals (lines). A schematic depiction of the respective transversal segment is displayed by a 3D drawing of the left subiculum and CA1 subregions. Repeated measures ANOVAs revealed a significant interaction between condition and transversal segment in the subiculum only. The displayed significant difference was obtained with FDR-corrected post-hoc tests and refers to p<0.05. During the object condition, participants were presented with 3D rendered objects on screen, during the scene condition with 3D rendered rooms and during the baseline condition they saw scrambled pictures. The shaded area around the lines refers to standard errors of the mean, sample size n = 32. *Appendix 1—figure 4—source data 1* contains extracted parameter values for each subregion (Sub – subiculum and CA1, respectively) per individual and transversal segment (1–5 and 1–3, respectively from proximal to distal).

The online version of this article includes the following source data for appendix 1—figure 4:

**Appendix 1—figure 4—source data 1.** Individual parameter estimates for scene and object processing in left transversal subiculum and CA1 segments.

## Appendix 2

### Quantitative assessment of entorhinal seeds

To assess the main location of each cortical source preferences within the EC, we cut the left and right EC in four quadrants. This was performed in T1 template space. First, the middle slice of all coronal slices which capture the EC was determined separately for each hemisphere. This slice was used to cut the EC in quadrants I, III and II, IV. Second, the middle slice of all axial slices which capture the EC was determined. This slice served to cut the EC in quadrants I, II and III, IV (see *Appendix 2—figure 1*). Note, to determine the most superior axial slice, the most posterior coronal level of the EC was used. Subsequently, we counted the number of voxels that have been assigned to each of the four cortical source regions after the initial functional connectivity analyses (that served to determined EC seeds). Averaged across hemispheres, most voxels assigned to the retrosplenial source are in EC quadrant I, most voxels assigned to the Area 35 source in EC quadrant II, most voxels assigned to the parahippocampal cortex in EC quadrant III and most voxels assigned to Area 36 in EC quadrant IV (see *Appendix 2—table 1* for detailed voxel counts). Note that these quadrants do not refer to anatomically defined EC subregions.

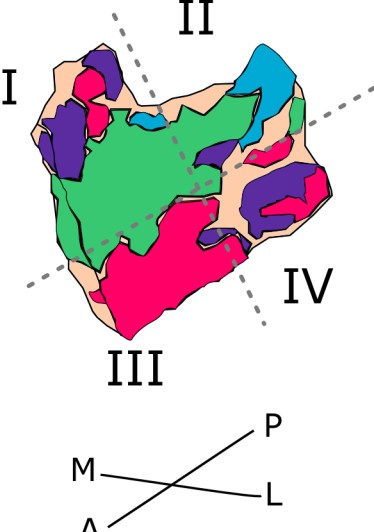

**Appendix 2—figure 1.** Entorhinal cortex cut in four quadrants. *Illustrated is the schematic entorhinal cutting in four quadrants (I, II, III and IV) in the right hemisphere. Stippled lines illustrate approximate cuts. M – medial, L – lateral, A – anterior, P – posterior.*

**Appendix 2—table 1.** Number of voxels attributed to have a preferred functional connectivity to either cortical source (RSC, PHC, A35, A36) within each EC quadrant (I.-IV.).

Bold voxel numbers refer to the highest number across EC quadrants. EC – entorhinal cortex, RSC – retrosplenial cortex, PHC – parahippocampal cortex, A35 – perirhinal Area 35, A36 – perirhinal Area 36.

| EC quadrant | I. | II. | III. | IV. |
|---|---|---|---|---|
| RSC-source | **599** | 421 | 337 | 173 |
| PHC-source | 13 | **132** | 0 | 0 |
| A35-source | 71 | 80 | **433** | 167 |
| A36-source | 103 | 51 | 39 | **201** |

## Appendix 3

### Functional connectivity gradients by source and seed region

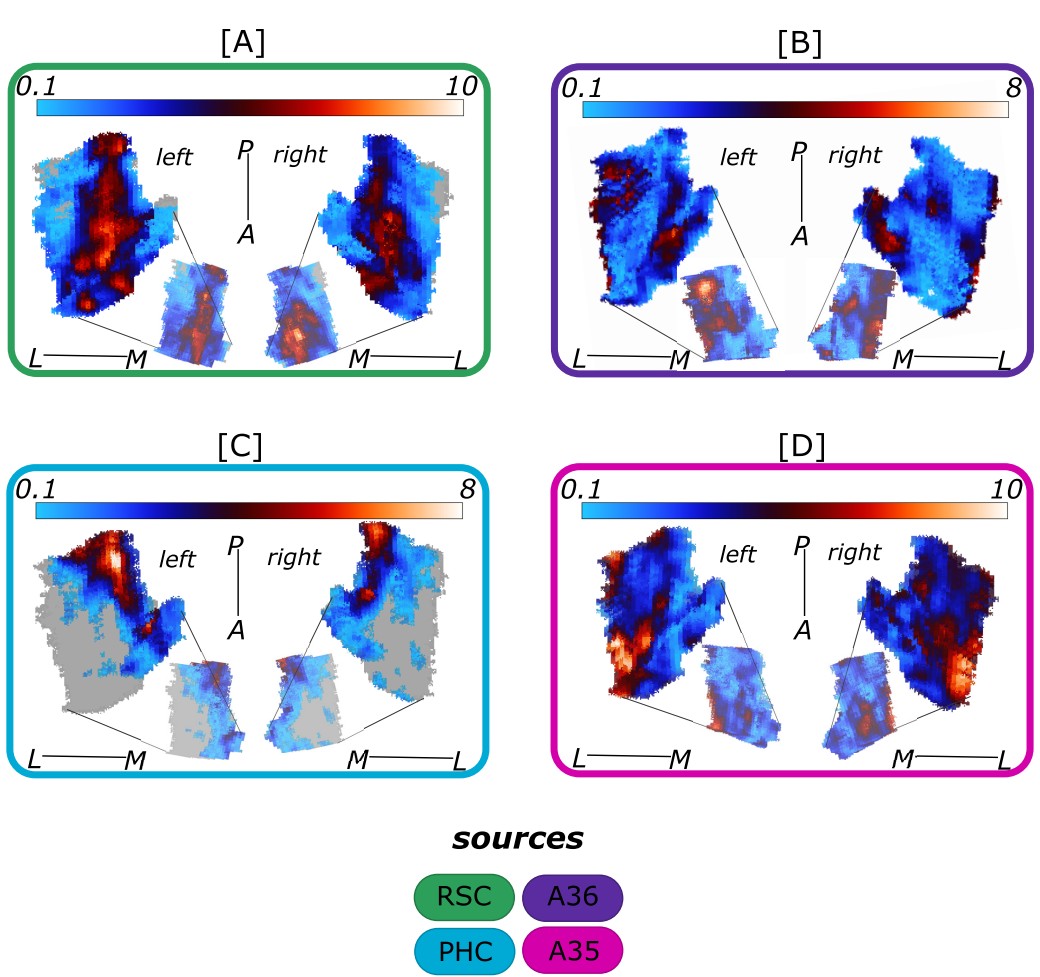

**Appendix 3—figure 1.** Entorhinal functional connectivity with isolated cortical sources. Displayed are the voxel-wise functional connectivity values (T values) of the EC with the respective cortical sources [A] retrosplenial cortex (RSC, green, left: *Source code 3*, right: *Source code 7*), [B] perirhinal Area 36 (A36, purple, left: *Source code 2*, right: *Source code 6*), [C] parahippocampal cortex (PHC, blue, left: *Source code 4*, right: *Source code 8*) and [D] perirhinal Area 35 (A35, pink, left: *Source code 1*, right: *Source code 5*). Results from left and right hemisphere one-sample T-tests for the functional connectivity with the respective source are displayed alongside each other for each cortical source, sample size n = 32. The smaller entorhinal cortex maps in the middle of each rectangle are medial reflections of the respective results. Colorbars reflect the range of T values. Grey areas refer to T values of T<0.1. L – lateral; M – medial; A – anterior; P – posterior.

# Subiculum/CA1 voxel-wise connectivity
## (T values)

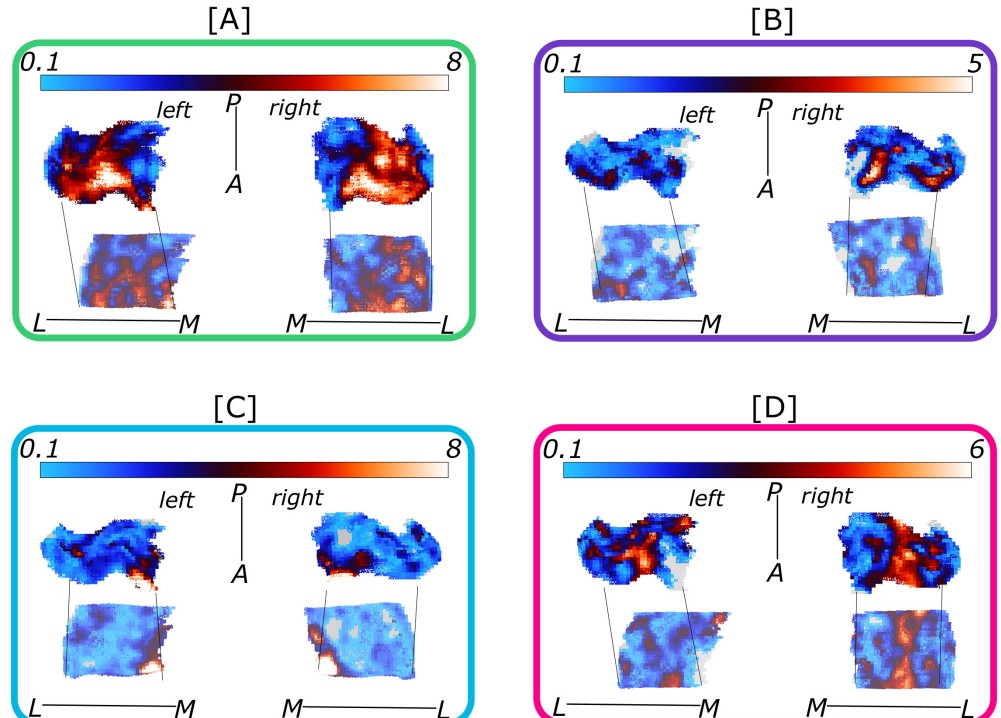

**EC seeds**

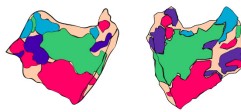

**Appendix 3—figure 2.** Subiculum/CA1 functional connectivity with isolated entorhinal (EC) seeds. Displayed are the voxel-wise functional connectivity values (T values) of the subiculum and CA1 to the respective [A] green (EC$_{RSC-based}$, left: *Source code 11*, right: *Source code 15*) [B] purple (EC$_{Area36-based}$, left: *Source code 10*, right: *Source code 14*), [C] blue (EC$_{PHC-based}$, left: *Source code 12*, right: Source*Source code 16*) and [D] pink (EC$_{Area35-based}$, left: *Source code 9*, right: *Source code 13*) EC seeds. The respective seeds are illustrated in the lower panel. Results from left and right hemisphere one-sample T-test for the functional connectivity with the respective seed are displayed alongside each other, sample size n = 32. The lower subiculum/CA1 maps within each rectangle are inferior reflections of the respective results. Colorbars reflect the range of T values. Grey areas refer to T values of T<0.1. L – lateral; M – medial; A – anterior; P – posterior.

## Appendix 4

### Superior and inferior view on voxel-wise functional connectivity preferences to entorhinal seeds

**voxel-wise connectivity preferences**

*left*                                                                                     *right*

### [A] *subiculum*

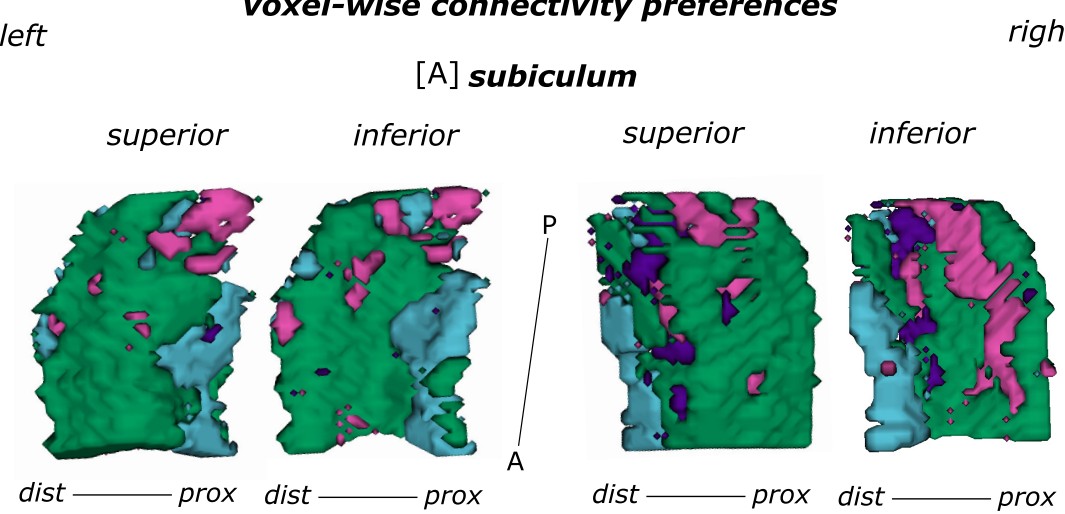

### [B] *CA1*

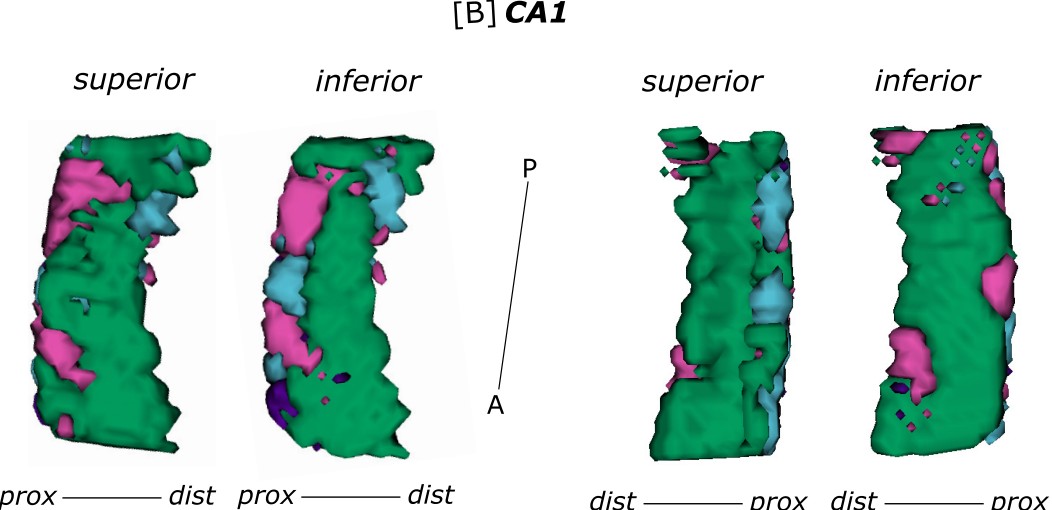

**Appendix 4—figure 1.** *Functional connectivity preferences to entorhinal seeds along the subiculum and CA1 transversal axis.* Displayed are the results of a seed-to-voxel functional connectivity analysis between entorhinal seeds and the left and right subiculum [A] and CA1 [B] subregion. Voxel-wise connectivity preferences to the entorhinal seeds on group level are shown from a superior and an inferior perspective on the respective subregion. The figure displays the same data as in *Appendix 1—figure 2* and *Figure 2* and is based on *Source code 9–16*. The color coding refers to the respective entorhinal seed: green - $EC_{RSC-based}$; purple - $EC_{Area36-based}$; blue - $EC_{PHC-based}$ and pink - $EC_{Area35-based}$ seed. M – medial; L – lateral; A – anterior; P – posterior; prox – proximal; dist – distal.

## Appendix 5

### Object and scene processing in cortical source regions

To examine whether lower parameter estimates for object processing could be due to increased noise in this condition, we evaluated object and scene processing in the four cortical source regions. Therefore, we extracted parameter estimates for the object versus baseline and the scene versus baseline contrast from the retrosplenial and parahippocampal cortex and from perirhinal Area 36 and Area 35, respectively. All parameter estimates were extracted from the previously segmented regions of interests, coregistered to the individual EPI space.

Repeated-measures ANOVAs in both hemispheres showed a significant interaction effect between condition and region (right: $F_{(3,93)}$ = 60.4229; p<0.001; left: $F_{(3,93)}$ = 47.3421; p<0.001). Subsequent paired-samples T-tests show significantly more functional activity in the object than scene condition in Area 36 (bilateral: $p_{FDR}$ < 0.001) and the left Area 35 ($p_{FDR}$= 0.0011). No significant difference between object and scene conditions is observed in the right Area 35 ($p_{FDR}$ = 0.9821). There is a significant effect of more functional activity in the scene than object condition in the parahippocampal (bilateral: $p_{FDR}$<0.001) and retrosplenial cortex (bilateral: $p_{FDR}$<0.001, see *Appendix 5—figure 1*).

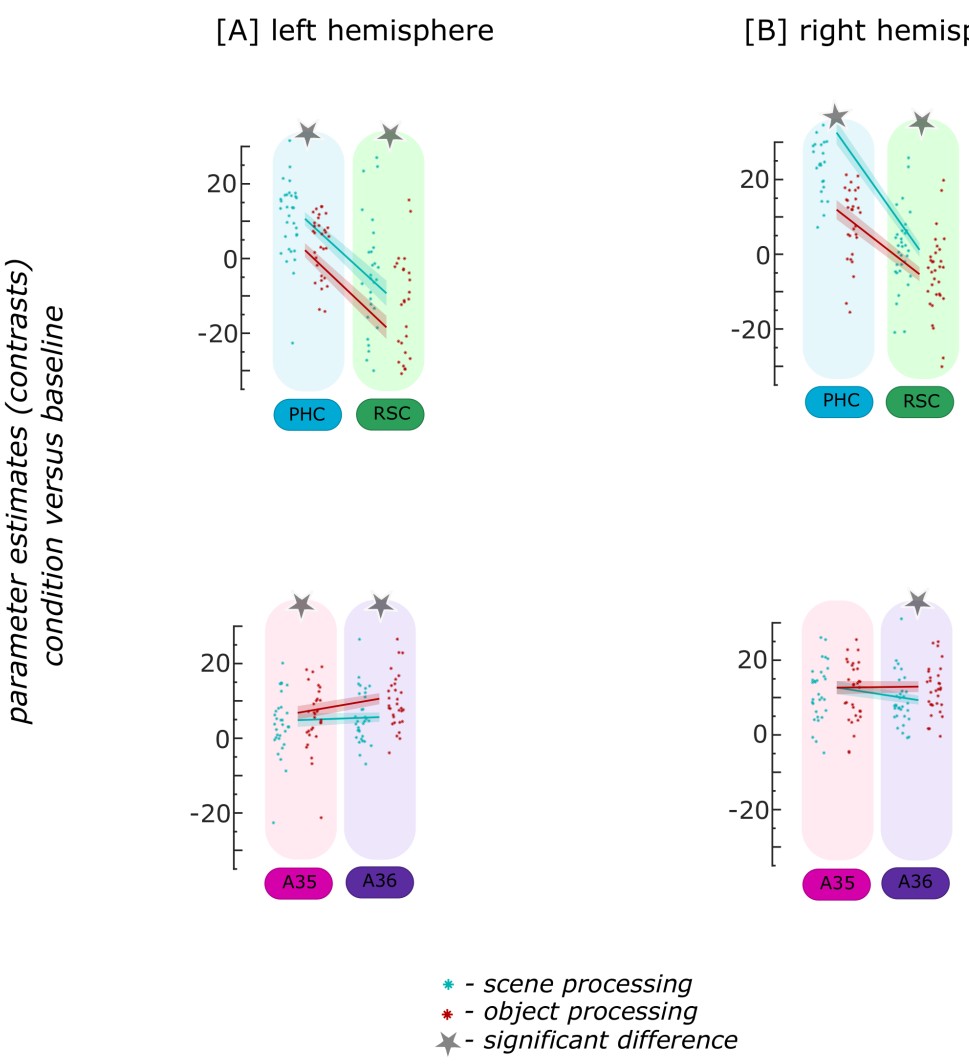

**Appendix 5—figure 1.** Functional activity during scene and object conditions in cortical source regions. Displayed are the extracted parameter estimates for the object versus baseline contrast (red) and the scene versus baseline contrast (cyan) from four cortical source regions in the [A] left and [B] right hemisphere, per individual *Appendix 5—figure 1 continued on next page*

*Appendix 5—figure 1 continued*

(dots) and summarized across individuals (lines). Repeated measures ANOVAs revealed a significant interaction between condition and cortical source region in both hemispheres. The displayed significant differences (asterisks) were obtained with FDR-corrected post-hoc tests and refer to $p < 0.05$, sample size n = 32. During the object condition, participants were presented with 3D rendered objects on screen, during the scene condition with 3D rendered rooms and during the baseline condition they saw scrambled pictures. The shaded area around the lines refer to standard errors of the mean. PHC – parahippocampal cortex (blue), RSC – retrosplenial cortex (green), A35 – perirhinal Area 35 (pink), A36 – perirhinal Area 36 (purple). *Appendix 5—figure 1—source data 1* contains extracted parameter values from cortical source regions (left – lSources, right – rSources, isthmuscingulate – retrosplenial) for the object versus baseline and scene versus baseline conditions per individual.

The online version of this article includes the following source data for appendix 5—figure 1:

**Appendix 5—figure 1—source data 1.** Individual parameter estimates for scene and object processing in cortical source regions.

The increased object processing in adjacent cortical source regions indicates that noise differences across conditions are not likely to cause the lack of increased object processing within entorhinal seed regions and hippocampal subregions.

## Appendix 6

### Functional connectivity analysis to determine entorhinal seeds

Before performing the core functional connectivity analysis between entorhinal seeds and hippocampal voxels, we had to determine the entorhinal seeds, that is, the functional subregions of the EC. We largely followed *Maass et al., 2015* approach to assure comparability of results. The seeds were determined based on their functional connectivity with functionally relevant sources from the cortical object and scene information processing streams, that are the perirhinal Area 35 and Area 36, the parahippocampal cortex and the retrosplenial cortex (see *Nilssen et al., 2019*).

The CONN toolbox (*Whitfield-Gabrieli and Nieto-Castanon, 2012*) was applied to perform a seed-to-voxel semipartial correlation analysis on the residual fMRI data between the retrosplenial, parahippocampal, Area 35 and Area 36 sources and the voxels within the segmented EC mask of each individual (see the description of the core functional connectivity analysis for the precise parameters). The resulting z-transformed correlation maps were then aligned for each participant to the group template T1 space and subjected to four one-sample T-tests (one for each source preference map) to reveal significant clusters of entorhinal connectivity preferences per source across all other entorhinal seeds, respectively. The functional subregions in the EC that we identify on group level generally overlap for the preferences towards the perirhinal cortex (Area 35 and Area 36) and towards the parahippocampal cortex with the findings by *Maass et al., 2015*. The exact procedure to determine the entorhinal seeds for further analysis is described in the main article.

## Appendix 7

### Co-registration procedure and alignment assessment

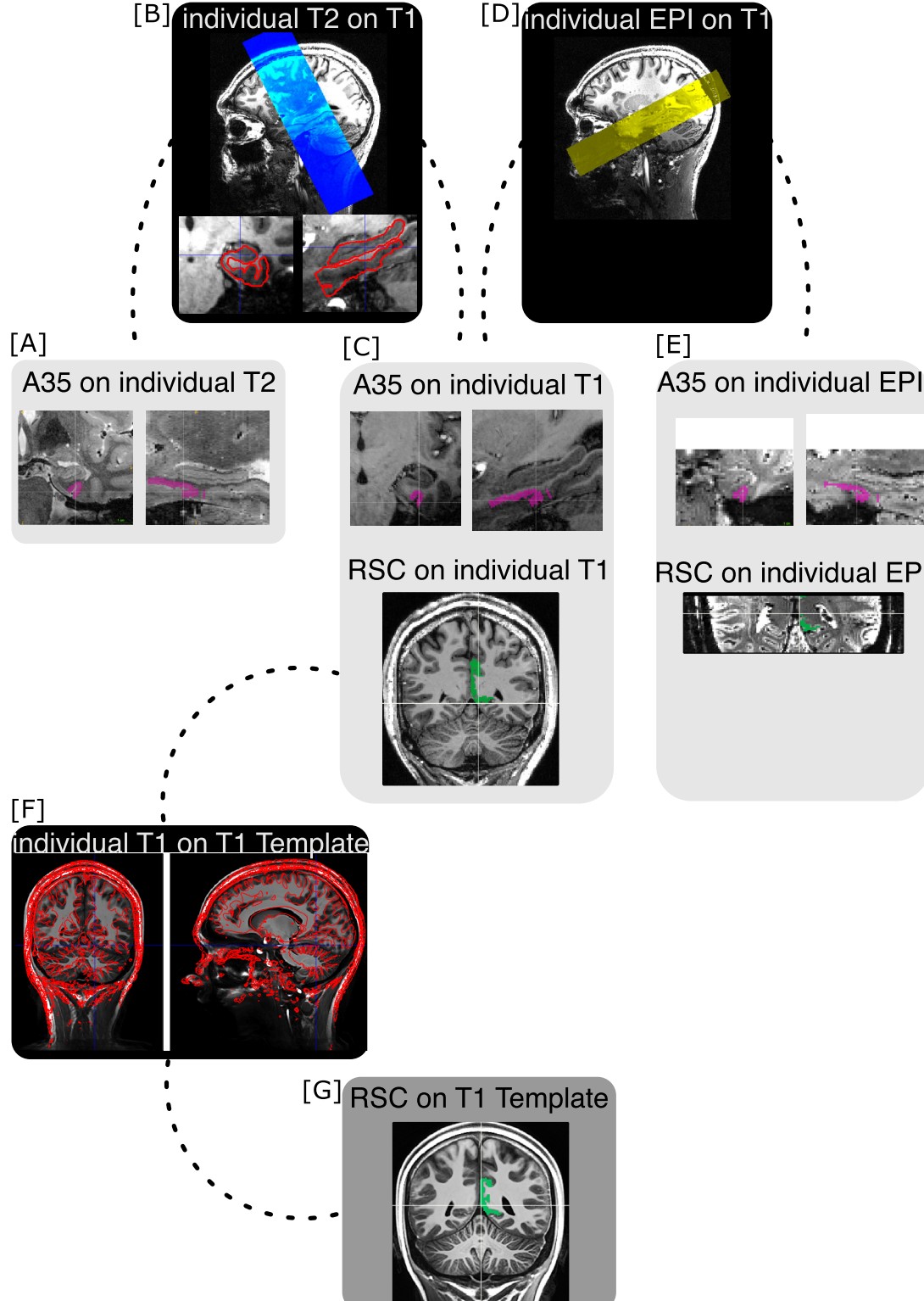

**Appendix 7—figure 1.** Co-registration procedure. [A] Medial temporal lobe regions of interest (ROIs) were segmented on individual T2 images. Displayed is an example region (perirhinal Area 35, A35, pink) on a

*Appendix 7—figure 1 continued on next page*

*Appendix 7—figure 1 continued*

representative example individual T2 image. [B] Individual T2 images (blue overlay) were co-registered to whole-brain individual T1 images (upper image). The resulting warping matrices were applied to transfer the segmented ROIs from individual T2 space to individual T1 space. The co-registration procedure was manually evaluated based on landmarks (lower two images and [C]). [C] Displayed is the same example region Area 35 of the same example individual on corresponding coronal (left) and sagittal (right) slices on the individual's T1 image (upper two images). [D] Individual echo-planar images (EPI, yellow overlay) have been co-registered to the whole-brain individual T1 images as well. [E] The inverse warping matrices were applied to warp segmented ROIs from individual T1 space to the individual EPI space. Displayed is the same example region Area 35 of the same example individual on corresponding coronal (left) and sagittal (right) slices on the individual's EPI image (upper two images). The warping result was manually evaluated based on landmarks. [F] To evaluate results on group level, all individual T1 images were averaged to create a sample-specific T1 template. Displayed is the overlay (red boundaries) of an individual T1 image on the sample-specific T1 template. The resulting warping matrices were applied to move segmented ROIs from the individual T1 space to the sample-specific T1 template. [G] The retrosplenial cortex (RSC, green) ROI was originally segmented on the sample-specific T1 template. Respective (inverse) warping matrices were applied to move the retrosplenial ROI from the sample-specific T1 template to the individual T1 ([C], lower image) and EPI ([E], lower image) spaces. Landmark-based manual evaluation was applied to all co-registration steps. Displayed is the retrosplenial ROI (green) of an example individual on corresponding coronal slices on the individual T1 image ([C], lower image) and EPI ([E], lower image).

## Appendix 8

### Quality assurance measures of manually segmented regions-of-interest

The individual regions of interest were segmented by the same two experienced raters that also segmented a subsample of our data (24 hemispheres of 22 participants) for a previous publication (*Berron et al., 2017*). Quality assurance measures were calculated for that subsample. Regarding intra-rater reliability, the dice similarity coefficients are above 0.88 for all segmented regions (region-specific means (SD) are as follows: PHC 0.93 (0.03); Area 36 0.91 (0.02); Area 35 0.88 (0.02); EC 0.91 (0.01)). The intraclass-correlation coefficients for intra-rater reliability are all above 0.95 (PHC 0.99; Area 36 0.96; Area 35 0.97; EC 0.98). For the inter-rater reliability, dice similarity coefficients are above 0.84 for all segmented regions (region-specific means (SD) are as follows: PHC 0.86 (0.12); Area 36 0.91 (0.02); Area 35 0.84 (0.05); EC 0.87 (0.02)). The intraclass-correlation coefficients for inter-rater reliability are all above 0.78 (PHC 0.94; Area 36 0.88; Area 35 0.87; EC 0.94; see *Berron et al., 2017*).

## Appendix 9

### Metrics for transversal subiculum and CA1 segments

Transversal subiculum and CA1 segments were cut on the group template T1 images. The average number of voxels contained in each subiculum segment was 460.8 voxels for the left subiculum (standard deviation 104.36) and 458 voxels for the right subiculum (standard deviation 75.09). For the left CA1 the average equals 360 voxels (standard deviation 27.58) and 335 voxels for the right CA1 segments (standard deviation 3.56, see *Appendix 9—table 1* for segment-specific values and *Appendix 9—figure 1* for an illustration).

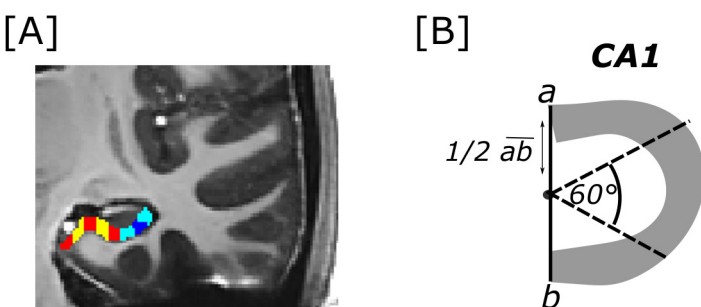

**Appendix 9—figure 1.** Transversal subiculum and CA1 segments. [A] Displayed are segments cut along the transversal subiculum (red and yellow) and CA1 (cyan and dark blue) axis in the right hemisphere. Segments were cut on coronal images (as displayed in the example image) on the study-specific T1 template. [B] To cut CA1 segments, the endpoints of the transversal CA1 axis (**a and b**) were connected. From the middle point of that line CA1 was cut into three segments by two lines oriented in 60° angles from the line that connected a and b.

**Appendix 9—table 1.** Number of voxels in transversal subiculum and CA1 segments for each hemisphere.

| | left hemisphere (distal to proximal segments) | | | | | right hemisphere (distal to proximal segments) | | | | |
|---|---|---|---|---|---|---|---|---|---|---|
| subiculum | 340 | 419 | 511 | 396 | 638 | 338 | 465 | 451 | 460 | 575 |
| CA1 | 399 | | 341 | 340 | | 337 | | 330 | | 338 |

