## [Editor Report]

Grande and colleagues provide important new insights into how different regions of the entorhinal cortex functionally interact with specific cortical brain areas and how, in turn, subregions of the entorhinal cortex interact with the hippocampus during 'scene' and 'object' processing. The study is well-motivated, well-designed, and provides convincing evidence using appropriate methodology. This paper is relevant to cognitive neuroscientists with an interest in the entorhinal cortex – hippocampal pathways and 'scene' and 'object' representation in the medial temporal lobe.

---

## [Decision Letter]

**Decision letter after peer review:**

Thank you for submitting your article "Functional connectivity and information pathways in the human entorhinal-hippocampal circuitry" for consideration by *eLife*. Your article has been reviewed by 3 peer reviewers, and the evaluation has been overseen by a Reviewing Editor and Laura Colgin as the Senior Editor. The following individuals involved in review of your submission have agreed to reveal their identity: Marshall A Dalton (Reviewer #1); Menno P Witter (Reviewer #2).

Essential revisions:

Introduction/terminology:

1. There are many different terms used interchangeably throughout the manuscript and this is confusing since it is not immediately apparent whether they cover similar notions. Throughout the paper the phrase transversal axis is used differently in many different places; usage varies from in the abstract: 'cortical – entorhinal interaction and the circuitry's inner communication along the transversal axis'; in the introduction: 'from the EC towards the transversal hippocampal axis', or 'information continues to flow in a segregated manner along the transversal human entorhinal-hippocampal axis'; in the beginning of the discussion, page 14 line 9: the 'transversal entorhinal-hippocampal axis' and there are many more. It seems that you really refer to the transverse/transversal axis of CA1/subiculum so define that, give it a name or an abbreviation if you want, and use that consistently.

2. On page 6 line 15 you state/define that you are working with two parallel input streams traditionally associated with information on object ("item") versus scene ("contextual"). In the discussion on page 14 you start with a heading that carries 'contextual information. In the following text you talk about scene representation, which is ok if one remembers that you have defined the two as synonyms, but it becomes confusing when you suddenly jump to spatial or contextual information (line 31). So, I suggest that again you start with defining the terms you want to use and use them throughout; scene=context=space and object = item.

3. In general, the Introduction could be streamlined and the research question(s) described more clearly. The authors should incorporate hypotheses/predictions in the Introduction given that numerous theoretical models make very specific predictions regarding the neuroanatomical underpinnings of scene/object processing (i.e., see Dalton and Maguire, 2017. Curr Opin Behav Sci. 17: 34-40 for an example relating to human hippocampal subfields).

4. On page 18, lines 15 to 20, there follows a paragraph that is so dense of information, that the two concepts that I think the authors are trying to convey both might get lost. You mention 'contextual features of an item' that might be the result of convergence taking place in the anterior portion of EC, i.e. outside the hippocampus, and you contrast that with a pure contextual loop that allows for subsequent converge (with what) in the hippocampus. That second loop is associated with the distal subiculum, whereas the first is actually associated with the proximal subiculum/distal CA1 border. You might want to emphasize that more deliberately. Moreover, what is the evidence that this assumed convergence in EC actually occurs, i.e. what is the evidence that anterior-lateral EC and anterior-medial EC communicate with one another; actually available data in rodents and monkeys argue against that and favor a predominant interaction along the AP-axis. Note that the summary in the abstract adds to the confusion referred to at the second bullet: you mention that the anterior-medial EC, defined through the RSC source processing scene information, whereas the second loop processes only contextual information, implying that scene and context are two different things.

Methods:

5. Considering a major strength of the paper is the detail with which anatomical ROIs were created, I'm curious to know how the authors accounted for variability in MTL anatomy across participants on steps that required warping of ROIs into a group space or, in the case of the RSC mask, warping from group template space to individual participant space. Importantly, how were the ROIs assessed for their accuracy of alignment on the EPI images?

6. Related to this, no quality assurance measures have been presented for the manually created ROIs. More information relating to these should be included in the methods section. For example, could the authors describe how many people were involved in manually segmenting the ROI's? If only one person, could the authors provide the results of intra-rater reliability analyses using, for example, the DICE overlap metric for each ROI? If more than one person was involved, could the authors also provide the results of inter-rater reliability analyses? In addition, considering it can sometimes be difficult to warp small ROIs such as hippocampal subfields from structural to EPI images with sufficient accuracy, could the authors provide some visual representations (perhaps as a supplemental figure) showing, for example, that the subiculum and CA1 ROIs created on the structural images were well aligned with the subiculum and CA1 on EPI images.

7. For ROIs whose borders lie immediately adjacent each other (e.g., medial EC and distal subiculum; anterior PHC and posterior EC), did the authors take steps to reduce the possibility of fMRI signal 'bleeding' between ROIs by, for example, ensuring the distance between borders of each ROI was greater than the spatial smoothing kernel used on the functional data? If so, could you please describe these steps in the methods section?

8. On page 20, lines 29-30, the authors describe the anatomical landmark used to create the anterior most slice of the hippocampal subfield masks but do not describe the landmark used to demarcate the posterior most extent of the masks. This should also be described.

9. The authors state that the subiculum and CA1 subfields were sagittally cut into "equally wide" segments (page 20; line 32) but no metrics relating to this are provided. Could the authors provide some metrics to support this? For example, was the mean number of functional voxels contained within each segment equivalent or did these substantially differ?

Results:

10. Although this is described in the methods section, perhaps the authors could briefly mention early in the Results section that the fMRI dataset used in Analysis 1 and 2 are derived from a task-based dataset where task-related effects have been regressed out to create a "rest-like" dataset. It may also be helpful to readers interested in (or cautious of) this particular method, to cite previous work showing that these "rest-like" datasets yield similar results to those obtained from resting-state data (e.g., Gavrilescu et al., Hum Brain Mapp 29:1040-1052, 2008 or similar).

11. Functional subregions of the EC associated with RSC, PHC, A35 and A36 are displayed very clearly in Figure 1. It appears that the EC seed relating to A36 is less 'localised' than the other seeds and displays roughly equal size clusters in both the posterior lateral and anterior medial EC. However, in the text, the authors suggest that A36 shows preferential connectivity with the posterior lateral EC (page 7; lines 7-8). How did the authors come to preferentially associate A36 with the posterior lateral EC rather than with the anterior medial EC? Did the authors conduct any additional quantitative assessments to determine this?

12. While I understand the primary focus of the subiculum and CA1 analyses relate to their transverse axis, it seems clear when looking at Figure 2, that there may also be interesting anterior-posterior differences along the body of these subfields. For example, functional clusters are largely absent in anterior portions of segment 3 of the subiculum but posterior portions of segment 3 contain functional clusters. Observations relating to the anterior-posterior axis are not discussed in text but may be informative to those with an interest in anterior-posterior variations in hippocampal function.

13. All the data are described in terms of preferred connectivity, which is what the data show. However, by mapping the preferred connectivity as absolute color maps with sharp borders, a lot of the complexities, i.e. gradients are lost. It would be great if the authors could provide supplementary maps of each source in isolation and how it maps across the surface of EC and the transverse axis of CA1/subiculum.

Interpretation:

14. The authors created ROI masks for the subiculum and refer to medial portions of this mask as 'distal subiculum'. However, the mask appears to encompass the entire 'subicular complex' inclusive of the subiculum, presubiculum and parasubiculum. Importantly, the area referred to as 'distal subiculum' throughout the manuscript (specifically, segments 1 and 2 of the subiculum mask) most likely aligns with the location of the pre- and parasubiculum in addition to containing portions of the distal subiculum. This is important considering the growing body of evidence that the pre- and parasubiculum are preferentially engaged during scene-based cognition. While this is briefly touched on in the discussion (page 13; line 5-6), I feel the interpretation of results as they relate to the 'distal subiculum' could benefit from being placed more firmly in the context of this growing body of evidence relating to the involvement of medial aspects of the hippocampus (inclusive of the pre- and parasubiculum) in scene-based cognition.

15. I am afraid that the central premise of the 'two segregated memory streams' is somewhat outdated, certainly at the level of the entorhinal cortex and hippocampus (see Doan et al. 2019). Regarding the rodent homologue of the parahippocampal cortex (referred to as POR, or postrhinal cortex) Doan et al. write: 'Postrhinal cortex preferably targets lateral instead of medial entorhinal cortex', and that '..dorsolateral parts of LEC receive inputs from both POR and PER and that both these projections show very similar topological features..'

16. Nevertheless, in the present study the authors claim that 'The lateral EC preferentially communicates with the perirhinal cortex' (page 3, line 20). In the same paragraph they write that context processing should be specific to the MEC, in contrast to the LEC that would rather process non-spatial and item information. Strikingly, they then go on to cite Doan et al. 2019 and effectively contradict themselves within the same paragraph '…parahippocampal cortex communicates with the EC along its full extend' (as Doan et al. write, the LEC is similarly connected to POR (PHC) and PER). This means that contextual information from PHC should not uniquely be passed on to the MEC. On the contrary, it should even be more strongly passed on to the LEC than the MEC.

17. I am rather puzzled why the authors cite Nilssen et al. and Doan et al. and still uphold the segregated stream hypothesis of entorhinal and hippocampal subregions. In my reading, those studies are entirely incompatible with the view of two parallel, segregated memory streams involving the LEC and MEC. Ironically, the null finding of category specific processing in the lateral cluster of EC voxels, and associated regions, might be due to the convergent, multimodal input patterns. Unfortunately, the two segregated stream hypothesis seems to be a central tenet in the current study.

18. There seems to be no main effect of activations to object stimuli (difference to zero) in any region. Nevertheless the authors claim that regions show processing of object-related information. This is not warranted based on the results. For example in the abstract: 'The regions of another route, that connects the anterior-lateral EC and a newly identified retrosplenial-based anterior-medial EC subregion with the CA1/subiculum border, process object and scene information similarly'. If there is no evidence of any level of processing, it is not warranted to claim 'similar processing'.

19. Based on the introduction, a double dissociation between object and scene processing in two segregated sets of regions is expected, but the authors don't provide evidence for this. In my view the interaction effect (and post-hoc tests) for higher scene processing in segment 1and2 in Figure 4 is a very interesting and meaningful finding. However, neither main effects nor interactions are presented to support claims of any level of 'object processing'. How can we be sure that we are not looking at pure noise in the object condition? Absence of evidence is not equal to evidence of absence (of different object/ scene processing).

20. No 'information flow' is actually assessed in this study. In my view, this would require a directed connectivity analysis such as dynamic causal modeling or transfer entropy. The research question or hypotheses should pertain to what is actually being done. For example, 'we predict functional connectivity between X-Y to reflect the structural connectivity described in rodents/ humans previously', and 'we expect specific sensitivity to scene stimuli in distal subiculum, because of connectivity to the MEC region in rodents'. This would be 'consistent with information flow' between those regions, but it is not directly or conclusively showing it.

21. The approach to segment the EC into sub-clusters based on known connectivity to other regions seems fine in general. However, this should be guided by gold-standard, a priori knowledge of entorhinal subregions such as the rodent MEC and LEC (or finer cyto/ myeloarchitectonic subdivisions, for example described in Kimer et al. 1997). In the present study, the authors select four cortical seed regions to segment entorhinal subregions based on functional connectivity, without providing appropriate justification how some of those regions would uniquely connect to specific, cytoarchitectonically-defined entorhinal subregions. For example, BA35, BA36, and PHC all project to the rodent LEC, and PHC also project (albeit weaker) to the MEC. Which a-priori defined entorhinal subregions should be uniquely identified with these four seed regions? Following the semipartial correlation analyses, clusters of EC voxels are then segmented and labeled posterior-medial, posterior-lateral, anterior-medial and anterior-lateral EC. Those names are then used quasi synonymously with coherent and established, cytoarchitectonically defined subregions such as the rodent LEC and MEC. This seems egregious, not only because of the questionable choice of initial seed regions, but more so because of the discontinuous topography of the segmentations shown in Figure 1. What is the ground-truth (cytoarchitecture, myeloarchitecture, tracing studies, gene expression) evidence for a salt-and-pepper organization of entorhinal subregions? The clusters can't correspond to coherent cytoarchitectonic regions, so why would they be referred to as such.

22. Why did the authors not simply use the ROIs from their previous identification of the human homologues of the MEC and LEC (Maass et al. 2015) to address their current research questions? In summary, this could be reconciled by a consistently used naming scheme for the entorhinal seeds that avoids confusion with cytoarchitectonic subregions.

23. In the results you describe data on proximal CA1, but you do not mention them anywhere in the paper explicitly. If your data do not allow you to functionally 'interpret' proximal CA1, you might come back to that in the discussion, state this and mention that in the rodent literature there is a gradient along transverse CA1 with more precise spatial information in proximal than in distal. Any ideas of why that does not seem to hold in humans? In particular on page 15, line 31 you make a very general statement about the transverse organization of CA1, so that might be the place to elaborate a bit more.

Clinical relevance:

24. The reported findings may have clinical relevance, but this is to be determined by future studies. An entire paragraph in the Introduction is dedicated to this topic (starting on page 4, line34). The authors state that the findings show that tau pathology spreads through 'functionally connected' regions. Functionally connected regions must also be structurally connected (through mono or polysynaptic connections). It seems the authors are insinuating that functional and structural connections are independent. The prediction of functional connectivity between parahippocampal, perirhinal and entorhinal cortices, and the subiculum is already abundantly well founded on previous findings of tracing studies in rodents and primates, and even the authors' previously published functional MRI findings (Maass et al. 2015). The fact that the spreading of tau pathology follows structural and functional connections provides no additional predictive value to inform the research question (in my understanding these are: (1) is there a specific functional connectivity pattern between regions, (2) do specific regions show differential fMRI activations for object vs scene stimuli). It appears unwarranted and misleading to portray the clinical findings as a formal motivation (meaning, previous findings that form the basis for the research hypotheses) for the present study – page 5, line 8. If this was not the intention, then I feel this needs to be formulated more clearly.

*Reviewer #1 (Recommendations for the authors):*

I'd first like to congratulate the authors for their fine and detailed work investigating entorhinal-hippocampal information pathways. It is exciting to see functional imaging experiments with such a focus on anatomical detail and I look forward to reading future work from these authors. The paper does a very good job at underscoring the importance of characterising the functional organisation of entorhinal-hippocampal pathways and their relationship to surrounding extra-hippocampal cortices. The authors present exciting new evidence that extra-hippocampal cortical areas display preferential patterns of functional connectivity with subregions of the entorhinal cortex (EC). These EC subregions, in turn, display preferential patterns of functional connectivity along the transverse axis of the subiculum and area CA1 of the hippocampus. In addition, the authors show that the posterior medial EC and distal subiculum are preferentially engaged during 'scene' processing. In contrast, anterior portions of the EC and the CA1/subiculum border engage equally during both 'scene' and 'object' processing. Overall, this work has important implications for our understanding of information transfer between the entorhinal cortex and hippocampus.

My suggestions below, rather than criticisms, largely reflect potential issues/questions that I believe, if addressed, may improve the clarity and interpretation of results.

1) Considering a major strength of the paper is the detail with which anatomical ROIs were created, I'm curious to know how the authors accounted for variability in MTL anatomy across participants on steps that required warping of ROIs into a group space or, in the case of the RSC mask, warping from group template space to individual participant space. Importantly, how were the ROIs assessed for their accuracy of alignment on the EPI images?

2) Related to this, no quality assurance measures have been presented for the manually created ROIs. More information relating to these should be included in the methods section. For example, could the authors describe how many people were involved in manually segmenting the ROI's? If only one person, could the authors provide the results of intra-rater reliability analyses using, for example, the DICE overlap metric for each ROI? If more than one person was involved, could the authors also provide the results of inter-rater reliability analyses? In addition, considering it can sometimes be difficult to warp small ROIs such as hippocampal subfields from structural to EPI images with sufficient accuracy, could the authors provide some visual representations (perhaps as a supplemental figure) showing, for example, that the subiculum and CA1 ROIs created on the structural images were well aligned with the subiculum and CA1 on EPI images. I believe such a representation would increase reader confidence.

3) For ROIs whose borders lie immediately adjacent each other (e.g., medial EC and distal subiculum; anterior PHC and posterior EC), did the authors take steps to reduce the possibility of fMRI signal 'bleeding' between ROIs by, for example, ensuring the distance between borders of each ROI was greater than the spatial smoothing kernel used on the functional data? If so, could you please describe these steps in the methods section?

4) The authors created ROI masks for the subiculum and refer to medial portions of this mask as 'distal subiculum'. However, the mask appears to encompass the entire 'subicular complex' inclusive of the subiculum, presubiculum and parasubiculum. Importantly, the area referred to as 'distal subiculum' throughout the manuscript (specifically, segments 1 and 2 of the subiculum mask) most likely aligns with the location of the pre- and parasubiculum in addition to containing portions of the distal subiculum. This is important considering the growing body of evidence that the pre- and parasubiculum are preferentially engaged during scene-based cognition. While this is briefly touched on in the discussion (page 13; line 5-6), I feel the interpretation of results as they relate to the 'distal subiculum' could benefit from being placed more firmly in the context of this growing body of evidence relating to the involvement of medial aspects of the hippocampus (inclusive of the pre- and parasubiculum) in scene-based cognition.

5) Functional subregions of the EC associated with RSC, PHC, A35 and A36 are displayed very clearly in Figure 1. It appears that the EC seed relating to A36 is less 'localised' than the other seeds and displays roughly equal size clusters in both the posterior lateral and anterior medial EC. However, in the text, the authors suggest that A36 shows preferential connectivity with the posterior lateral EC (page 7; lines 7-8). How did the authors come to preferentially associate A36 with the posterior lateral EC rather than with the anterior medial EC? Did the authors conduct any additional quantitative assessments to determine this?

6) I am a little confused by the colour scheme in the 3D hippocampal subfield models presented in Figures 2, 4 and S2. No EC seeds are colour coded brown but functional clusters colour coded brown are present in both the subiculum and CA1. Does brown represent an overlap between clusters? If not, could the authors please explain what these brown functional clusters correspond to?

7) While I understand the primary focus of the subiculum and CA1 analyses relate to their transverse axis, it seems clear when looking at Figure 2, that there may also be interesting anterior-posterior differences along the body of these subfields. For example, functional clusters are largely absent in anterior portions of segment 3 of the subiculum but posterior portions of segment 3 contain functional clusters. Observations relating to the anterior-posterior axis are not discussed in text but may be informative to those with an interest in anterior-posterior variations in hippocampal function.

8) The authors state that the subiculum and CA1 subfields were sagittally cut into "equally wide" segments (page 20; line 32) but no metrics relating to this are provided. Could the authors provide some metrics to support this? For example, was the mean number of functional voxels contained within each segment equivalent or did these substantially differ?

9) For Analysis 3, the authors report the results of contrasts between the 'object' condition vs 'baseline and the 'scene' condition vs 'baseline'. Is there a reason the authors did not report results relating to the direct contrast of 'scene' vs. 'object' conditions? While acknowledging the visual stimuli between these two conditions would not be equally matched, it would nonetheless be interesting to see if, for 'scenes' > 'objects', the posterior medial EC – distal subiculum effect remains. It would also be interesting if the 'object' > 'scene' contrast revealed effects not observed for the contrast of 'object' > 'baseline'. Could the authors offer some insights into why these direct contrasts were not analysed?

10) Although this is described in the methods section, perhaps the authors could briefly mention early in the Results section that the fMRI dataset used in Analysis 1 and 2 are derived from a task-based dataset where task-related effects have been regressed out to create a "rest-like" dataset. It may also be helpful to readers interested in (or cautious of) this particular method, to cite previous work showing that these "rest-like" datasets yield similar results to those obtained from resting-state data (e.g., Gavrilescu et al., Hum Brain Mapp 29:1040-1052, 2008 or similar).

11) It may also be helpful to specify early in the Results section that hippocampal subfield analyses were conducted only on the body of the hippocampus.

*Reviewer #2 (Recommendations for the authors):*

The authors are to be complimented with an impressive combinatorial attempt to understand the organization of the entorhinal cortex and its position as part of the cortico-hippocampal processing streams in the human brain, which are critical components of the human episodic memory system. This is an important and as yet unresolved issue, largely due to the lack of proper imaging technology. Even in experimental animal studies with all the current versatile experimental tools to selectively manipulate parts of the circuits, the organization of these parallel cortical streams is as yet not clear. The conclusions are supported by strong data, obtained through experiments that, as far as I can evaluate, are sound. In particular the use of several carefully chosen cortical sources to differentiate between four 'functionally' different entorhinal seeds that subsequently are mapped onto CA1 and subiculum is a major contribution to the field.

Having said this, I strongly feel that the manuscript could improve substantially; a critical reappraisal on how the experimental data are presented and how the overall topic is introduced and handled is recommended. There are many different terms used interchangeably throughout the manuscript and this is confusing since it is not immediately apparent whether they actually cover similar notions:

• Throughout the paper you use transversal axis in many different phrases and context resulting in massive confusion; usage varies from in the abstract: 'cortical – entorhinal interaction and the circuitry's inner communication along the transversal axis'; in the introduction: 'from the EC towards the transversal hippocampal axis', or 'information continues to flow in a segregated manner along the transversal human entorhinal-hippocampal axis'; in the beginning of the discussion, page 14 line 9: the 'transversal entorhinal-hippocampal axis' and there are many more. It seems that you really refer to the transverse/transversal axis of CA1/subiculum so define that, give it a name or an abbreviation if you want, and use that consistently. This is important since EC has a transverse axis of its own, which you apparently refer to as the lateral-to-medial axis, but that does not simply relate to the transversal hippocampal axis.

• On page 6 line 15 you state/define that you are working with two parallel input streams traditionally associated with information on object ("item") versus scene ("contextual"). In the discussion on page 14 you start with a heading that carries 'contextual information. IN the following text you talk about scene representation, which is ok if one remembers that you have defined the two as synonyms, but it becomes confusing when you suddenly jump to spatial or contextual information (line 31). So, I suggest that again you start with defining the terms you want to use and use them throughout; scene=context=space and object = item. If you want to use all these different concepts interchangeable, you might want to consider to expand this even further; although you have not tested sequence coding, would you be inclined to extend object=item=sequence and how do the two sets of terms relate to the proposal of allocentric versus egocentric representations as proposed among others by Knierim in some of his recent papers. To further complicate this potential confusion: objects may be part of a context or a scene. (Note that I do not have the solution for this nomenclatural nightmare either, but try to keep it as simple as possible in this manuscript).

• On page 18, lines 15 to 20, there follows a paragraph that is so dense of information, that the two concepts that I think the authors are trying to convey both might get lost. You mention 'contextual features of an item' that might be the result of convergence taking place in the anterior portion of EC, i.e. outside the hippocampus, and you contrast that with a pure contextual loop that allows for subsequent converge (with what) in the hippocampus. That second loop is associated with the distal subiculum, whereas the first is actually associated with the proximal subiculum/distal CA1 border. You might want to emphasize that more deliberately. Moreover, what is the evidence that this assumed convergence in EC actually occurs, i.e. what is the evidence that anterior-lateral EC and anterior-medial EC communicate with one another; actually available data in rodents and monkeys argue against that and favor a predominant interaction along the AP-axis. Note that the summary in the abstract adds to the confusion referred to at the second bullet: you mention that the anterior-medial EC, defined through the RSC source processing scene information, whereas the second loop processes only contextual information, implying that scene and context are two different things.

Four more general comments:

• In the results you do describe data on proximal CA1, but you do not mention them anywhere in the paper explicitly. If your data do not allow you to functionally 'interpret' proximal CA1, you might come back to that in the discussion, state this and mention that in the rodent literature there is a gradient along transverse CA1 with more precise spatial information in proximal than in distal. Any ideas of why that does not seem to hold in humans? In particular on page 15, line 31 you make a very general statement about the transverse organization of CA1, so that might be the place to elaborate a bit more.

• I found it of interest that in the most medial parts, likely the ambiens gyrus area in both the left and right hemisphere there seems to be an indication of a second representation of four seeds associated with the four sources. Is this correct and would the authors care to comment on this?

• You divide the hippocampus along the transverse axis and EC along the lateral-medial (transverse) axis and the anterior-posterior axis. The hippocampus also has an AP axis and even though I learned from the method section that you only sampled the body, leaving out the uncal portion and the tail, for good reasons, there is still a pretty substantial AP axis left to sample from; any differences along the AP axis? If not, it might be relevant to mention that explicitly.

• All the data are described in terms of preferred connectivity, which is what the data show. However, by mapping the preferred connectivity as absolute color maps with sharp borders, a lot of the complexities, i.e. gradients are lost. It would be great if the authors could provide supplementary maps of each source in isolation and how it maps across the surface of EC and the transverse axis of CA1/subiculum.

I have several more detailed, smaller comments listed below in sequence as they appear in the paper (not relevance):

Page 5, line 16/17: The way the EC communicates with different regions of these two cortical streams implies topographical differences in information processing within the EC. Please rephrase and make it more concrete.

Page 6, line 29: This is to our knowledge…; does this refer to Syvertsen et all or to the present study?

Page 6, last line you mention tau pathology as being specifically occurring in this part of the brain. Of course, that is correct in AD but not in many other tau-opathies so you might want to add AD, which your first do only 7 lines further down.

Page 12, line 3: Please carefully read this heading: it states the opposite of the sentence in lines 10/11. The latter is likely correct as this is repeated throughout the paper.

Page 15, line 2: insert full stop between 'Our'

Page 16 lines 15-19, please rephrase. The projections from RSC to deep layers on EC in the rodent are really limited to the area defined as MEC based on connectivity and that is true in the monkey as well. The area defined here as the RSC connected area seems a lot more anterior, and has been 'considered' to be the medial part of LEC. So this needs better wording to avoid further confusion.

Likewise, page 16, lines 23 to 25 are a bit misleading, again as the result of the complexity of terminology and the risk of circular definitions/statements. If RSC is a defining input of MEC, then the statement 'The mapping of the anterior-medial EC (identified by retrosplenial connectivity) to the subiculum/CA1 border opposes conventional views that the medial EC communicates with the distal subiculum and proximal CA1 (based on rodent anatomy – see e.g. Nilssen et al., 2019)' is correct and it shows that in the human the connectivity or anterior medial entorhinal cortex might be different. However, the next sentence: 'It is feasible that complex interactions within the EC underlie this observation' opens up for another interpretation, namely that what the authors define as the anterior-medial portion of EC is not MEC but might be LEC and that the resulting signal correlations to not indicate real anatomical connectivity.

Though I credit the authors that they clearly defined the potential methodological difficulties, in parts like the above, they might critically reappraise their writing, as to avoid possible confusion in the readership.

Figure 1: the L-M axis is the wrong way around, flip L and M.

Figure 2: add in the figure text indicating that left is subiculum, right is CA1 and explain the numbers so indicate the proximal-distal axis in the two 3D figures 1 is distal in sub but proximal in CA1. Without such indications the figure is incomprehensible for non-anatomical experts and one needs to filter through the methods to get that. Do the same in figure 4.

In both figures 2 and 4 the overlap of various colors results in some vague brown coloring that is hard to interpret. Please indicate which inputs result in this color pattern.

*Reviewer #3 (Recommendations for the authors):*

– There seems to be no main effect of activations to object stimuli (difference to zero) in any region. Nevertheless the authors claim that regions show processing of object-related information. This is not warranted based on the results. For example in the abstract: 'The regions of another route, that connects the anterior-lateral EC and a newly identified retrosplenial-based anterior-medial EC subregion with the CA1/subiculum border, process object and scene information similarly'. If there is no evidence of any level of processing, it is not warranted to claim 'similar processing'. Based on the introduction, a double dissociation between object and scene processing in two segregated sets of regions is expected, but the authors don't provide evidence for this. In my view the interaction effect (and post-hoc tests) for higher scene processing in segment 1and2 in Figure 4 is a very interesting and meaningful finding. However, neither main effects nor interactions are presented to support claims of any level of 'object processing'. How can we be sure that we are not looking at pure noise in the object condition? Absence of evidence is not equal to evidence of absence (of different object/ scene processing).

– The introduction left me wondering which research questions are actually being pursued. In general the introduction could be streamlined, as it seemed to contain apparently irrelevant information. Most importantly, the research question(s) need to be described much more clearly. The closest thing to a research question was written on page 4 line 17-18: '..the hypothesis from rodent research that information continues to flow in a segregated manner along the transversal human entorhinal-hippocampal axis.' I take issue with a couple of things here.

1) No 'information flow' is actually being assessed in this study. In my view, this would require a directed connectivity analysis such as dynamic causal modeling or transfer entropy. The research question or hypotheses should pertain to what is actually being done. For example, 'we predict functional connectivity between X-Y to reflect the structural connectivity described in rodents/ humans previously', and 'we expect specific sensitivity to scene stimuli in distal subiculum, because of connectivity to the MEC region in rodents'. This would be 'consistent with information flow' between those regions, but it is not directly or conclusively showing it.

2) More importantly, I am afraid that the central premise of the 'two segregated memory streams' is by now somewhat outdated, certainly at the level of the entorhinal cortex and hippocampus (see Doan et al. 2019). Regarding the rodent homologue of the parahippocampal cortex (referred to as POR, or postrhinal cortex) Doan et al. write: 'Postrhinal cortex preferably targets lateral instead of medial entorhinal cortex', and that '..dorsolateral parts of LEC receive inputs from both POR and PER and that both these projections show very similar topological features..'

Nevertheless, in the present study the authors claim that 'The lateral EC preferentially communicates with the perirhinal cortex' (page 3, line 20). In the same paragraph they write that context processing should be specific to the MEC, in contrast to the LEC that would rather process non-spatial and item information. Strikingly, they then go on to cite Doan et al. 2019 and effectively contradict themselves within the same paragraph '…parahippocampal cortex communicates with the EC along its full extend' (as Doan et al. write, the LEC is similarly connected to POR (PHC) and PER). This means that contextual information from PHC should not uniquely be passed on to the MEC. On the contrary, it should even be more strongly passed on to the LEC than the MEC.

Nilssen et al. 2019 (whom the authors also cite) discuss the multimodal inputs to the LEC, including from the parahippocampal cortex (POR in rodents). They write 'In line with this shared input, we propose that it is the PER/LEC interface that provides the optimal substrate to detect changes in the context.'

I am rather puzzled why the authors cite Nilssen et al. and Doan et al. and still uphold the segregated stream hypothesis of entorhinal and hippocampal subregions. In my reading, those studies are entirely incompatible with the view of two parallel, segregated memory streams involving the LEC and MEC. Ironically, the null finding of category specific processing in the lateral cluster of EC voxels, and associated regions, might be due to the convergent, multimodal input patterns. Unfortunately, the two segregated stream hypothesis seems to be a central tenet in the current study. But it is entirely possible that I am overemphasizing or misunderstanding the premise of the study, in which case a critical rewriting could clear out the issues pointed out above.

– The approach to segment the EC into sub-clusters based on known connectivity to other regions seems fine in general. However, this should be guided by gold-standard, a priori knowledge of entorhinal subregions such as the rodent MEC and LEC (or finer cyto/ myeloarchitectonic subdivisions, for example described in Kimer et al. 1997). In the present study, the authors select four cortical seed regions to segment entorhinal subregions based on functional connectivity, without providing appropriate justification how some of those regions would uniquely connect to specific, cytoarchitectonically-defined entorhinal subregions. For example, BA35, BA36, and PHC all project to the rodent LEC, and PHC also project (albeit weaker) to the MEC. Which a-priori defined entorhinal subregions should be uniquely identified with these four seed regions? Following the semipartial correlation analyses, clusters of EC voxels are then segmented and labeled posterior-medial, posterior-lateral, anterior-medial and anterior-lateral EC. Those names are then used quasi synonymously with coherent and established, cytoarchitectonically defined subregions such as the rodent LEC and MEC. This seems egregious, not only because of the questionable choice of initial seed regions, but more so because of the discontinuous topography of the segmentations shown in Figure 1. What is the ground-truth (cytoarchitecture, myeloarchitecture, tracing studies, gene expression) evidence for a salt-and-pepper organization of entorhinal subregions? The clusters can't correspond to coherent cytoarchitectonic regions, so why would they be referred to as such. The BA36-related cluster seems to be all over the place. Why is it referred to as 'posterior-lateral EC'?. The discontinuities are arguably the result of sources of noise and error in fMRI. I am not saying that this general approach is necessarily useless, but the naming of the clusters should not be misleading. E.g. they could be referred to as 'RSC-connected voxels in EC' etc., or their predominant anatomical location could be used, but after actual quantification of their predominant location and including the specification that it is a discontinuous cluster of voxels – not a coherent subregion that has cytoarchitectonic counterpart. Why did the authors not simply use the ROIs from their previous identification of the human homologues of the MEC and LEC (Maass et al. 2015) to address their current research questions? In summary, this could be reconciled by a different, consistently used naming scheme for the entorhinal seeds that avoids confusion with cytoarchitectonic subregions.

---

## [Author Response]

Essential revisions:introduction/terminology:1. There are many different terms used interchangeably throughout the manuscript and this is confusing since it is not immediately apparent whether they cover similar notions. Throughout the paper the phrase transversal axis is used differently in many different places; usage varies from in the abstract: 'cortical – entorhinal interaction and the circuitry's inner communication along the transversal axis'; in the introduction: 'from the EC towards the transversal hippocampal axis', or 'information continues to flow in a segregated manner along the transversal human entorhinal-hippocampal axis'; in the beginning of the discussion, page 14 line 9: the 'transversal entorhinal-hippocampal axis' and there are many more. It seems that you really refer to the transverse/transversal axis of CA1/subiculum so define that, give it a name or an abbreviation if you want, and use that consistently.

We thank you for that helpful observation and we have changed our wording towards “transversal sub/CA1 axis” throughout the entire manuscript.

We introduce the name on page 3, line 7:

“…the transversal axis of hippocampal subiculum and CA1 (here referred to as transversal sub/CA1 axis).”

2. On page 6 line 15 you state/define that you are working with two parallel input streams traditionally associated with information on object ("item") versus scene ("contextual"). In the discussion on page 14 you start with a heading that carries 'contextual information. In the following text you talk about scene representation, which is ok if one remembers that you have defined the two as synonyms, but it becomes confusing when you suddenly jump to spatial or contextual information (line 31). So, I suggest that again you start with defining the terms you want to use and use them throughout; scene=context=space and object = item.

Indeed, we see and agree that it eases readability if we stick to one set of terms. We appreciate your comment and decided to use “object versus scene information” throughout. This wording remains closest to our operationalization. Nevertheless, we mention that other terms have been used to define these representations. See therefore the footnote in the introduction on page 3, line 11:

“^1^ In light of confusing nomenclature, here we adhere to scene and object information (elsewhere referred to as contextual, spatial or ‘Where’ and content, non-spatial, item or ‘What’ information, respectively).”

3. In general, the Introduction could be streamlined and the research question(s) described more clearly. The authors should incorporate hypotheses/predictions in the Introduction given that numerous theoretical models make very specific predictions regarding the neuroanatomical underpinnings of scene/object processing (i.e., see Dalton and Maguire, 2017. Curr Opin Behav Sci. 17: 34-40 for an example relating to human hippocampal subfields).

We appreciate your helpful advice. To streamline the introduction better (and likewise incorporate comment #15 #17) we deleted the rather historical overview on the former parallel mapping hypothesis and point earlier to the problem at hand.

See therefore page 3, line 10:

“Large-scale cortical information streams, that originate in the visual ‘Where’ and ‘What’ pathways and process scene and object information (Berron et al., 2018; Haxby et al., 1991; Ranganath and Ritchey, 2012; Ritchey et al., 2015; Ungerleider and Haxby, 1994), map onto the EC in a complex manner and define functional EC subregions. Recent rodent research updates the former conception of a parallel mapping of scene and object information via parahippocampal and perirhinal cortices onto medial versus lateral EC subregions (cf. posterior-medial versus anterior-lateral EC subregions as the human homologues; Maass, Berron et al., 2015; Navarro Schröder et al., 2015). Instead of a strict parallel mapping, profound cross-projections exist from the parahippocampal cortex towards the perirhinal cortex and the lateral EC (Nilssen et al., 2019). In accordance, information seems to converge in the rodent lateral EC (Doan et al., 2019). The update, thus, implies a more complex functional organization than parallel scene and object information mapping. Moreover, this advance highlights the retrosplenial cortex as an additional source to convey information directly from the cortical scene processing stream onto the EC. The retrosplenial cortex projects to the medial EC and, like the parahippocampal cortex, is part of the scene processing stream (e.g. involved in scene translation; Vann et al., 2009; Nilssen et al., 2019; Witter et al., 2017). The update, furthermore, evokes the question how cortical sources of information uniquely map onto the EC and which kind of information is processed in the resulting functional EC subregions.

Within the entorhinal-hippocampal circuitry, an important direct way of communication exists between the EC and hippocampal subiculum and CA1. How functional EC subregions communicate towards the transversal sub/CA1 axis in humans is, however, unclear. Similarly, the extent to which specific scene and object information processing routes might emerge, despite information convergence in the EC, is unknown.”

In addition, we directly point out that evidence on transversal information processing in the hippocampal subiculum and CA1 is mixed. We here also incorporated the work by e.g. Dalton and Maguire (2017). See page 3, line 32:

“On one hand, rodent research indicates a transversal organization where scene and object information is processed along two anatomically wired routes, the medial EC – distal subiculum – proximal CA1 route and the lateral EC – proximal subiculum – distal CA1 route, respectively (Witter et al., 2017; note sparse functional evidence in the subiculum: Ku et al., 2017; Cembrowski et al., 2018; but frequent reports in the rodent CA1 region: Henriksen et al., 2010; Nakamura et al., 2013; Igarashi et al., 2014; Nakazawa et al., 2016; Beer et al., 2018). Initial functional and structural connectivity data also indicate such a transversal connectivity profile in humans (Maass, Berron et al., 2015; Syversen et al., 2021). In accordance, scene information seems to be preferentially processed in the distal subiculum (Dalton et al., 2018; Dalton and Maguire, 2017; Zeidman et al., 2015) and hints exist for preferential object processing at the subiculum/CA1 border (Dalton et al., 2018). On the other hand, anatomical projections in the monkey show a longitudinal profile on top of the transversal profile with mainly the anterior-lateral and posterior-lateral entorhinal portions projecting to the distal subiculum – proximal CA1 and proximal subiculum – distal CA1, respectively (Witter and Amaral, 2020). According to information convergence in the EC, a recent report finds convergence along the rodent transversal CA1 axis (Vandrey et al., 2021). In humans, visual stream projections towards the entorhinal-hippocampal circuitry similarly suggest convergence of scene and object information in the subiculum/CA1 border region but preserved scene processing in the distal subiculum (Dalton and Maguire, 2017). A detailed examination of the latter hypothesis is, however, lacking. The diversity of findings emerging from the literature calls for a thorough investigation to elucidate whether multiple transversal processing routes exist within the human entorhinal-hippocampal circuitry.”

We also make the specific problem more apparent and clearly state which questions we aim to answer.

See therefore page 4, line 20:

“To summarize, our conception of how information travels towards the entorhinal-hippocampal circuitry underwent key changes which warrant an extensive exploration of the circuitry’s functional organization. First, rodent research shows that there is no strict parallel mapping of cortical information from the perirhinal and parahippocampal cortex towards the EC. Second, information seems to converge already before the hippocampus.”

Page 4, line 35:

“With a combination of functional connectivity and information processing analyses, we seek to answer two sets of questions. Regarding functional connectivity, we ask where the parahippocampal, perirhinal and retrosplenial cortical sources uniquely map onto the human EC and how these functionally connected routes continue between EC subregions and the transversal sub/CA1 axis. Regarding information processing, we ask whether and where scene and object information are specifically processed in the EC and along the transversal sub/CA1 axis.”

We, moreover, clearly state our hypotheses in the introduction on page 5, line 5:

“We test the hypotheses of (1) a transversal functional connectivity pattern and (2) multiple information processing routes within the entorhinal-hippocampal circuitry. Thus, following the updated conception of a non-parallel cortical scene and object information mapping onto the EC in rodents, we will show how cortical information streams map onto the EC in humans. This mapping will then be our detailed starting point to investigate the functional connectivity and information processing within the entorhinal-hippocampal circuitry.”

To keep the introduction concise, we specifically outline our predictions based on previous literature at the respective places in the results. See page 7, line 13:

“Following the characterization of entorhinal seeds, we focused on the functional connectivity between these entorhinal subregions and hippocampal subiculum and CA1 to test the hypothesis of a transversal functional connectivity pattern. We predicted that while some EC subregions have a preference to functionally connect with the subiculum/CA1 border, others preferentially connect with the distal subiculum and proximal CA1. In the previous step we identified EC subregions based on unique cortical source contributions. Therefore, our predictions remained in accordance with Maass, Berron et al. (2015): We expected that the EC subregion preferentially connected with the parahippocampal cortex (EC_PHC-based_ seed) maps towards the distal subiculum and EC subregions connected with the perirhinal cortex (EC_Area35-based_ seed, EC_Area36-based_ seed) map towards the proximal subiculum, a mapping that we predicted to be extended towards the distal CA1.”

Page 10, line 4:

“Besides the intrinsic functional connectivity patterns within the entorhinal-hippocampal circuitry, we also examined the characteristics of scene and object information processing to test the hypothesis of multiple information processing routes within the entorhinal-hippocampal circuitry. We predicted a route of specific scene processing and another route of convergent information processing. Following the proposal by Dalton and Maguire (2017) and the updated cross-projections from the scene to the object information processing stream (Nilssen et al., 2019), we expected scene processing in the distal subiculum. The updated parahippocampal cross-projections imply convergence wherever specific object processing had been expected previously. Thus, we explored whether any entorhinal-hippocampal subregions still process object information specifically. However, we largely expected to find evidence consistent with convergent processing of scene and object information within the entorhinal-hippocampal circuitry.”

Finally, we excluded the reference to potential clinical implications from the introduction and moved this aspect entirely to the discussion.

4. On page 18, lines 15 to 20, there follows a paragraph that is so dense of information, that the two concepts that I think the authors are trying to convey both might get lost. You mention 'contextual features of an item' that might be the result of convergence taking place in the anterior portion of EC, i.e. outside the hippocampus, and you contrast that with a pure contextual loop that allows for subsequent converge (with what) in the hippocampus. That second loop is associated with the distal subiculum, whereas the first is actually associated with the proximal subiculum/distal CA1 border. You might want to emphasize that more deliberately. Moreover, what is the evidence that this assumed convergence in EC actually occurs, i.e. what is the evidence that anterior-lateral EC and anterior-medial EC communicate with one another; actually available data in rodents and monkeys argue against that and favor a predominant interaction along the AP-axis. Note that the summary in the abstract adds to the confusion referred to at the second bullet: you mention that the anterior-medial EC, defined through the RSC source processing scene information, whereas the second loop processes only contextual information, implying that scene and context are two different things.

Thank you for this thoughtful comment. We realize that the paragraph is not only dense but that our wording was also misleading. Indeed, we are not aware of explicit evidence that anterior-lateral EC and anterior-medial EC communicate with each other. What we observe, however, is that functionally adjacent cortical regions do not show convergence but specific object versus scene processing (perirhinal and retrosplenial source regions, respectively). As we interpret our functional activity pattern in the two anterior EC seeds EC_Area35-based_ and EC_RSC-based_ as consistent with convergence, we thus conclude that convergence happens before the hippocampus – and it may occur in the anterior EC. Future research can obtain the exciting investigation of how and where convergence happens.

We hope that the following changes to that paragraph make our point clearer on page 17, line 11:

“Dalton and Maguire (2017), however, made a relevant proposal based on visual processing pathways and information processing. In correspondence to our results, they proposed the subiculum/CA1 border as a point of convergence between scene and object information processing streams. While their conclusion was based on direct parahippocampal, retrosplenial and perirhinal connections to the hippocampus, we found that both, the EC_Area35-based_ (that is connected with the cortical object processing stream) and the EC_RSC-based_ (that is connected with the cortical scene processing stream) show connectivity with the subiculum/CA1 border (see also appendix V for information processing in cortical source regions). Convergence is potentially also achieved via recurrency within the entorhinal-hippocampal system and cortical regions (cf. Koster et al., 2018 for evidence on recurrency). These considerations are an exciting future research avenue and remain speculative based on the current data due to insufficient temporal resolution. We nevertheless hypothesize the existence of two processing routes: one that processes converged object and scene information and one that processes scene information specifically. Thus, scene and object information processing might converge before the hippocampus. This presumably occurs within the anterior EC, given object-specific and scene-specific processing take place in the cortical source regions of the EC_Area35-based_ and EC_RSC-based_ subregions, respectively (see appendix V). Here, objects may be bound together with their defining scene-like or contextual features (akin to the “object-in-location” idea in Connor and Knierim, 2017; Knierim et al., 2014). In addition, the dedicated scene processing that we observe along the EC_PHC-based_ – distal subiculum route, may functionally underpin ideas about an anatomically graded contextual scaffold that the hippocampus utilizes to incorporate detailed information from the object-in-scene route into meaningful chunks of cohesive memory representations ("events"; Behrens et al., 2018; Clewett et al., 2019; Robin, 2018; Robin and Olsen, 2019).”

To avoid confusion, we moreover changed the wording in the abstract on page 2, line 9:

“Our data show specific scene processing in the functionally connected EC_PHC-based_ and distal subiculum. Another route, that functionally connects the EC_Area35-based_ and a newly identified EC_RSC-based_ with the subiculum/CA1 border, however, shows no selectivity between object and scene conditions. Our results are consistent with transversal information-specific pathways in the human entorhinal-hippocampal circuitry, with anatomically organized convergence of cortical processing streams and a unique route for scene information.”

Methods:5. Considering a major strength of the paper is the detail with which anatomical ROIs were created, I'm curious to know how the authors accounted for variability in MTL anatomy across participants on steps that required warping of ROIs into a group space or, in the case of the RSC mask, warping from group template space to individual participant space. Importantly, how were the ROIs assessed for their accuracy of alignment on the EPI images?

Thank you for raising that important point. To assess accurate warping, we performed careful manual checks on the co-registered and warped single-subject data based on landmark overlays. To illustrate the process, we now incorporated Figure 1 in Appendix 7. The figure shows the stepwise co-registration process between group and individual participant space. It provides examples of individual region of interest alignment.

However, anatomical variability is particularly challenging for the segmentation of the perirhinal cortex due to variable sulcal patterns and the variability in the depth of the collateral sulcus (see e.g. Berron, Vieweg et al., 2017, Ding and van Hoesen, 2010). Thus, all analyses including perirhinal Area 35 and Area 36 were performed in individual participant space. Within the hippocampus, the hippocampal body is mostly consistent across individuals with respect to major anatomical landmarks (we excluded the hippocampal head and tail from analyses).

In addition, please note that we utilized a study-specific group template. This decreases the amount of variability incorporated into the group space. For an appropriate study-specific template, good results can be achieved in averaging structural images when the sample size exceeds ten participants.

We refer to the appendix figure in the main text on page 24, line 20 and line 32.

Please see Appendix 7 – Figure 1 and figure caption:

6. Related to this, no quality assurance measures have been presented for the manually created ROIs. More information relating to these should be included in the methods section. For example, could the authors describe how many people were involved in manually segmenting the ROI's? If only one person, could the authors provide the results of intra-rater reliability analyses using, for example, the DICE overlap metric for each ROI? If more than one person was involved, could the authors also provide the results of inter-rater reliability analyses? In addition, considering it can sometimes be difficult to warp small ROIs such as hippocampal subfields from structural to EPI images with sufficient accuracy, could the authors provide some visual representations (perhaps as a supplemental figure) showing, for example, that the subiculum and CA1 ROIs created on the structural images were well aligned with the subiculum and CA1 on EPI images.

We appreciate this important request. Indeed, the regions have been segmented by two experienced raters. As the regions-of-interest (ROIs) have been part of a previously published study (Berron,Vieweg et al., 2017) for which the authors evaluated quality assurance measures, we report these as follows in appendix VIII now:

“The individual regions of interest were segmented by the same two experienced raters that also segmented a subsample of our data (24 hemispheres of 22 participants) for a previous publication (Berron, Vieweg et al., 2017). Quality assurance measures were calculated for that subsample. Regarding intra-rater reliability, the dice similarity coefficients are above 0.88 for all segmented regions (region-specific means (SD) are as follows: PHC 0.93 (0.03); Area 36 0.91 (0.02); Area 35 0.88 (0.02); EC 0.91 (0.01)). The intraclass-correlation coefficients for intra-rater reliability are all above 0.95 (PHC 0.99; Area 36 0.96; Area 35 0.97; EC 0.98). For the inter-rater reliability, dice similarity coefficients are above 0.84 for all segmented regions (region-specific means (SD) are as follows: PHC 0.86 (0.12); Area 36 0.91 (0.02); Area 35 0.84 (0.05); EC 0.87 (0.02)). The intraclass-correlation coefficients for inter-rater reliability are all above 0.78 (PHC 0.94; Area 36 0.88; Area 35 0.87; EC 0.94; see Berron, Vieweg et al., 2017).”

Please note that we falsely reported CA1 and subiculum segmentation in individual space and removed this now from page 24, line 13 in the methods section.

To assure accurate co-registration and warping, careful manual assessment based on anatomical landmarks was performed. To illustrate the procedure, we now include Figure 1 in appendix VII. This figure shows not only the coregistration process but also illustrates successful coregistration with images from an example participant. Note that this figure illustrates the coregistration process for Area35 and RSC. We applied the same resulting warping matrices to all other subregions of interest respectively.

7. For ROIs whose borders lie immediately adjacent each other (e.g., medial EC and distal subiculum; anterior PHC and posterior EC), did the authors take steps to reduce the possibility of fMRI signal 'bleeding' between ROIs by, for example, ensuring the distance between borders of each ROI was greater than the spatial smoothing kernel used on the functional data? If so, could you please describe these steps in the methods section?

Thank you for raising attention to this point. We, indeed, cannot rule out that signal from adjacent regions leaks into the regions of interest. This is yet another reason for why we stress the importance of future studies employing different methods to solidify our findings even more. See page 20, line 21:

“Future research is needed to evaluate how the functionally derived entorhinal seeds in this study relate to histologically derived entorhinal subregions (Oltmer et al., 2022) or entorhinal subregions based on structural connectivity (Syversen et al., 2021). For a dedicated comparison of subregions, it is essential to pay close attention to the segmentation of the EC itself.”

Page 20, line 14:

“In combination with closely matched histological or structural magnetic resonance imaging data, future work can further reveal the nature of retrosplenial mapping on the human EC.”

To diminish the influence of neighbouring regions on signal in target regions, we used a smoothing kernel smaller than two times the voxel size. Most seed and target regions are further than a voxel apart from each other. We established that functional connectivity analyses with source and seed regions in the contralateral hemisphere, provided roughly comparable results to analyses with ipsilateral sources and seeds performed across spatially proximal voxels (see especially the extracted estimates from individual participants and statistical effects in Author response images 1- 4).

We address this aspect as follows in the limitations section of the discussion in the manuscript on page 19, line 10:

“Second, while it is unlikely that our functional connectivity pattern is the result of spatial proximity, increased correlation between spatially adjacent regions is an inherent problem of functional connectivity analyses. Distances between seed and target regions differ and may determine patterns in the functional connectivity data. To diminish the influence of proximity, our smoothing kernel was smaller than two times the voxel size. It is important to stress moreover, that the pattern of our results is not easily explainable by spatial distance between seed and target regions. The EC_Area35-based_ or EC_RSC-based_, for instance, are not adjacent to the subiculum/CA1 border. Furthermore, we observed roughly comparable results for neighboring seeds and targets (e.g. EC_PHC-based_ and distal subiculum) when we performed the functional connectivity analyses with seed and source regions in the contralateral hemisphere.”

**Author response image 1. sa2fig1:** Functional connectivity preferences to contralateral entorhinal seeds along the transversal axis of subiculum and CA1. Displayed are the results of a seed-to-voxel functional connectivity analysis between the displayed right entorhinal seeds and the left subiculum and CA1 subregion. The 3D figure displays voxel-wise connectivity preferences to the entorhinal seeds (color coded to refer to the respective entorhinal seed [E]) on group level ([A] – subiculum; [B] – CA1). To display mean connectivity preferences across participants along the transversal sub/CA1 axis, β estimates were extracted and averaged from equally sized segments from proximal to distal ends (five segments in subiculum [A], three segments in CA1 [B]; schematized in white on the 3D figures) on each coronal slice and averaged along the longitudinal axis. Repeated measures ANOVAs revealed significant differences in connectivity estimates along the transversal axis of the subiculum ([C]; overall effect of seed by transversal segment: F(12,372) = 4.554; p <.001; main transversal effect of seed EC_Area35-based_: F(4,124) = 5.856, p_FDR_ = .0012; EC_PHC-based_: F(4,124) = 3.147; p_FDR_ = .037; EC_RSC-based_: F(4,124) = 7.828, p_FDR_ <.001; EC_Area36-based_: F(4,124) = 0.856, p_FDR_ = 0.493) with interaction effects (between EC_PHC-based_ and EC_RSC-based_: F(4,124) = 9.249, p_FDR_ <.001; between EC_Area35-based_ and EC_PHC-based_: F(4,124) = 5.923, p_FDR_ <.001; between EC_Area35-based_ and EC_RSC-based_: p = 0.051 (uncorrected)) and in CA1 ([D] main seed by transversal effect: F(6,186) = 2.722, p = .034; main transversal effect of seed EC_RSC-based_: F(2,62) = 13.782; p_FDR_ <.001; EC_Area35-based_: F(2,62) = .221; p_FDR_ = .821; EC_Area36-based_: F(2,62) = 3.598, p_FDR_ = .069; EC_PHC-based_: F(2,62) = .198, p_FDR_ = .821). Displayed significances obtained by FDR-corrected post-hoc tests and refer to p <.05. Shaded areas in the graphs refer to standard errors of the mean. EC – entorhinal; M – medial; L – lateral; A – anterior; P – posterior; prox – proximal; dist – distal.

**Author response image 2. sa2fig2:** Functional connectivity preferences to contralateral entorhinal seeds along the transversal axis of subiculum and CA1. Displayed are the results of a seed-to-voxel functional connectivity analysis between the displayed left entorhinal seeds and the right subiculum and CA1 subregion. The 3D figure displays voxel-wise connectivity preferences to the entorhinal seeds (color coded to refer to the respective entorhinal seed [E]) on group level ([A] – subiculum; [B] – CA1). To display mean connectivity preferences across participants along the transversal sub/CA1 axis, β estimates were extracted and averaged from equally sized segments from proximal to distal ends (five segments in subiculum [A], three segments in CA1 [B]; schematized in white on the 3D figures) on each coronal slice and averaged along the longitudinal axis. Repeated measures ANOVAs revealed significant differences in connectivity estimates along the transversal axis of the subiculum ([C]; overall effect of seed by transversal segment: F(12,372) = 6.273; p <.001; main transversal effect of seed EC_PHC-based_: F(4,124) = 16.660; p_FDR_ <.001; EC_RSC-based_: F(4,124) = 3.543, p_FDR_ = .018; EC_Area35-based_: F(4,124) = 2.374, p_FDR_ = .086; EC_Area36-based_: F(4,124) = 1.436, p_FDR_ = .226) but not in CA1 ([D] main seed by transversal effect: F(6,186) = 1.812, p = .145). Displayed significances obtained by FDR-corrected post-hoc tests and refer to p <.05. Shaded areas in the graphs refer to standard errors of the mean. EC – entorhinal; M – medial; L – lateral; A – anterior; P – posterior; prox – proximal; dist – distal.

**Author response image 3. sa2fig3:** Entorhinal seed regions based on connectivity preferences to contralateral cortical regions. Displayed is the left EC as a 3D image with colored subregions. The subregions have been identified based on a source-to-voxel functional connectivity analysis and resulting connectivity preference to either the right retrosplenial (RSC, green) cortex, parahippocampal cortex (PHC, blue), Area 36 (A36, purple) or Area 35 (A35, pink) sources. Subregions have been determined based on the thresholded (T > 3.1) maximum voxels across four one-sample t-tests at group level, one per source. M – medial; L – lateral; A – anterior; P – posterior.

**Author response image 4. sa2fig4:** Entorhinal seed regions based on connectivity preferences to contralateral cortical regions. Displayed is the right EC as a 3D image with colored subregions. The subregions have been identified based on a source-to-voxel functional connectivity analysis and resulting connectivity preference to either the left retrosplenial (RSC, green) cortex, parahippocampal cortex (PHC, blue), Area 36 (A36, purple) or Area 35 (A35, pink) sources. Subregions have been determined based on the thresholded (T > 3.1) maximum voxels across four one-sample t-tests at group level, one per source. M – medial; L – lateral; A – anterior; P – posterior.

8. On page 20, lines 29-30, the authors describe the anatomical landmark used to create the anterior most slice of the hippocampal subfield masks but do not describe the landmark used to demarcate the posterior most extent of the masks. This should also be described.

Thank you for pointing out that missing information. We added the following specification on page 23, line 30:

“The last segmented slice was the one at which both, the inferior and superior colliculi had completely disappeared, applied for each hemisphere separately.”

9. The authors state that the subiculum and CA1 subfields were sagittally cut into "equally wide" segments (page 20; line 32) but no metrics relating to this are provided. Could the authors provide some metrics to support this? For example, was the mean number of functional voxels contained within each segment equivalent or did these substantially differ?

We appreciate your comment to provide more details here. The way we created the segments has been described in the methods section on page 23, line 32:

“Moreover, to evaluate results across the transversal sub/CA1 axis, the subiculum masks in each hemisphere were cut in five equally wide segments from medial to lateral within each coronal image. As the CA1 region gets more and more tilted towards the hippocampal tail, the three transversal CA1 segments were determined based on manual segmentation following a geometrical rule. Therefore, the two outer borders along the transversal axis of CA1 were connected with a line. From the middle point of that line, two straight lines were drawn in a 60° angle to determine roughly equally sized transversal CA1 segments within each coronal slice and hemisphere (a figure displaying the cuts, the procedure for the CA1 segments and the numbers of voxels within each segment can be found in appendix IX).”

Please note that the segments have been created in the T1 group template space. To illustrate, we now provide an illustration of the transversal subiculum and CA1 segments and the procedure for CA1 cuts, exemplified on the right hemisphere in the appendix. In addition, we provide the number of voxels within each segment and the following summary in appendix IX:

“Transversal subiculum and CA1 segments were cut on the group template T1 images. The average number of voxels contained in each subiculum segment was 460.8 voxels for the left subiculum (standard deviation 104.36) and 458 voxels for the right subiculum (standard deviation 75.09). For the left CA1 the average equals 360 voxels (standard deviation 27.58) and 335 voxels for the right CA1 segments (standard deviation 3.56, see Appendix 9 -Table 1 for segment-specific values and Figure 1 for an illustration).”

Results:10. Although this is described in the methods section, perhaps the authors could briefly mention early in the Results section that the fMRI dataset used in Analysis 1 and 2 are derived from a task-based dataset where task-related effects have been regressed out to create a "rest-like" dataset. It may also be helpful to readers interested in (or cautious of) this particular method, to cite previous work showing that these "rest-like" datasets yield similar results to those obtained from resting-state data (e.g., Gavrilescu et al., Hum Brain Mapp 29:1040-1052, 2008 or similar).

Thank you for pointing out that we need to be more specific here. We now added the following lines to the manuscript on page 5 line 27 and on page 25, line 15:

“All functional connectivity analyses were performed on the dataset where task-related effects have been regressed out before, creating a dataset that resembles resting-state data (Gavrilescu et al., 2008; Maass, Berron et al., 2015).”

“Both functional connectivity analyses were performed on residuals of task-related functional data, creating a dataset that resembles resting-state data (Gavrilescu et al., 2008; Maass, Berron et al., 2015).”

11. Functional subregions of the EC associated with RSC, PHC, A35 and A36 are displayed very clearly in Figure 1. It appears that the EC seed relating to A36 is less 'localised' than the other seeds and displays roughly equal size clusters in both the posterior lateral and anterior medial EC. However, in the text, the authors suggest that A36 shows preferential connectivity with the posterior lateral EC (page 7; lines 7-8). How did the authors come to preferentially associate A36 with the posterior lateral EC rather than with the anterior medial EC? Did the authors conduct any additional quantitative assessments to determine this?

We appreciate this thoughtful comment. We based our conclusion on the following, additional metric: We evaluated for each source in which entorhinal part it yields the highest number of preferentially connected voxels.

Therefore, we cut the EC into quadrants (see Appendix 2 – Figure 1).

We then counted the number of voxels that have been assigned to be preferentially connected to each cortical source within each quadrant.

When we assign each source to the quadrant in which the respective source yields most functionally connected voxels we find the following organization, averaged across hemispheres:

‘anterior-medial’ (EC_RSC-based_) – quadrant number 1

‘posterior-medial’ (EC_PHC-based_) – quadrant number 2

‘anterior-lateral’ (EC_A35-based_) – quadrant number 3

‘posterior-lateral’ (EC_A36-based_) – quadrant number 4

However, we understand the concern that these functional connectivity clusters in the EC are not all locally constrained. Given reviewer comments number #10 and #20 and our methodology’s lack of anatomical precision, we now decided to name the entorhinal seed regions in a non-biased manner that does not reflect their anatomical position.

In accordance, we changed the naming of the EC seed regions throughout the entire manuscript into: EC_RSC-based_, EC_PHC-based_, EC_Area35-based_ and EC_Area36-based_. Future studies with a stronger anatomical focus can then determine the exact location of entorhinal subregions.

To clarify these aspects, we added the following sentences to the manuscript on page 6, line 23:

For the EC_PHC-based_ seed, the majority of voxels can roughly be described as clustering in the posterior-medial entorhinal portion, for the EC_RSC-based_ seed in the anterior-medial portion, for the EC_Area35-based_ seed in the anterior-lateral portion and for the EC_Area36-based_ seed in the posterior-lateral entorhinal portion (see appendix II for exact voxel counts). Note that both perirhinal-based entorhinal seeds extended along the anterior to posterior axis such that the EC_Area35-based_ progressed more along the outer EC (i.e. laterally, with a main focus anteriorly) and the EC_Area36-based_ along the inner EC (i.e. medially, with a main focus posteriorly, see Figure 1 and the medial reflection of the EC seeds). It is important to note that these are rough qualitative descriptions of the main clusters, without quantification or an established relationship to coherent cytoarchitectonic regions. We will therefore continue to refer to them as EC_RSC-based_, EC_PHC-based_, EC_Area35-based_ and EC_Area36-based_ seeds.”

Please also note that the EC_Area36-based_ focus in the posterior entorhinal portion, particularly in the left hemisphere, becomes visible when looking at the EC from a different angle. To point this out better, we therefore included another viewpoint in our 3D figure and mirror the EC.

We include the quadrant-based counting in appendix II. The supplement reads as follows:

“To assess the main location of each cortical source preferences within the EC, we cut the left and right EC in four quadrants. This was performed in T1 template space. First, the middle slice of all coronal slices that capture the EC was determined separately for each hemisphere. This slice was used to cut the EC in quadrants I, III and II, IV. Second, the middle slice of all axial slices that capture the EC was determined. This slice served to cut the EC in quadrants I, II and III, IV (see Appendix 2 – Figure 1). Note, to determine the most superior axial slice, the most posterior coronal level of the EC was used. Subsequently, we counted the number of voxels that have been assigned to each of the four cortical source regions after the initial functional connectivity analyses (that served to determined EC seeds). Averaged across hemispheres, most voxels assigned to the retrosplenial source are in EC quadrant I, most voxels assigned to the Area 35 source in EC quadrant II, most voxels assigned to the parahippocampal cortex in EC quadrant III and most voxels assigned to Area 36 in EC quadrant IV (see Appendix 2 – Table 1 for detailed voxel counts). Note that these quadrants do not refer to anatomically defined EC subregions.”

12. While I understand the primary focus of the subiculum and CA1 analyses relate to their transverse axis, it seems clear when looking at Figure 2, that there may also be interesting anterior-posterior differences along the body of these subfields. For example, functional clusters are largely absent in anterior portions of segment 3 of the subiculum but posterior portions of segment 3 contain functional clusters. Observations relating to the anterior-posterior axis are not discussed in text but may be informative to those with an interest in anterior-posterior variations in hippocampal function.

We agree that the longitudinal effects are interesting and important to the scientific community and thank you for raising that point. While we feel it would stretch the scope of a single paper too much to also thoroughly analyse the longitudinal axis statistically, we now added a qualitative description of longitudinal effects to the discussion on page 17, line 34:

“For completeness, we noted differences in functional connectivity along the longitudinal axis of the subiculum. We observed, for instance, more widespread functional connectivity of the EC_Area35-based_ in the posterior subiculum whereas functional connectivity with the EC_PHC-based_ portion seems more prominent in the anterior subiculum. The latter is consistent with previous reports (Dalton et al., 2019). The former, however, needs to be explored further by taking different segmentation protocols and seed regions into account. Note, that Maass, Berron et al. (2015) did not report longitudinal differences in connectivity strength between the EC and the subiculum. Future work needs to investigate how these observations relate to the reported gradient in functional connectivity and information resolution along the hippocampal longitudinal axis (e.g. Brunec et al., 2018 but many more).”

13. All the data are described in terms of preferred connectivity, which is what the data show. However, by mapping the preferred connectivity as absolute color maps with sharp borders, a lot of the complexities, i.e. gradients are lost. It would be great if the authors could provide supplementary maps of each source in isolation and how it maps across the surface of EC and the transverse axis of CA1/subiculum.

We fully agree that gradient maps are a very interesting supplement and provide the respective maps now in appendix III:

Interpretation:14. The authors created ROI masks for the subiculum and refer to medial portions of this mask as 'distal subiculum'. However, the mask appears to encompass the entire 'subicular complex' inclusive of the subiculum, presubiculum and parasubiculum. Importantly, the area referred to as 'distal subiculum' throughout the manuscript (specifically, segments 1 and 2 of the subiculum mask) most likely aligns with the location of the pre- and parasubiculum in addition to containing portions of the distal subiculum. This is important considering the growing body of evidence that the pre- and parasubiculum are preferentially engaged during scene-based cognition. While this is briefly touched on in the discussion (page 13; line 5-6), I feel the interpretation of results as they relate to the 'distal subiculum' could benefit from being placed more firmly in the context of this growing body of evidence relating to the involvement of medial aspects of the hippocampus (inclusive of the pre- and parasubiculum) in scene-based cognition.

Thank you for that insightful comment. We agree and have now added the following paragraph in the discussion of our results on page 13, line 28:

“Our observation is in line with the hypothesis that the distal subiculum is more involved in processing scenes than objects based on previous findings in the human brain. While the subiculum in general was associated with scene discrimination (Hodgetts et al., 2017), a growing body of evidence relates particularly the medial hippocampus to scene processing. This entails two medial areas, the pre- and parasubiculum, that we attribute to the distal subiculum in our current segmentation. Especially the area that resembles the pre- (or here: distal) subiculum has been shown to be involved in scene construction (Dalton et al., 2018, Zeidman et al., 2015).”

15. I am afraid that the central premise of the 'two segregated memory streams' is somewhat outdated, certainly at the level of the entorhinal cortex and hippocampus (see Doan et al. 2019). Regarding the rodent homologue of the parahippocampal cortex (referred to as POR, or postrhinal cortex) Doan et al. write: 'Postrhinal cortex preferably targets lateral instead of medial entorhinal cortex', and that '..dorsolateral parts of LEC receive inputs from both POR and PER and that both these projections show very similar topological features..'

We appreciate that concern. Indeed, our rather historical approach was due to our manuscript considered as advancing the Maass et al., 2015 publication. To circumvent the impression that the two parallel entorhinal-hippocampal stream hypothesis is the current state of research, we now rephrased the introduction considerably.

From the very beginning, we make clear that this hypothesis was recently updated on page 3, line 10:

“Large-scale cortical information streams, that originate in the visual ‘Where’ and ‘What’ pathways and process scene and object information (Berron et al., 2018; Haxby et al., 1991; Ranganath and Ritchey, 2012; Ritchey et al., 2015; Ungerleider and Haxby, 1994), map onto the EC in a complex manner and define functional EC subregions. Recent rodent research updates the former conception of a parallel mapping of scene and object information via parahippocampal and perirhinal cortices onto medial versus lateral EC subregions (cf. posterior-medial versus anterior-lateral EC subregions as the human homologues; Maass, Berron et al., 2015; Navarro Schröder et al., 2015). Instead of a strict parallel mapping, profound cross-projections exist from the parahippocampal cortex towards the perirhinal cortex and the lateral EC (Nilssen et al., 2019). In accordance, information seems to converge in the rodent lateral EC (Doan et al., 2019). The update, thus, implies a more complex functional organization than parallel scene and object information mapping.”

We state this again, before sketching our study on page 4, line 20:

“To summarize, our conception of how information travels towards the entorhinal-hippocampal circuitry underwent key changes which warrant an extensive exploration of the circuitry’s functional organization. First, rodent research shows that there is no strict parallel mapping of cortical information from the perirhinal and parahippocampal cortex towards the EC. Second, information seems to converge already before the hippocampus.”

16. Nevertheless, in the present study the authors claim that 'The lateral EC preferentially communicates with the perirhinal cortex' (page 3, line 20). In the same paragraph they write that context processing should be specific to the MEC, in contrast to the LEC that would rather process non-spatial and item information. Strikingly, they then go on to cite Doan et al. 2019 and effectively contradict themselves within the same paragraph '…parahippocampal cortex communicates with the EC along its full extend' (as Doan et al. write, the LEC is similarly connected to POR (PHC) and PER). This means that contextual information from PHC should not uniquely be passed on to the MEC. On the contrary, it should even be more strongly passed on to the LEC than the MEC.

Thank you for this important comment, we apologize if we raised the impression that the parallel stream connection and related information segregation are the current state of research. This overview was rather meant as a brief historical introduction. We hope that our rephrased introduction makes this clear. For example on page 3, line 17:

“Instead of a strict parallel mapping, profound cross-projections exist from the parahippocampal cortex towards the perirhinal cortex and the lateral EC (Nilssen et al., 2019). In accordance, information seems to converge in the rodent lateral EC (Doan et al., 2019). The update, thus, implies a more complex functional organization than parallel scene and object information mapping. Moreover, this advance highlights the retrosplenial cortex as an additional source to convey information directly from the cortical scene processing stream onto the EC. The retrosplenial cortex projects to the medial EC and, like the parahippocampal cortex, is part of the scene processing stream (e.g. involved in scene translation; Vann et al., 2009; Nilssen et al., 2019; Witter et al., 2017). The update, furthermore, evokes the question how cortical sources of information uniquely map onto the EC and which kind of information is processed in the resulting functional EC subregions.”

When we refer to current literature we also incorporate functional predictions based on the parahippocampal cross-projections on page 4, line 12:

“According to information convergence in the EC, a recent report finds convergence along the rodent transversal CA1 axis (Vandrey et al., 2021). In humans, visual stream projections towards the entorhinal-hippocampal circuitry similarly suggest convergence of scene and object information in the subiculum/CA1 border region but preserved scene processing in the distal subiculum (Dalton and Maguire, 2017). A detailed examination of the latter hypothesis is, however, lacking. The diversity of findings emerging from the literature calls for a thorough investigation to elucidate whether multiple transversal processing routes exist within the human entorhinal-hippocampal circuitry.”

We clearly summarize this in the introduction now on page 4, line 22:

“First, rodent research shows that there is no strict parallel mapping of cortical information from the perirhinal and parahippocampal cortex towards the EC. Second, information seems to converge already before the hippocampus.”

Likewise our predictions now state clearly that we expect convergence due to the parahippocampal cross-projections on page 7, line 13 and on page 10, line 4:

“Following the characterization of entorhinal seeds, we focused on the functional connectivity between these entorhinal subregions and hippocampal subiculum and CA1 to test the hypothesis of a transversal functional connectivity pattern. We predicted that while some EC subregions have a preference to functionally connect with the subiculum/CA1 border, others preferentially connect with the distal subiculum and proximal CA1. In the previous step we identified EC subregions based on unique cortical source contributions. Therefore, our predictions remained in accordance with Maass, Berron et al. (2015): We expected that the EC subregion preferentially connected with the parahippocampal cortex (EC_PHC-based_ seed) maps towards the distal subiculum and EC subregions connected with the perirhinal cortex (EC_Area35-based_ seed, EC_Area36-based_ seed) map towards the proximal subiculum, a mapping that we predicted to be extended towards the distal CA1.”

“Besides the intrinsic functional connectivity patterns within the entorhinal-hippocampal circuitry, we also examined the characteristics of scene and object information processing to test the hypothesis of multiple information processing routes within the entorhinal-hippocampal circuitry. We predicted a route of specific scene processing and another route of convergent information processing. Following the proposal by Dalton and Maguire (2017) and the updated cross-projections from the scene to the object information processing stream (Nilssen et al., 2019), we expected scene processing in the distal subiculum. The updated parahippocampal cross-projections imply convergence wherever specific object processing had been expected previously. Thus, we explored whether any entorhinal-hippocampal subregions still process object information specifically. However, we largely expected to find evidence consistent with convergent processing of scene and object information within the entorhinal-hippocampal circuitry.”

17. I am rather puzzled why the authors cite Nilssen et al. and Doan et al. and still uphold the segregated stream hypothesis of entorhinal and hippocampal subregions. In my reading, those studies are entirely incompatible with the view of two parallel, segregated memory streams involving the LEC and MEC. Ironically, the null finding of category specific processing in the lateral cluster of EC voxels, and associated regions, might be due to the convergent, multimodal input patterns. Unfortunately, the two segregated stream hypothesis seems to be a central tenet in the current study.

We appreciate your comment. Our intention was not to uphold an outdated hypothesis. Given that most published research so far was conducted in light of this hypothesis, we however wanted to give an overview of the resulting ideas in our introduction. This applies in particular to information processing. While we agree that both hypotheses are not aligned with each other, we still think that at least for the MEC, the segregated stream hypothesis and research in light of that provides valuable information. Moreover, future research will need to explicitly investigate why and how previous studies could confirm the segregated stream hypothesis. As our research is an advancement to a previous study conducted in the light of the segregated stream hypothesis, our aim was to provide a historical introduction to the entangled research about functional connectivity and information processing and current idea of heavy cross-projections before the entorhinal cortex.

One goal of the current study was to show what is “left” of the original segregated stream hypothesis and to focus on where unique cortical information is mapping on the entorhinal cortex.

To prevent the impression that we uphold an outdated hypothesis, we now rephrased our introduction considerably.

In the very beginning, we mention that the parallel mapping hypothesis has been updated on page 3, line 13:

“Recent rodent research updates the former conception of a parallel mapping of scene and object information via parahippocampal and perirhinal cortices onto medial versus lateral EC subregions (cf. posterior-medial versus anterior-lateral EC subregions as the human homologues; Maass, Berron et al., 2015; Navarro Schröder et al., 2015). Instead of a strict parallel mapping, profound cross-projections exist from the parahippocampal cortex towards the perirhinal cortex and the lateral EC (Nilssen et al., 2019). In accordance, information seems to converge in the rodent lateral EC (Doan et al., 2019). The update, thus, implies a more complex functional organization than parallel scene and object information mapping.”

We summarize the recent changes and their implications later on page 4, line 20:

“To summarize, our conception of how information travels towards the entorhinal-hippocampal circuitry underwent key changes which warrant an extensive exploration of the circuitry’s functional organization. First, rodent research shows that there is no strict parallel mapping of cortical information from the perirhinal and parahippocampal cortex towards the EC. Second, information seems to converge already before the hippocampus.”

Then, we state how we aim to advance the existing research that has largely been based on the parallel mapping hypothesis on page 4, line 35:

“With a combination of functional connectivity and information processing analyses, we seek to answer two sets of questions. Regarding functional connectivity, we ask where the parahippocampal, perirhinal and retrosplenial cortical sources uniquely map onto the human EC and how these functionally connected routes continue between EC subregions and the transversal sub/CA1 axis. Regarding information processing, we ask whether and where scene and object information are specifically processed in the EC and along the transversal sub/CA1 axis. We test the hypotheses of (1) a transversal functional connectivity pattern and (2) multiple information processing routes within the entorhinal-hippocampal circuitry. Thus, following the updated conception of a non-parallel cortical scene and object information mapping onto the EC in rodents, we will show how cortical information streams map onto the EC in humans. This mapping will then be our detailed starting point to investigate the functional connectivity and information processing within the entorhinal-hippocampal circuitry.”

In the predictions that we outline now in each result’s respective section, we illustrate how the update to the parallel mapping hypothesis shapes our current expectations on page 7, line 13 and on page 10, line 4:

“Following the characterization of entorhinal seeds, we focused on the functional connectivity between these entorhinal subregions and hippocampal subiculum and CA1 to test the hypothesis of a transversal functional connectivity pattern. We predicted that while some EC subregions have a preference to functionally connect with the subiculum/CA1 border, others preferentially connect with the distal subiculum and proximal CA1. In the previous step we identified EC subregions based on unique cortical source contributions. Therefore, our predictions remained in accordance with Maass, Berron et al. (2015): We expected that the EC subregion preferentially connected with the parahippocampal cortex (EC_PHC-based_ seed) maps towards the distal subiculum and EC subregions connected with the perirhinal cortex (EC_Area35-based_ seed, EC_Area36-based_ seed) map towards the proximal subiculum, a mapping that we predicted to be extended towards the distal CA1.”

“Besides the intrinsic functional connectivity patterns within the entorhinal-hippocampal circuitry, we also examined the characteristics of scene and object information processing to test the hypothesis of multiple information processing routes within the entorhinal-hippocampal circuitry. We predicted a route of specific scene processing and another route of convergent information processing. Following the proposal by Dalton and Maguire (2017) and the updated cross-projections from the scene to the object information processing stream (Nilssen et al., 2019), we expected scene processing in the distal subiculum. The updated parahippocampal cross-projections imply convergence wherever specific object processing had been expected previously. Thus, we explored whether any entorhinal-hippocampal subregions still process object information specifically. However, we largely expected to find evidence consistent with convergent processing of scene and object information within the entorhinal-hippocampal circuitry.”

18. There seems to be no main effect of activations to object stimuli (difference to zero) in any region. Nevertheless the authors claim that regions show processing of object-related information. This is not warranted based on the results. For example in the abstract: 'The regions of another route, that connects the anterior-lateral EC and a newly identified retrosplenial-based anterior-medial EC subregion with the CA1/subiculum border, process object and scene information similarly'. If there is no evidence of any level of processing, it is not warranted to claim 'similar processing'.

We appreciate this comment and understand the concern.

Out additional analysis of cortical source regions could show significantly increased activity to object stimuli in the perirhinal cortex (Area 36 and left Area 35; see the following comment #19 for a figure). In the EC, subiculum and CA1, we however do not find significantly increased activity to object stimuli anywhere. We believe that the nature of our functional data does not allow us to conclude that no (object/scene) information processing occurs there at all, instead we agree that we can only compare between different information processing conditions (object/scene). Based on this comparison we also agree that we do not see differences in information processing between object and scene conditions. We see that our wording is potentially misleading and therefore changed it to “no differences in information processing conditions”.

To elaborate on this important point further:

The difference to zero refers to a difference from the baseline condition (scrambled pictures). As functional MRI is a relative measure, at least with our methodological set up, the baseline condition may not suffice as an indicator of “zero activation” or “no level of processing”. Instead, it is a relative condition towards which we could relate both, the object and the scene condition. Technically, we did not assess whether, for example, the EC does not activate at all during the presentation of scrambled pictures (baseline). Given the setup of our study, we have no means to assess this question. It is possible that individuals engaged in some sort of functional processing, even of the scrambled pictures (e.g. they may have tried to “see” something in them). While we cannot assess this question, we can say that based on our data, the relationship between object processing and baseline compared to scene processing and baseline appears to be similar in several regions of interest.

To stress that point, we included a paragraph in the “limitations section” of the manuscript on page 19, line 24 (new sentences in bold):

“Note that as a first step towards an understanding of the system’s functional organization and to increase comparability with earlier studies, we assessed functional connectivity and information processing within the entorhinal-hippocampal circuitry with univariate methods. These allow relative comparisons between functional activity levels in different conditions. Consequently, we are neither able to assess what the EC is processing during the baseline condition, meaning the absolute level of functional activity, nor are we able to verify that information processing is similar across conditions in for example the EC_Area35-based_ seed. Univariate methods, moreover, average the signal over regions of interest. To capture hidden voxel-wise patterns of activity that scale with the processing of certain representations, future studies could examine information pathways with multivariate methods that evaluate informational content in the activity pattern of voxels instead of in an averaged manner (Kragel et al., 2018; Kriegeskorte et al., 2008). Moreover, recent methodological advances can be employed in the future that study functional connectivity based on the underlying content representations between regions (Basti et al., 2020).”

Functional activation levels from the object condition and from the scene condition are both contrasted with the baseline condition, mainly for plotting purposes. It allowed us to show object and scene processing with separate lines. We realize that our figures may potentially have been misleading in that sense and now we excluded the stippled line indicating “zero”:

19. Based on the introduction, a double dissociation between object and scene processing in two segregated sets of regions is expected, but the authors don't provide evidence for this. In my view the interaction effect (and post-hoc tests) for higher scene processing in segment 1and2 in Figure 4 is a very interesting and meaningful finding. However, neither main effects nor interactions are presented to support claims of any level of 'object processing'. How can we be sure that we are not looking at pure noise in the object condition? Absence of evidence is not equal to evidence of absence (of different object/ scene processing).

Indeed, we did not see any processing biased towards the object condition in any of our tested regions of interest. Your comment is very thoughtful and important and we agree that absence of evidence is not equal to evidence of absence. We performed an additional set of analyses regarding information processing in the cortical source regions and include it in the appendix. Here, we find evidence for significantly more object than scene processing in the bilateral perirhinal Areas 36 and the left Area 35 while in the retrosplenial and parahippocampal cortex scene processing is significantly higher than object processing. This additional finding suggests that we are not looking at pure noise in the object condition. We think it rather provides an additional hint towards multimodal object and scene processing in the lateral entorhinal regions.

We refer to the analysis in the supplement in the main manuscript on page 14, line 27:

“Note that a supplemental analysis of information processing in the cortical source regions showed indeed, specific object processing in perirhinal source regions (see appendix V). The lack of increased object processing in the anterior EC subregions and subiculum/CA1 border is thus likely not a result of increased noise in the object condition. Instead, increased object processing in perirhinal cortical source regions indicates subsequent convergence in entorhinal-hippocampal subregions, as hypothesized based on the updated cortical mapping scheme onto the EC.”

We describe the analysis in the appendix V as follows:

“To examine whether lower parameter estimates for object processing could be due to increased noise in this condition, we evaluated object and scene processing in the four cortical source regions. Therefore, we extracted parameter estimates for the object versus baseline and the scene versus baseline contrast from the retrosplenial and parahippocampal cortex and from perirhinal Area 36 and Area 35, respectively. All parameter estimates were extracted from the previously segmented regions of interests, coregistered to the individual EPI space.

Repeated-measures ANOVAs in both hemispheres showed a significant interaction effect between condition and region (right: F(3,93) = 60.4229; p <.001; left: F(3,93) = 47.3421; p <.001). Subsequent paired-samples T-tests show significantly more functional activity in the object than scene condition in Area 36 (bilateral: p_FDR_ <.001) and the left Area 35 (p_FDR_ =.0011). No significant difference between object and scene conditions is observed in the right Area 35 (right: p_FDR_ = 0.9821). There is a significant effect of more functional activity in the scene than object condition in the parahippocampal (bilateral: p_FDR_ <.001) and retrosplenial cortex (bilateral: p_FDR_ <.001, see Appendix 5 – Figure 1).

The increased object processing in adjacent cortical source regions indicates that noise differences across conditions are not likely to cause the lack of increased object processing within entorhinal seed regions and hippocampal subregions.”

20. No 'information flow' is actually assessed in this study. In my view, this would require a directed connectivity analysis such as dynamic causal modeling or transfer entropy. The research question or hypotheses should pertain to what is actually being done. For example, 'we predict functional connectivity between X-Y to reflect the structural connectivity described in rodents/ humans previously', and 'we expect specific sensitivity to scene stimuli in distal subiculum, because of connectivity to the MEC region in rodents'. This would be 'consistent with information flow' between those regions, but it is not directly or conclusively showing it.

We agree with this potential point of misunderstanding and changed our wording accordingly throughout the manuscript on page 2, line 12:

“Our results are consistent with transversal information-specific pathways”

Page 19, line 20:

“Third, our perspective was entirely functional and we cannot conclude on the directionality of our results. […] Note that as a first step towards an understanding of the system’s functional organization and to increase comparability with earlier studies, we assessed functional connectivity and information processing within the entorhinal-hippocampal circuitry with univariate methods.”

Page 21, line 7:

“Our high-resolution approach revealed unknown characteristics of functional connectivity and scene processing within the human entorhinal-hippocampal circuitry.”

21. The approach to segment the EC into sub-clusters based on known connectivity to other regions seems fine in general. However, this should be guided by gold-standard, a priori knowledge of entorhinal subregions such as the rodent MEC and LEC (or finer cyto/ myeloarchitectonic subdivisions, for example described in Kimer et al. 1997). In the present study, the authors select four cortical seed regions to segment entorhinal subregions based on functional connectivity, without providing appropriate justification how some of those regions would uniquely connect to specific, cytoarchitectonically-defined entorhinal subregions. For example, BA35, BA36, and PHC all project to the rodent LEC, and PHC also project (albeit weaker) to the MEC. Which a-priori defined entorhinal subregions should be uniquely identified with these four seed regions? Following the semipartial correlation analyses, clusters of EC voxels are then segmented and labeled posterior-medial, posterior-lateral, anterior-medial and anterior-lateral EC. Those names are then used quasi synonymously with coherent and established, cytoarchitectonically defined subregions such as the rodent LEC and MEC. This seems egregious, not only because of the questionable choice of initial seed regions, but more so because of the discontinuous topography of the segmentations shown in Figure 1. What is the ground-truth (cytoarchitecture, myeloarchitecture, tracing studies, gene expression) evidence for a salt-and-pepper organization of entorhinal subregions? The clusters can't correspond to coherent cytoarchitectonic regions, so why would they be referred to as such.

Thank you for raising that important concern. Indeed, we realized that our naming of entorhinal seed regions may potentially be misleading. Our findings capture functional connectivity but as we raised in the limitations, this does not necessarily correspond to anatomically defined regions. See page 20, line 21:

“Future research is needed to evaluate how the functionally derived entorhinal seeds in this study relate to histologically derived entorhinal subregions (Oltmer et al., 2022) or entorhinal subregions based on structural connectivity (Syversen et al., 2021). For a dedicated comparison of subregions, it is essential to pay close attention to the segmentation of the EC itself.”

This is also illustrated in a new paragraph of the limitations section on page 20, line 3:

“Fourth, our study was originally conducted within the assumption that (functional) connectivity profiles reveal functional subregions. Based on that approach, the medial EC is identified based on i.a. retrosplenial connectivity. We, therefore conclude a surprisingly anterior yet medial EC mapping of the retrosplenial cortex. This approach has been followed by Maass, Berron et al. (2015) and also in numerous anatomical connectivity studies in animals (see Witter et al., 2017). It is possible that species differences lead to our EC_RSC-based_ to be more anterior than one would expect based on animal studies. However, given that the medial subregion in the primate EC remains posterior (cf. posterior-medial EC homologue in Maass, Berron et al., 2015), another possibility is that our retrosplenial functional connectivity cluster maps onto the human anterior-lateral EC. Our data does not allow us to verify this latter option. It is unclear, however, why functional subregions in line with predictions from animal research can be identified for some cortical source-to-EC mappings (like the parahippocampal cortex) but not for others. In combination with closely matched histological or structural magnetic resonance imaging data, future work can further reveal the nature of retrosplenial mapping on the human EC.”

To prevent raising the impression that our seed regions relate to cytoarchitectonically defined regions, we have renamed the entorhinal seed regions according to their functional source regions throughout the manuscript on page 6, line 23:

“For the EC_PHC-based_ seed, the majority of voxels can roughly be described as clustering in the posterior-medial entorhinal portion, for the EC_RSC-based_ seed in the anterior-medial portion, for the EC_Area35-based_ seed in the anterior-lateral portion and for the EC_Area36-based_ seed in the posterior-lateral entorhinal portion (see appendix II for exact voxel counts). Note that both perirhinal-based entorhinal seeds extended along the anterior to posterior axis such that the EC_Area35-based_ progressed more along the outer EC (i.e. laterally, with a main focus anteriorly) and the EC_Area36-based_ along the inner EC (i.e. medially, with a main focus posteriorly, see Figure 1 and the medial reflection of the EC seeds). It is important to note that these are rough qualitative descriptions of the main clusters, without quantification or an established relationship to coherent cytoarchitectonic regions. We will therefore continue to refer to them as EC_RSC-based_, EC_PHC-based_, EC_Area35-based_ and EC_Area36-based_ seeds”

To explain why we initially opted the anatomical terms for the seed regions, we point out that we find a considerable topographical organization of our EC seed regions in four quadrants of the entorhinal cortex. We included this information now in appendix II:

“To assess the main location of each cortical source preferences within the EC, we cut the left and right EC in four quadrants. This was performed in T1 template space. First, the middle slice of all coronal slices that capture the EC was determined separately for each hemisphere. This slice was used to cut the EC in quadrants I, III and II, IV. Second, the middle slice of all axial slices that capture the EC was determined. This slice served to cut the EC in quadrants I, II and III, IV (see Appendix 2 – Figure 1). Note, to determine the most superior axial slice, the most posterior coronal level of the EC was used. Subsequently, we counted the number of voxels that have been assigned to each of the four cortical source regions after the initial functional connectivity analyses (that served to determined EC seeds). Averaged across hemispheres, most voxels assigned to the retrosplenial source are in EC quadrant I, most voxels assigned to the Area 35 source in EC quadrant II, most voxels assigned to the parahippocampal cortex in EC quadrant III and most voxels assigned to Area 36 in EC quadrant IV (see Appendix 2 – Table 1 for detailed voxel counts). Note that these quadrants do not refer to anatomically defined EC subregions.”

As our manuscript is an advancement of the Maass et al. (2015) publication, we opted for a comparable terminology. Therefore, we initially stretched their anterior-lateral (for the EC voxels preferentially connected to the perirhinal seed) and anterior-medial (for the EC voxels preferentially connected to the parahippocampal seed) terminology further and applied it to the additional two EC seed regions we acquire. We agree, however, that this may falsely imply more profound anatomical grounding of our results. Hence, our new naming of EC seed regions. We hope, this prevents further misunderstandings.

22. Why did the authors not simply use the ROIs from their previous identification of the human homologues of the MEC and LEC (Maass et al. 2015) to address their current research questions? In summary, this could be reconciled by a consistently used naming scheme for the entorhinal seeds that avoids confusion with cytoarchitectonic subregions.

We appreciate that comment. The human homologues of the MEC and LEC were defined within the concept of the two parallel streams hypothesis (in Maass et al., 2015). Each voxel was identified by either preferential functional connection to the perirhinal or the parahippocampal cortex. This approach neither acknowledged further cortical sources like the retrosplenial cortex nor did it allow to test unique mappings of the perirhinal and parahippocampal cortices. In addition, we opted to separate perirhinal Area 35 and Area 36 given reports about early cortical tau pathology in Alzheimer’s disease that appears specifically in Area 35 (Lace et al., 2009).

As we understand the concern regarding the anatomical correspondence of our entorhinal seed regions, we renamed them as follows:

EC_RSC-based_

EC_PHC-based_

EC_Area36-based_

EC_Area35-based_

23. In the results you describe data on proximal CA1, but you do not mention them anywhere in the paper explicitly. If your data do not allow you to functionally 'interpret' proximal CA1, you might come back to that in the discussion, state this and mention that in the rodent literature there is a gradient along transverse CA1 with more precise spatial information in proximal than in distal. Any ideas of why that does not seem to hold in humans? In particular on page 15, line 31 you make a very general statement about the transverse organization of CA1, so that might be the place to elaborate a bit more.

Thank you for that interesting point. We added the following discussion of that issue now to the manuscript on page 15, line 11:

“Regarding the human proximal CA1, a firm conclusion is limited with our data. First, the functional connectivity results varied between hemispheres. In both hemispheres, proximal CA1 showed a different connectivity profile compared to distal CA1. However, even though statistically not significant, the preferences at the group level indicated increased functional connectivity with the EC_PHC-based_ portion in the right but with the EC_Area35-based_ portion in the left hemisphere. Second, we do not prove similar information processing along the transversal CA1 axis. Instead, we find no significant difference in information processing along the transversal CA1 axis. As indicated in the previous paragraph, we cannot rule out that our object versus scene processing conditions may not have been sensitive enough to tackle functional differences in CA1. Thus, future research will have to identify defining characteristics of information processing along the transversal CA1 axis in a less constraint manner to allow conclusions on distinct information processing in proximal CA1.”

Future projects may evaluate longitudinal effects that potentially average out our analysis of the transversal axis. In addition, individual variability in the subregion’s borders along the longitudinal axis may be captured by analyses methods that can be applied independently of segmentation protocols and that focus on single-voxel results only.

Clinical relevance:24. The reported findings may have clinical relevance, but this is to be determined by future studies. An entire paragraph in the Introduction is dedicated to this topic (starting on page 4, line34). The authors state that the findings show that tau pathology spreads through 'functionally connected' regions. Functionally connected regions must also be structurally connected (through mono or polysynaptic connections). It seems the authors are insinuating that functional and structural connections are independent. The prediction of functional connectivity between parahippocampal, perirhinal and entorhinal cortices, and the subiculum is already abundantly well founded on previous findings of tracing studies in rodents and primates, and even the authors' previously published functional MRI findings (Maass et al. 2015). The fact that the spreading of tau pathology follows structural and functional connections provides no additional predictive value to inform the research question (in my understanding these are: (1) is there a specific functional connectivity pattern between regions, (2) do specific regions show differential fMRI activations for object vs scene stimuli). It appears unwarranted and misleading to portray the clinical findings as a formal motivation (meaning, previous findings that form the basis for the research hypotheses) for the present study – page 5, line 8. If this was not the intention, then I feel this needs to be formulated more clearly.

Thank you for raising that important concern. We agree, that the spreading of tau pathology does not provide additional predictive value to inform our research questions. However, we would like to emphasize that according to our view it makes sense from a clinical point of view to investigate our research questions. It has previously been shown that tau pathology may spread in an activity-dependent manner across connected regions (Adams et al., 2019; Berron et al., 2020; Berron et al., 2021; Franzmeier et al., 2020; Maass et al., 2019; Vogel et al., 2020). Therefore, knowledge on these potential routes and hence on functional connectivity and related information processing, as we observe it, is clinically relevant. Thus, we do not see the clinical literature as holding predictions (besides a potential different functional role for Area 35 and Area 36 due to differences in tau pathology here).

Still, in our view it appears logical to us to investigate functional architecture, even from a mere clinical perspective. Clinical research can rely on our basic findings for future, more precise clinical predictions. This was likewise the case for the previous Maass et al. (2015) publication (as well as Navarro Schröder et al., 2015) which informed subsequent publications in the clinical population by the functional architecture they revealed (e.g. Tran et al., 2022; Berron et al., 2022; Berron et al., 2021; Maass et al., 2020; Reagh et al., 2018; Olsen et al., 2017; Bastin et al., 2019; Yeung et al., 2019). Therefore, we do see the clinical literature as a strong motivator to investigate functional architecture. Hence, we opted to refer to this aspect and thereby also make our basic findings accessible to the clinical research community.

We hope, that this clarification helps to prevent any potential misunderstandings in this regard.

Moreover, we stress in the manuscript, that the clinical perspective is yet another important motivator for our study, but does not provide any predictive value. We therefore deleted the paragraph on our clinically-driven motivation in the introduction.

When interpreting our findings, we refer to the clinical relevance at the end of the Discussion section, when we give a more general outlook on the applications of our basic results on page 18, line 15:

“From a clinical research perspective, it is remarkable that the current functional connectivity pattern resembles the topology of early cortical tau pathology in Alzheimer’s disease (Lace et al., 2009). An influential hypothesis suggests tau progression in Alzheimer’s disease along functionally connected pathways in the human brain (Franzmeier et al., 2020; Vogel et al., 2020). Earliest cortical tau pathology in Alzheimer’s disease accumulates in perirhinal Area 35 (also referred to as transentorhinal region) and the anterior-lateral EC before it can be found along the subiculum/CA1 border (Braak and Braak , 1995; Berron et al., 2021; Kaufman et al., 2018; Lace et al., 2009). The topology of early tau pathology in Alzheimer’s disease thus mirrors the regions that we find biased towards EC_Area35-based_ connectivity (Braak and Braak, 1991; Lace et al., 2009; Roussarie et al., 2020). Tau pathology in Alzheimer’s disease is associated with memory impairment (Bejanin et al., 2017; Berron et al., 2021; Nelson et al., 2012) and information processing might be affected accordingly as reports have shown an association between Alzheimer’s related tau pathology and object memory in early disease stages (Berron et al., 2019; Maass et al., 2019). However, given our finding of activity patterns consistent with object – scene convergence in those subregions of the hippocampal-entorhinal circuitry that are affected by early tau pathology, object-in-scene memory tasks might have increased sensitivity to memory impairment. Moreover, both, the entorhinal portion based on retrosplenial connectivity (EC_RSC-based_) and the entorhinal portion based on Area 35 connectivity (EC_Area35-based_), are functionally connected to the subiculum/CA1 border. This overlapping functional connectivity pattern in the hippocampus might be a way along which tau and amyloid pathologies in Alzheimer’s disease could interact. This is consistent with early hypometabolism and cortical tau progression in the retrosplenial cortex and early amyloid in posterior parietal regions (Grothe et al., 2017; Palmqvist et al., 2017; Ziontz et al., 2021). The revealed functional connectivity and information processing profile may guide future hypotheses on the propagation of Alzheimer’s pathology and related functional and cognitive impairment.”

In case you continue to see a potentially misleading aspect in our introduction, please let us know as it is not our intention.